# A UNIFYING VIEW ON IMPLICIT BIAS IN TRAINING LINEAR NEURAL NETWORKS

**Chulhee Yun**[*]
MIT
chulheey@mit.edu

**Shankar Krishnan**
Google Research
skrishnan@google.com

**Hossein Mobahi**
Google Research
hmobahi@google.com

## ABSTRACT

We study the implicit bias of gradient flow (i.e., gradient descent with infinitesimal step size) on linear neural network training. We propose a tensor formulation of neural networks that includes fully-connected, diagonal, and convolutional networks as special cases, and investigate the linear version of the formulation called linear tensor networks. With this formulation, we can characterize the convergence direction of the network parameters as singular vectors of a tensor defined by the network. For $L$-layer linear tensor networks that are orthogonally decomposable, we show that gradient flow on separable classification finds a stationary point of the $\ell_{2/L}$ max-margin problem in a "transformed" input space defined by the network. For underdetermined regression, we prove that gradient flow finds a global minimum which minimizes a norm-like function that interpolates between weighted $\ell_1$ and $\ell_2$ norms in the transformed input space. Our theorems subsume existing results in the literature while removing standard convergence assumptions. We also provide experiments that corroborate our analysis.

## 1 INTRODUCTION

Overparametrized neural networks have infinitely many solutions that achieve zero training error, and such global minima have different generalization performance. Moreover, training a neural network is a high-dimensional nonconvex problem, which is typically intractable to solve. However, the success of deep learning indicates that first-order methods such as gradient descent or stochastic gradient descent (GD/SGD) not only (a) succeed in finding global minima, but also (b) are biased towards solutions that generalize well, which largely has remained a mystery in the literature.

To explain part (a) of the phenomenon, there is a growing literature studying the convergence of GD/SGD on overparametrized neural networks (e.g., Du et al. (2018a;b); Allen-Zhu et al. (2018); Zou et al. (2018); Jacot et al. (2018); Oymak & Soltanolkotabi (2020), and many more). There are also convergence results that focus on linear networks, without nonlinear activations (Bartlett et al., 2018; Arora et al., 2019a; Wu et al., 2019; Du & Hu, 2019; Hu et al., 2020). These results typically focus on the convergence of loss, hence do not address *which* of the many global minima is reached.

Another line of results tackles part (b), by studying the implicit bias or regularization of gradient-based methods on neural networks or related problems (Gunasekar et al., 2017; 2018a;b; Arora et al., 2018; Soudry et al., 2018; Ji & Telgarsky, 2019a; Arora et al., 2019b; Woodworth et al., 2020; Chizat & Bach, 2020; Gissin et al., 2020). These results have shown interesting progress that even without explicit regularization terms in the training objective, algorithms such as GD applied on neural networks have an *implicit bias* towards certain solutions among the many global minima. However, in proving such results, many results rely on *convergence assumptions* such as global convergence of loss to zero and/or directional convergence of parameters and gradients. Ideally, such convergence assumptions should be removed because they cannot be tested *a priori* and there are known examples where GD does not converge to global minima under certain initializations (Bartlett et al., 2018; Arora et al., 2019a).

---

[*]Based on work performed during internship at Google Research

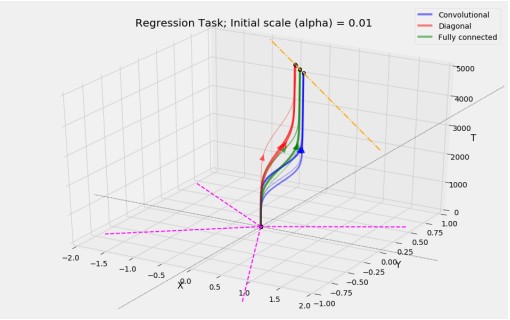 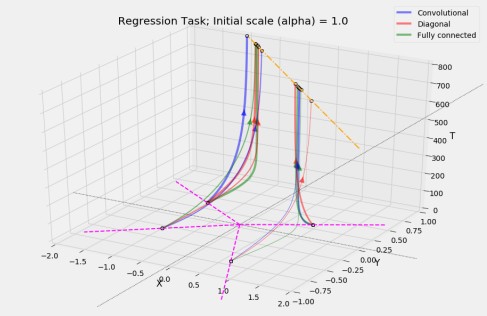

Figure 1: Gradient descent trajectories of linear coefficients of linear fully-connected, diagonal, and convolutional networks on a regression task, initialized with different initial scales $\alpha = 0.01, 1$. Networks are initialized at the same coefficients (circles on purple lines), but follow different trajectories due to implicit biases of networks induced from their architecture. The figures show that our theoretical predictions on limit points (circles on yellow line, the set of global minima) agree with the solution found by GD. For details of the experimental setup, see Section 6.

## 1.1 SUMMARY OF OUR CONTRIBUTIONS

We study the implicit bias of gradient flow (GD with infinitesimal step size) on linear neural networks. Following recent progress on this topic, we consider classification and regression problems that have multiple solutions with zero training error. Our analyses apply to a **general class of networks**, and prove **both convergence and implicit bias**, providing a more *complete characterization* of the algorithm trajectory without relying on convergence assumptions.

- We propose a general *tensor formulation* of nonlinear neural networks which includes many network architectures considered in the literature. In this paper, we focus on the linear version of this formulation (i.e., no nonlinear activations), called *linear tensor networks*.
- For linearly separable classification, we prove that linear tensor network parameters converge in direction to *singular vectors* of a tensor defined by the network. As a *corollary*, we show that linear fully-connected networks converge to the $\ell_2$ max-margin solution (Ji & Telgarsky, 2020).
- For separable classification, we further show that if the linear tensor network is orthogonally decomposable (Assumption 1), the gradient flow finds the $\ell_{2/\text{depth}}$ max-margin solution in the singular value space, leading the parameters to converge to the *top singular vectors* of the tensor when $\text{depth} = 2$. This theorem subsumes known results on linear convolutional networks and diagonal networks proved in Gunasekar et al. (2018b), *without* using convergence assumptions.
- For underdetermined linear regression, we study the limit points of gradient flow on orthogonally decomposable networks (Assumption 1), and provide a full characterization of the limit points. This theorem covers results on deep matrix sensing (Arora et al., 2019b) as a special case, and extends a similar recent result (Woodworth et al., 2020) to a broader class of networks.
- For underdetermined linear regression with deep linear fully-connected networks, we prove that the network converges to the minimum $\ell_2$ norm solutions as we scale the initialization to zero.
- Lastly, we present simple experiments that corroborate our theoretical analysis. Figure 1 shows that our predictions of limit points match with solutions found by GD.

## 2 PROBLEM SETTINGS AND RELATED WORKS

We first define notation used in the paper. Given a positive integer $a$, let $[a] := \{1, \ldots, a\}$. We use $\boldsymbol{I}_d$ to denote the $d \times d$ identity matrix. Given a matrix $\boldsymbol{A}$, we use $\text{vec}(\boldsymbol{A})$ to denote its vectorization, i.e., the concatenation of all columns of $\boldsymbol{A}$. For two vectors $\boldsymbol{a}$ and $\boldsymbol{b}$, let $\boldsymbol{a} \otimes \boldsymbol{b}$ be their tensor product, $\boldsymbol{a} \odot \boldsymbol{b}$ be their element-wise product, and $\boldsymbol{a}^{\odot k}$ be the element-wise $k$-th power of $\boldsymbol{a}$. Given an order-$L$ tensor $\boldsymbol{\mathsf{A}} \in \mathbb{R}^{k_1 \times \cdots \times k_L}$, we use $[\boldsymbol{\mathsf{A}}]_{j_1, \ldots, j_L}$ to denote the $(j_1, j_2, \ldots, j_L)$-th element of $\boldsymbol{\mathsf{A}}$, where $j_l \in [k_l]$ for all $l \in [L]$. In element indexing, we use $\cdot$ to denote all indices in the corresponding dimension, and $a : b$ to denote all indices from $a$ to $b$. For example, for a matrix $\boldsymbol{A}$, $[\boldsymbol{A}]_{\cdot, 4:6}$ denotes a submatrix that consists of 4th–6th columns of $\boldsymbol{A}$. The square bracket notation for indexing overloads

with $[a]$ when $a \in \mathbb{N}$, but they will be distinguishable from the context. Since element indices start from 1, we re-define the modulo operation $a \bmod d := a - \lfloor \frac{a-1}{d} \rfloor d \in [d]$ for $a > 0$. We use $e_j^k$ to denote the $j$-th stardard basis vector of the vector space $\mathbb{R}^k$. Lastly, we define the multilinear multiplication between a tensor and linear maps, which can be viewed as a generalization of left- and right-multiplication on a matrix. Given a tensor $\mathbf{A} \in \mathbb{R}^{k_1 \times \cdots \times k_L}$ and linear maps $\boldsymbol{B}_l \in \mathbb{R}^{p_l \times k_l}$ for $l \in [L]$, we define the multilinear multiplication $\circ$ between them as

$$\mathbf{A} \circ (\boldsymbol{B}_1^T, \boldsymbol{B}_2^T, \ldots, \boldsymbol{B}_L^T) = \sum\nolimits_{j_1,\ldots,j_L} [\mathbf{A}]_{j_1,\ldots,j_L} (e_{j_1}^{k_1} \otimes \cdots \otimes e_{j_L}^{k_L}) \circ (\boldsymbol{B}_1^T, \ldots, \boldsymbol{B}_L^T)$$
$$:= \sum\nolimits_{j_1,\ldots,j_L} [\mathbf{A}]_{j_1,\ldots,j_L} (\boldsymbol{B}_1 e_{j_1}^{k_1} \otimes \cdots \otimes \boldsymbol{B}_L e_{j_L}^{k_L}) \in \mathbb{R}^{p_1 \times \cdots \times p_L}.$$

## 2.1 PROBLEM SETTINGS

We are given a dataset $\{(\boldsymbol{x}_i, y_i)\}_{i=1}^n$, where $\boldsymbol{x}_i \in \mathbb{R}^d$ and $y_i \in \mathbb{R}$. We let $\boldsymbol{X} \in \mathbb{R}^{n \times d}$ and $\boldsymbol{y} \in \mathbb{R}^n$ be the data matrix and the label vector, respectively. We study binary classification and linear regression in this paper, focusing on the settings where there exist *many* global solutions. For binary classification, we assume $y_i \in \{\pm 1\}$ and that the data is separable: there exists a unit vector $\boldsymbol{z}$ and a constant $\gamma > 0$ such that $y_i \boldsymbol{x}_i^T \boldsymbol{z} \geq \gamma$ for all $i \in [n]$. For regression, we consider the underdetermined case ($n \leq d$) where there are many parameters $\boldsymbol{z} \in \mathbb{R}^d$ such that $\boldsymbol{X}\boldsymbol{z} = \boldsymbol{y}$. Throughout the paper, we assume that $\boldsymbol{X}$ has full row rank.

We use $f(\cdot; \boldsymbol{\Theta}) : \mathbb{R}^d \to \mathbb{R}$ to denote a neural network parametrized by $\boldsymbol{\Theta}$. Given the network and the dataset, we consider minimizing the training loss $\mathcal{L}(\boldsymbol{\Theta}) := \sum_{i=1}^n \ell(f(\boldsymbol{x}_i; \boldsymbol{\Theta}), y_i)$ over $\boldsymbol{\Theta}$. Following previous results (e.g., Lyu & Li (2020); Ji & Telgarsky (2020)), we use the exponential loss $\ell(\hat{y}, y) = \exp(-\hat{y}y)$ for classification problems. For regression, we use the squared error loss $\ell(\hat{y}, y) = \frac{1}{2}(\hat{y} - y)^2$. On the algorithm side, we minimize $\mathcal{L}$ using gradient flow, which can be viewed as GD with infinitesimal step size. The gradient flow dynamics is defined as $\frac{d}{dt}\boldsymbol{\Theta} = -\nabla_{\boldsymbol{\Theta}}\mathcal{L}(\boldsymbol{\Theta})$.

## 2.2 RELATED WORKS

**Gradient flow/descent in separable classification.** For linear models $f(\boldsymbol{x}; \boldsymbol{z}) = \boldsymbol{x}^T \boldsymbol{z}$ with separable data, Soudry et al. (2018) show that the GD run on $\mathcal{L}$ drives $\|\boldsymbol{z}\|$ to $\infty$, but $\boldsymbol{z}$ converges in direction to the $\ell_2$ max-margin classifier. The limit direction of $\boldsymbol{z}$ is aligned with the solution of

$$\text{minimize}_{\boldsymbol{z} \in \mathbb{R}^d} \quad \|\boldsymbol{z}\| \quad \text{subject to} \quad y_i \boldsymbol{x}_i^T \boldsymbol{z} \geq 1 \text{ for } i \in [n], \tag{1}$$

where the norm in the cost is the $\ell_2$ norm. Nacson et al. (2019b;c); Gunasekar et al. (2018a); Ji & Telgarsky (2019b;c) extend these results to other (stochastic) algorithms and non-separable settings.

Gunasekar et al. (2018b) study the same problem on linear neural networks and show that GD exhibits different implicit bias depending on the architecture. The authors show that the linear coefficients of the network converges in direction to the solution of (1) with different norms: $\ell_2$ norm for linear fully-connected networks, $\ell_{2/L}$ (quasi-)norm for diagonal networks, and DFT-domain $\ell_{2/L}$ (quasi-)norm for convolutional networks with full-length filters. Here, $L$ denotes the depth. We note that Gunasekar et al. (2018b) assume that GD globally minimizes the loss, and the network parameters and the gradient with respect to the linear coefficients converge in direction. Subsequent results (Ji & Telgarsky, 2019a; 2020) remove such assumptions for linear fully-connected networks.

A recent line of results (Nacson et al., 2019a; Lyu & Li, 2020; Ji & Telgarsky, 2020) studies general homogeneous models and show divergence of parameters to infinity, monotone increase of smoothed margin, directional convergence and alignment of parameters (see Section 4 for details). Lyu & Li (2020) also characterize the limit direction of parameters as the KKT point of a nonconvex max-margin problem similar to (1), but this characterization does not provide useful insights for the functions $f(\cdot; \boldsymbol{\Theta})$ represented by specific architectures, because the formulation is in the parameter space $\boldsymbol{\Theta}$. Also, these results require that gradient flow/descent has already reached 100% training accuracy. Although we study a more restrictive set of networks (i.e., deep linear), we provide a more complete characterization of the implicit bias for the functions $f(\cdot; \boldsymbol{\Theta})$, without assuming 100% training accuracy.

**Gradient flow/descent in linear regression.** It is known that for linear models $f(\boldsymbol{x}; \boldsymbol{z}) = \boldsymbol{x}^T \boldsymbol{z}$, GD converges to the global minimum that is closest in $\ell_2$ distance to the initialization (see e.g.,

Gunasekar et al. (2018a)). However, relatively less is known for deep networks, even for linear networks. This is partly because the parameters do not diverge to infinity, hence making limit points highly dependent on the initialization; this dependency renders analysis difficult. A related problem of matrix sensing aims to minimize $\sum_{i=1}^{n}(y_i - \langle \boldsymbol{A}_i, \boldsymbol{W}_1 \cdots \boldsymbol{W}_L \rangle)^2$ over $\boldsymbol{W}_1, \ldots, \boldsymbol{W}_L \in \mathbb{R}^{d \times d}$. It is shown in Gunasekar et al. (2017); Arora et al. (2019b) that if the sensor matrices $\boldsymbol{A}_i$ commute and we initialize all $\boldsymbol{W}_l$'s to $\alpha \boldsymbol{I}$, GD finds the minimum nuclear norm solution as $\alpha \to 0$.

Chizat et al. (2019) show that if a network is zero at initialization, and we scale the network output by a factor of $\alpha \to \infty$, then the GD dynamics enters a "lazy regime" where the network behaves like a first-order approximation at its initialization, as also seen in results studying kernel approximations of neural networks and convergence of GD in the corresponding RKHS (e.g., Jacot et al. (2018)).

Woodworth et al. (2020) study linear regression with a diagonal network of the form $f(\boldsymbol{x}; \boldsymbol{w}_+, \boldsymbol{w}_-) = \boldsymbol{x}^T(\boldsymbol{w}_+^{\odot L} - \boldsymbol{w}_-^{\odot L})$, where $\boldsymbol{w}_+$ and $\boldsymbol{w}_-$ are identically initialized $\boldsymbol{w}_+(0) = \boldsymbol{w}_-(0) = \alpha \bar{\boldsymbol{w}}$. The authors show that the global minimum reached by GD minimizes a norm-like function which interpolates between (weighted) $\ell_1$ norm ($\alpha \to 0$) and $\ell_2$ norm ($\alpha \to \infty$). In our paper, we consider a more general class of orthogonally decomposable networks, and obtain similar results interpolating between weighted $\ell_1$ and $\ell_2$ norms. We also remark that our results include the results in Arora et al. (2019b) as a special case, and we do not assume convergence to global minima, as done in Gunasekar et al. (2017); Arora et al. (2019b); Woodworth et al. (2020).

## 3 TENSOR FORMULATION OF NEURAL NETWORKS

In this section, we present a general tensor formulation of neural networks. Given an input $\boldsymbol{x} \in \mathbb{R}^d$, the network uses a linear map $\mathsf{M}$ that maps $\boldsymbol{x}$ to an order-$L$ tensor $\mathsf{M}(\boldsymbol{x}) \in \mathbb{R}^{k_1 \times \cdots \times k_L}$, where $L \geq 2$. Using parameters $\boldsymbol{v}_l \in \mathbb{R}^{k_l}$ and activation $\phi$, the network computes its layers as the following:

$$\mathsf{H}_1(\boldsymbol{x}) = \phi\left(\mathsf{M}(\boldsymbol{x}) \circ (\boldsymbol{v}_1, \boldsymbol{I}_{k_2}, \ldots, \boldsymbol{I}_{k_L})\right) \in \mathbb{R}^{k_2 \times \cdots \times k_L},$$
$$\mathsf{H}_l(\boldsymbol{x}) = \phi\left(\mathsf{H}_{l-1}(\boldsymbol{x}) \circ (\boldsymbol{v}_l, \boldsymbol{I}_{k_{l+1}}, \ldots, \boldsymbol{I}_{k_L})\right) \in \mathbb{R}^{k_{l+1} \times \ldots, k_L}, \quad \text{for } l = 2, \ldots, L-1, \quad (2)$$
$$f(\boldsymbol{x}; \boldsymbol{\Theta}) = \mathsf{H}_{L-1}(\boldsymbol{x}) \circ \boldsymbol{v}_L \in \mathbb{R}.$$

We use $\boldsymbol{\Theta}$ to denote the collection of all parameters $(\boldsymbol{v}_1, \ldots, \boldsymbol{v}_L)$. We call $\mathsf{M}(\boldsymbol{x})$ the *data tensor*. Although this new formulation may look a bit odd in the first glance, it is general enough to capture many network architectures considered in the literature, including fully-connected networks, diagonal networks, and circular convolutional networks. We formally define these architectures below.

**Diagonal networks.** An $L$-layer diagonal network is written as

$$f_{\text{diag}}(\boldsymbol{x}; \boldsymbol{\Theta}_{\text{diag}}) = \phi(\cdots \phi(\phi(\boldsymbol{x} \odot \boldsymbol{w}_1) \odot \boldsymbol{w}_2) \cdots \odot \boldsymbol{w}_{L-1})^T \boldsymbol{w}_L, \quad (3)$$

where $\boldsymbol{w}_l \in \mathbb{R}^d$ for $l \in [L]$. The representation of $f_{\text{diag}}$ as the tensor form (2) is straightforward. Let $\mathsf{M}_{\text{diag}}(\boldsymbol{x}) \in \mathbb{R}^{d \times \cdots \times d}$ have $[\mathsf{M}_{\text{diag}}(\boldsymbol{x})]_{j,j,\ldots,j} = [\boldsymbol{x}]_j$, while all the remaining entries of $\mathsf{M}_{\text{diag}}(\boldsymbol{x})$ are set to zero. We can set $\boldsymbol{v}_l = \boldsymbol{w}_l$ for all $l$, and $\mathsf{M} = \mathsf{M}_{\text{diag}}$ to verify that (2) and (3) are equivalent.

**Circular convolutional networks.** The tensor formulation (2) includes convolutional networks

$$f_{\text{conv}}(\boldsymbol{x}; \boldsymbol{\Theta}_{\text{conv}}) = \phi(\cdots \phi(\phi(\boldsymbol{x} \star \boldsymbol{w}_1) \star \boldsymbol{w}_2) \cdots \star \boldsymbol{w}_{L-1})^T \boldsymbol{w}_L, \quad (4)$$

where $\boldsymbol{w}_l \in \mathbb{R}^{k_l}$ with $k_l \leq d$ and $k_L = d$, and $\star$ defines the circular convolution: for any $\boldsymbol{a} \in \mathbb{R}^d$ and $\boldsymbol{b} \in \mathbb{R}^k$ ($k \leq d$), we have $\boldsymbol{a} \star \boldsymbol{b} \in \mathbb{R}^d$ defined as $[\boldsymbol{a} \star \boldsymbol{b}]_i = \sum_{j=1}^{k}[\boldsymbol{a}]_{(i+j-1) \bmod d}[\boldsymbol{b}]_j$, for $i \in [d]$. Define $\mathsf{M}_{\text{conv}}(\boldsymbol{x}) \in \mathbb{R}^{k_1 \times \cdots \times k_L}$ as $[\mathsf{M}_{\text{conv}}(\boldsymbol{x})]_{j_1, j_2, \ldots, j_L} = [\boldsymbol{x}]_{(\sum_{l=1}^{L} j_l - L + 1) \bmod d}$ for $j_l \in [k_l]$, $l \in [L]$. Setting $\boldsymbol{v}_l = \boldsymbol{w}_l$ and $\mathsf{M} = \mathsf{M}_{\text{conv}}$, we can verify that (2) and (4) are identical.

**Fully-connected networks.** An $L$-layer fully-connected network is defined as

$$f_{\text{fc}}(\boldsymbol{x}; \boldsymbol{\Theta}_{\text{fc}}) = \phi(\cdots \phi(\phi(\boldsymbol{x}^T \boldsymbol{W}_1) \boldsymbol{W}_2) \cdots \boldsymbol{W}_{L-1}) \boldsymbol{w}_L, \quad (5)$$

where $\boldsymbol{W}_l \in \mathbb{R}^{d_l \times d_{l+1}}$ for $l \in [L-1]$ (we use $d_1 = d$) and $\boldsymbol{w}_L \in \mathbb{R}^{d_L}$. One can represent $f_{\text{fc}}$ as the tensor form (2) by defining parameters $\boldsymbol{v}_l = \text{vec}(\boldsymbol{W}_l)$ for $l \in [L-1]$ and $\boldsymbol{v}_L = \boldsymbol{w}_L$, and constructing the tensor $\mathsf{M}_{\text{fc}}(\boldsymbol{x})$ by a recursive "block diagonal" manner. For example, if $L = 2$, we can define $\mathsf{M}_{\text{fc}}(\boldsymbol{x}) \in \mathbb{R}^{d_1 d_2 \times d_2}$ to be the Kronecker product of $\boldsymbol{I}_{d_2}$ and $\boldsymbol{x}$. For deeper networks, we defer the full description of $\mathsf{M}_{\text{fc}}(\boldsymbol{x})$ to Appendix B.

**Our focus: linear tensor networks.** Throughout this section, we have used the activation $\phi$ to motivate our tensor formulation (2) for neural networks with nonlinear activations. For the remaining of the paper, we study the case whose activation is *linear*, i.e., $\phi(t) = t$. In this case,

$$f(\boldsymbol{x}; \boldsymbol{\Theta}) = \mathbf{M}(\boldsymbol{x}) \circ (\boldsymbol{v}_1, \boldsymbol{v}_2, \ldots, \boldsymbol{v}_L). \tag{6}$$

We will refer to (6) as *linear tensor networks*, where "linear" is to indicate that the activation is linear. Note that as a function of parameters $\boldsymbol{v}_1, \ldots, \boldsymbol{v}_L$, $f(\boldsymbol{x}; \boldsymbol{\Theta})$ is in fact multilinear. We also remark that when depth $L = 2$, the data tensor $\mathbf{M}(\boldsymbol{x})$ is a $k_1 \times k_2$ matrix and the network formulation boils down to $f(\boldsymbol{x}; \boldsymbol{\Theta}) = \boldsymbol{v}_1^T \mathbf{M}(\boldsymbol{x}) \boldsymbol{v}_2$.

Since the data tensor $\mathbf{M}(\boldsymbol{x})$ is a linear function of $\boldsymbol{x}$, the linear tensor network is also a linear function of $\boldsymbol{x}$. Thus, the output of the network can also be written as $f(\boldsymbol{x}; \boldsymbol{\Theta}) = \boldsymbol{x}^T \boldsymbol{\beta}(\boldsymbol{\Theta})$, where $\boldsymbol{\beta}(\boldsymbol{\Theta}) \in \mathbb{R}^d$ denotes the *linear coefficients* computed as a function of the network parameters $\boldsymbol{\Theta}$. Since the linear tensor network $f(\boldsymbol{x}; \boldsymbol{\Theta})$ is linear in $\boldsymbol{x}$, the expressive power of $f$ is at best a linear model $\boldsymbol{x} \mapsto \boldsymbol{x}^T \boldsymbol{z}$. However, even though the models have the same expressive power, their architectural differences lead to different implicit biases in training, which is the focus of our investigation in this paper. Studying separable classification and underdetermined regression is useful for highlighting such biases because there are *infinitely many* coefficients that perfectly classify or fit the dataset.

For our linear tensor network, the evolution of the parameters $\boldsymbol{v}_l$ via gradient flow reads

$$\dot{\boldsymbol{v}}_l = -\nabla_{\boldsymbol{v}_l} \mathcal{L}(\boldsymbol{\Theta}) = -\sum_{i=1}^{n} \ell'(f(\boldsymbol{x}_i; \boldsymbol{\Theta}), y_i) \mathbf{M}(\boldsymbol{x}_i) \circ (\boldsymbol{v}_1, \ldots, \boldsymbol{v}_{l-1}, \boldsymbol{I}_{k_l}, \boldsymbol{v}_{l+1}, \ldots, \boldsymbol{v}_L)$$
$$= \mathbf{M}(-\boldsymbol{X}^T \boldsymbol{r}) \circ (\boldsymbol{v}_1, \ldots, \boldsymbol{v}_{l-1}, \boldsymbol{I}_{k_l}, \boldsymbol{v}_{l+1}, \ldots, \boldsymbol{v}_L), \ \ \forall l \in [L],$$

where we initialize $\boldsymbol{v}_l(0) = \alpha \bar{\boldsymbol{v}}_l$, for $l \in [L]$. We refer to $\alpha$ and $\bar{\boldsymbol{v}}_l$ as the *initial scale* and *initial direction*, respectively. We note that we do not restrict $\bar{\boldsymbol{v}}_l$'s to be unit vectors, in order to allow different scaling (at initialization) over different layers. The vector $\boldsymbol{r} \in \mathbb{R}^n$ is the *residual vector*, and each component of $\boldsymbol{r}$ is defined as

$$[\boldsymbol{r}]_i = \ell'(f(\boldsymbol{x}_i; \boldsymbol{\Theta}), y_i) = \begin{cases} -y_i \exp(-y_i f(\boldsymbol{x}_i; \boldsymbol{\Theta})) & \text{for classification,} \\ f(\boldsymbol{x}_i; \boldsymbol{\Theta}) - y_i & \text{for regression.} \end{cases} \tag{7}$$

## 4 IMPLICIT BIAS OF GRADIENT FLOW IN SEPARABLE CLASSIFICATION

In this section, we present our results on the implicit bias of gradient flow in binary classification with linearly separable data. Recent papers (Lyu & Li, 2020; Ji & Telgarsky, 2020) on this separable classification setup prove that after 100% training accuracy has been achieved by gradient flow (along with other technical conditions), the parameters of $L$-homogeneous models diverge to infinity, while converging in direction that aligns with the direction of the negative gradient. Mathematically,

$$\lim_{t \to \infty} \|\boldsymbol{\Theta}(t)\| = \infty, \quad \lim_{t \to \infty} \frac{\boldsymbol{\Theta}(t)}{\|\boldsymbol{\Theta}(t)\|} = \boldsymbol{\Theta}^\infty, \quad \lim_{t \to \infty} \frac{\boldsymbol{\Theta}(t)^T \nabla_{\boldsymbol{\Theta}} \mathcal{L}(\boldsymbol{\Theta}(t))}{\|\boldsymbol{\Theta}(t)\| \|\nabla_{\boldsymbol{\Theta}} \mathcal{L}(\boldsymbol{\Theta}(t))\|} = -1.$$

Since the linear tensor network satisfies the technical assumptions in the prior works, we apply these results to our setting and develop a new characterization of the limit directions of the parameters. Here, we present theorems on separable classification with general linear tensor networks. Corollaries for specific networks are deferred to Appendix A.

### 4.1 LIMIT DIRECTIONS OF PARAMETERS ARE SINGULAR VECTORS

Consider the singular value decomposition (SVD) of a matrix $\boldsymbol{A} = \sum_{j=1}^{m} s_j (\boldsymbol{u}_j \otimes \boldsymbol{v}_j)$, where $m$ is the rank of $\boldsymbol{A}$. Note that the tuples $(\boldsymbol{u}_j, \boldsymbol{v}_j, s_j)$ are solutions to the system of equations $s\boldsymbol{u} = \boldsymbol{A}\boldsymbol{v}$ and $s\boldsymbol{v} = \boldsymbol{A}^T \boldsymbol{u}$. Lim (2005) generalizes this definition of singular vectors and singular values to higher-order tensors: given an order-$L$ tensor $\mathbf{A} \in \mathbb{R}^{k_1 \times \cdots \times k_L}$, we define the singular vectors $\boldsymbol{u}_1, \boldsymbol{u}_2, \ldots, \boldsymbol{u}_L$ and singular value $s$ to be the solution of the following system of equations:

$$s\boldsymbol{u}_l = \mathbf{A} \circ (\boldsymbol{u}_1, \ldots, \boldsymbol{u}_{l-1}, \boldsymbol{I}_{k_l}, \boldsymbol{u}_{l+1}, \ldots, \boldsymbol{u}_L), \text{ for } l \in [L]. \tag{8}$$

Using the definition of the singular vectors of tensors, we can characterize the limit direction of parameters after reaching 100% training accuracy. In Appendix C, we prove the following:

**Theorem 1.** *Assume that the gradient flow satisfies $\mathcal{L}(\Theta(t_0)) < 1$ for some $t_0 \geq 0$ and $\boldsymbol{X}^T \boldsymbol{r}(t)$ converges in direction, say $\boldsymbol{u}^\infty := \lim_{t\to\infty} \frac{\boldsymbol{X}^T \boldsymbol{r}(t)}{\|\boldsymbol{X}^T \boldsymbol{r}(t)\|_2}$. Then, $\boldsymbol{v}_1, \ldots, \boldsymbol{v}_L$ converge to the singular vectors of $\mathsf{M}(-\boldsymbol{u}^\infty)$.*

For this theorem, we make some convergence assumptions, because the network is fully general; this is the *only* result where we assume convergence. It fact, for the special case of linear fully-connected networks, the directional convergence assumption is *not* required, and the linear coefficients $\boldsymbol{\beta}_{\text{fc}}(\Theta_{\text{fc}})$ converge in direction to the $\ell_2$ max-margin classifier. We state this corollary in Appendix A.1; this result also appears in Ji & Telgarsky (2020), but we provide an alternative proof.

## 4.2 LIMIT DIRECTIONS FOR ORTHOGONALLY DECOMPOSABLE NETWORKS

Admittedly, Theorem 1 is not a *full* characterization of the limit directions, because there are usually multiple solutions that satisfy (8). For example, in case of $L = 2$, the data tensor $\mathsf{M}(-\boldsymbol{u}^\infty)$ is a matrix and the number of possible limit directions (up to scaling) of $(\boldsymbol{v}_1, \boldsymbol{v}_2)$ is at least the rank of $\mathsf{M}(-\boldsymbol{u}^\infty)$. Singular vectors of high order tensors are much less understood than the matrix counterparts, and are much harder to deal with. Although their existence is implied from the variational formulation (Lim, 2005), they are intractable to compute. Testing if a given number is a singular value, approximating the corresponding singular vectors, and computing the best rank-1 approximation are all NP-hard (Hillar & Lim, 2013); let alone orthogonal decompositions.

Given this intractability, it might be reasonable to make some assumptions on the "structure" of the data tensor $\mathsf{M}(\boldsymbol{x})$, so that they are easier to handle. The following assumption defines a class of *orthogonally decomposable* data tensors, which includes **linear diagonal networks** and **linear full-length convolutional networks** as special cases (for the proof, see Appendix D.2 and D.3).

**Assumption 1.** *For the data tensor $\mathsf{M}(\boldsymbol{x}) \in \mathbb{R}^{k_1 \times \cdots \times k_L}$ of a linear tensor network (6), there exist a full column rank matrix $\boldsymbol{S} \in \mathbb{C}^{m \times d}$ ($d \leq m \leq \min_l k_l$) and matrices $\boldsymbol{U}_1 \in \mathbb{C}^{k_1 \times m}, \ldots, \boldsymbol{U}_L \in \mathbb{C}^{k_L \times m}$ such that $\boldsymbol{U}_l^H \boldsymbol{U}_l = \boldsymbol{I}_m$ for all $l \in [L]$, and the data tensor $\mathsf{M}(\boldsymbol{x})$ can be written as*

$$\mathsf{M}(\boldsymbol{x}) = \sum_{j=1}^m [\boldsymbol{S}\boldsymbol{x}]_j ([\boldsymbol{U}_1]_{\cdot,j} \otimes [\boldsymbol{U}_2]_{\cdot,j} \otimes \cdots \otimes [\boldsymbol{U}_L]_{\cdot,j}). \tag{9}$$

In this assumption, we allow $\boldsymbol{U}_1, \ldots, \boldsymbol{U}_L$ and $\boldsymbol{S}$ to be complex matrices, although $\mathsf{M}(\boldsymbol{x})$ and parameters $\boldsymbol{v}_l$ stay real, as defined earlier. For a complex matrix $\boldsymbol{A}$, we use $\boldsymbol{A}^*$ to denote its entry-wise complex conjugate, $\boldsymbol{A}^T$ to denote its transpose (without conjugating), and $\boldsymbol{A}^H$ to denote its conjugate transpose. In case of $L = 2$, Assumption 1 requires that the data tensor $\mathsf{M}(\boldsymbol{x})$ (now a matrix) has singular value decomposition $\mathsf{M}(\boldsymbol{x}) = \boldsymbol{U}_1 \operatorname{diag}(\boldsymbol{S}\boldsymbol{x})\boldsymbol{U}_2^T$; i.e., the left and right singular vectors are independent of $\boldsymbol{x}$, and the singular values are linear in $\boldsymbol{x}$. Using Assumption 1, the following theorem characterizes the limit directions.

**Theorem 2.** *Suppose a linear tensor network satisfies Assumption 1. If there exists $\lambda > 0$ such that the initial directions $\bar{\boldsymbol{v}}_1, \ldots, \bar{\boldsymbol{v}}_L$ of the network parameters satisfy $|[\boldsymbol{U}_l^T \bar{\boldsymbol{v}}_l]_j|^2 - |[\boldsymbol{U}_L^T \bar{\boldsymbol{v}}_L]_j|^2 \geq \lambda$ for all $l \in [L-1]$ and $j \in [m]$, then $\boldsymbol{\beta}(\Theta(t))$ converges in a direction that aligns with $\boldsymbol{S}^T \boldsymbol{\rho}^\infty$, where $\boldsymbol{\rho}^\infty \in \mathbb{C}^m$ denotes a stationary point of the following optimization problem*

$$\operatorname{minimize}_{\boldsymbol{\rho} \in \mathbb{C}^m} \quad \|\boldsymbol{\rho}\|_{2/L} \quad \text{subject to} \quad y_i \boldsymbol{x}_i^T \boldsymbol{S}^T \boldsymbol{\rho} \geq 1, \ \forall i \in [n].$$

*If $\boldsymbol{S}$ is invertible, then $\boldsymbol{\beta}(\Theta(t))$ converges in a direction that aligns with a stationary point $\boldsymbol{z}^\infty$ of*

$$\operatorname{minimize}_{\boldsymbol{z} \in \mathbb{R}^d} \quad \|\boldsymbol{S}^{-T} \boldsymbol{z}\|_{2/L} \quad \text{subject to} \quad y_i \boldsymbol{x}_i^T \boldsymbol{z} \geq 1, \ \forall i \in [n].$$

Theorem 2 shows that the gradient flow finds sparse $\boldsymbol{\rho}^\infty$ that minimizes the $\ell_{2/L}$ norm in the "singular value space," where the data points $\boldsymbol{x}_i$ are transformed into vectors $\boldsymbol{S}\boldsymbol{x}_i$ consisting of singular values of $\mathsf{M}(\boldsymbol{x}_i)$. Also, the proof of Theorem 2 reveals that in case of $L = 2$, the parameters $\boldsymbol{v}_l(t)$ in fact converge to the *top* singular vectors of the data tensor $\mathsf{M}(-\boldsymbol{X}^T \boldsymbol{r})$; thus, compared to Theorem 1, we have a more complete characterization of "which" singular vectors to converge to.

The proof of Theorem 2 is in Appendix D. Since the orthogonal decomposition (Assumption 1) of $\mathsf{M}(\boldsymbol{x})$ tells us that the singular vectors $\mathsf{M}(\boldsymbol{x})$ in $\boldsymbol{U}_1, \ldots, \boldsymbol{U}_L$ are independent of $\boldsymbol{x}$, we can transform the network parameters $\boldsymbol{v}_l$ to $\boldsymbol{U}_l^T \boldsymbol{v}_l$ and show that the network behaves similar to a linear diagonal network. This observation comes in handy in the characterization of limit directions.

**Remark 1** (Necessity of initialization assumptions). In order to remove the assumption that the loss converges to zero, at least some condition on initialization is *necessary*, because there are examples showing non-convergence of gradient flow for certain initializations (Bartlett et al., 2018; Arora et al., 2019a). In our theorems, we pose assumptions on initial directions $\bar{v}_l$ that are sufficient conditions for the loss $\mathcal{L}(\boldsymbol{\Theta}(t))$ to converge to zero. Although such sufficient conditions are "stronger" than assuming $\mathcal{L}(\boldsymbol{\Theta}(t)) \to 0$, they are useful because they can be easily checked *a priori*, i.e., before running gradient flow. We note an important fact that in Theorems 2 and onwards, the conditions on initialization are used **solely to prove convergence** of the loss to zero, and **our statements on the implicit bias hold whenever the loss converges to zero**, even for initializations that do not satisfy our conditions. In addition, we argue that our assumptions are not too restrictive; $\lambda$ can be arbitrarily small, so the conditions are satisfied *with probability 1* if we set $\bar{v}_L = \mathbf{0}$ and randomly sample other $\bar{v}_l$'s. Setting one layer to zero to prove convergence is also studied in Wu et al. (2019). Lastly, the condition that $\bar{v}_L$ is "small" can be replaced with any layer; e.g., convergence still holds if $|[\boldsymbol{U}_l^T \bar{v}_l]_j|^2 - |[\boldsymbol{U}_1^T \bar{v}_1]_j|^2 \geq \lambda$ for all $l = 2, \dots, L$ and $j \in [m]$.

**Remark 2** (Comparison to existing results). Theorem 2 leads to corollaries (stated in Appendix A.2) on linear diagonal and full-length convolutional networks, showing that diagonal (or convolutional) networks converge to the stationary point of the max-margin problem with respect to the $\ell_{2/L}$ norm (or DFT-domain $\ell_{2/L}$ norm). Theorem 2 recovers the results in Gunasekar et al. (2018b) without relying on assumptions such as directional convergence of parameters and gradients.

**Remark 3** (Implications to architecture design). Theorem 2 shows that the gradient flow finds a solution that is sparse in a "transformed" input space where all data points are transformed with $\boldsymbol{S}$. This implies something interesting about architecture design: if the sparsity of the solution under a certain linear transformation $\boldsymbol{T}$ is needed, one can design a network using Assumption 1 by setting $\boldsymbol{S} = \boldsymbol{T}$. Training such a network will give us a solution that has the desired sparsity property.

Other than Assumption 1, there is another setting where we can prove a full characterization of limit directions: when there is one data point ($n = 1$) and the network is 2-layer ($L = 2$). This "extremely overparametrized" case is motivated by an experimental paper (Zhang et al., 2019) which studies generalization performance of different architectures when there is only one training data point.

**Theorem 3.** *Suppose we have a 2-layer linear tensor network* (6) *and a single data point* $(\boldsymbol{x}, y)$. *Consider the compact SVD* $\mathsf{M}(\boldsymbol{x}) = \boldsymbol{U}_1 \operatorname{diag}(\boldsymbol{s}) \boldsymbol{U}_2^T$, *where* $\boldsymbol{U}_1 \in \mathbb{R}^{k_1 \times m}$, $\boldsymbol{U}_2 \in \mathbb{R}^{k_2 \times m}$, *and* $\boldsymbol{s} \in \mathbb{R}^m$ *for* $m \leq \min\{k_1, k_2\}$. *Let* $\boldsymbol{\rho}^\infty \in \mathbb{R}^m$ *be a solution of the following optimization problem*

$$\operatorname{minimize}_{\boldsymbol{\rho} \in \mathbb{R}^m} \quad \|\boldsymbol{\rho}\|_1 \quad \text{subject to} \quad y \boldsymbol{s}^T \boldsymbol{\rho} \geq 1.$$

*Assume that there exists* $\lambda > 0$ *such that the initial directions* $\bar{v}_1, \bar{v}_2$ *of the network parameters satisfy* $[\boldsymbol{U}_1^T \bar{v}_1]_j^2 - [\boldsymbol{U}_2^T \bar{v}_2]_j^2 \geq \lambda$ *for all* $j \in [m]$. *Then,* $\boldsymbol{v}_1$ *and* $\boldsymbol{v}_2$ *converge in direction to* $\boldsymbol{U}_1 \boldsymbol{\eta}_1^\infty$ *and* $\boldsymbol{U}_2 \boldsymbol{\eta}_2^\infty$, *where* $|\boldsymbol{\eta}_1^\infty| = |\boldsymbol{\eta}_2^\infty| = |\boldsymbol{\rho}^\infty|^{\odot 1/2}$, *and* $\operatorname{sign}(\boldsymbol{\eta}_1^\infty) = \operatorname{sign}(y) \odot \operatorname{sign}(\boldsymbol{\eta}_2^\infty)$.

The proof of Theorem 3 can be found in Appendix E. Since $\boldsymbol{\rho}^\infty$ is the minimum $\ell_1$ norm solution in the singular value space, the parameters $\boldsymbol{v}_1$ and $\boldsymbol{v}_2$ converge in direction to the top singular vectors. We would like to emphasize that this theorem can be applied to *any* network architecture that can be represented as a linear tensor network. Recall that the previous result (Gunasekar et al., 2018b) only considers full-length filters ($k_1 = d$), hence providing limited insights on networks with small filters, e.g., $k_1 = 2$. In light of this, we present a corollary in Appendix A.3 showing that linear coefficients of convolutional networks converge in direction to a "filtered" version of $\boldsymbol{x}$.

## 5 IMPLICIT BIAS OF GRADIENT FLOW IN UNDERDETERMINED REGRESSION

In Section 4, the limit directions of parameters we characterized do not depend on initialization. This is due to the fact that the parameters diverge to infinity in separable classification problems, so that the initialization becomes unimportant in the limit. This is not the case in regression setting, because parameters do not diverge to infinity. As we show in this section, the limit points are closely tied to initialization, and our analyses characterize the dependency between them.

### 5.1 LIMIT POINT CHARACTERIZATION FOR ORTHOGONALLY DECOMPOSABLE NETWORKS

For the orthogonally decomposable networks satisfying Assumption 1 with real $\boldsymbol{S}$ and $\boldsymbol{U}_l$'s, we consider how limit points of gradient flow change according to initialization. We consider a specific

initialization scheme that, in the special case of diagonal networks, corresponds to setting $\boldsymbol{w}_l(0) = \alpha \bar{\boldsymbol{w}}$ for $l \in [L-1]$ and $\boldsymbol{w}_L(0) = \boldsymbol{0}$. We use the following lemma on a relevant system of ODEs:

**Lemma 4.** *Consider the system of ODEs, where $p, q : \mathbb{R} \to \mathbb{R}$:*

$$\dot{p} = p^{L-2}q, \quad \dot{q} = p^{L-1}, \quad p(0) = 1, \quad q(0) = 0.$$

*Then, the solutions $p_L(t)$ and $q_L(t)$ are continuous on their maximal interval of existence of the form $(-c, c) \subset \mathbb{R}$ for some $c \in (0, \infty]$. Define $h_L(t) = p_L(t)^{L-1}q_L(t)$; then, $h_L(t)$ is odd and strictly increasing, satisfying $\lim_{t \uparrow c} h_L(t) = \infty$ and $\lim_{t \downarrow -c} h_L(t) = -\infty$.*

Using the function $h_L(t)$ from Lemma 4, we can obtain the following theorem that characterizes the limit points as the minimizer of a norm-like function $Q_{L,\alpha,\bar{\boldsymbol{\eta}}}$ among the global minima.

**Theorem 5.** *Suppose a linear tensor network satisfies Assumption 1. Assume further that the matrices $\boldsymbol{U}_1, \ldots, \boldsymbol{U}_L$ and $\boldsymbol{S}$ from Assumption 1 are all real matrices. For some $\lambda > 0$, choose any vector $\bar{\boldsymbol{\eta}} \in \mathbb{R}^m$ satisfying $[\bar{\boldsymbol{\eta}}]_j^2 \geq \lambda$ for all $j \in [m]$, and choose initial directions $\bar{\boldsymbol{v}}_l = \boldsymbol{U}_l \bar{\boldsymbol{\eta}}$ for $l \in [L-1]$ and $\bar{\boldsymbol{v}}_L = \boldsymbol{0}$. Then, the linear coefficients $\boldsymbol{\beta}(\boldsymbol{\Theta}(t))$ converge to $\boldsymbol{S}^T \boldsymbol{\rho}^\infty$, where $\boldsymbol{\rho}^\infty$ is the solution of*

$$\text{minimize}_{\boldsymbol{\rho} \in \mathbb{R}^m} \quad Q_{L,\alpha,\bar{\boldsymbol{\eta}}}(\boldsymbol{\rho}) := \alpha^2 \sum_{j=1}^m [\bar{\boldsymbol{\eta}}]_j^2 H_L\left(\frac{[\boldsymbol{\rho}]_j}{\alpha^L |[\bar{\boldsymbol{\eta}}]_j|^L}\right) \quad \text{subject to} \quad \boldsymbol{X}\boldsymbol{S}^T\boldsymbol{\rho} = \boldsymbol{y},$$

*where $Q_{L,\alpha,\bar{\boldsymbol{\eta}}} : \mathbb{R}^m \to \mathbb{R}$ is a norm-like function defined using $H_L(t) := \int_0^t h_L^{-1}(\tau)d\tau$. If $\boldsymbol{S}$ is invertible, then $\boldsymbol{\beta}(\boldsymbol{\Theta}(t))$ converges to the solution $\boldsymbol{z}^\infty$ of*

$$\text{minimize}_{\boldsymbol{z} \in \mathbb{R}^d} \quad Q_{L,\alpha,\bar{\boldsymbol{\eta}}}(\boldsymbol{S}^{-T}\boldsymbol{z}) \quad \text{subject to} \quad \boldsymbol{X}\boldsymbol{z} = \boldsymbol{y}.$$

The proofs of Lemma 4 and Theorem 5 are deferred to Appendix F.

**Remark 4** (Interpolation between $\ell_1$ and $\ell_2$). It can be checked that $H_L(t)$ grows like the absolute value function if $t$ is large, and grows like a quadratic function if $t$ is close to zero. This means that

$$\lim_{\alpha \to 0} Q_{L,\alpha,\bar{\boldsymbol{\eta}}}(\boldsymbol{\rho}) \propto \sum_{j=1}^m \frac{|[\boldsymbol{\rho}]_j|}{|[\bar{\boldsymbol{\eta}}]_j|^{L-2}}, \quad \lim_{\alpha \to \infty} Q_{L,\alpha,\bar{\boldsymbol{\eta}}}(\boldsymbol{\rho}) \propto \sum_{j=1}^m \frac{[\boldsymbol{\rho}]_j^2}{[\bar{\boldsymbol{\eta}}]_j^{2L-2}},$$

so $Q_{L,\alpha,\bar{\boldsymbol{\eta}}}$ interpolates between the weighted $\ell_1$ and weighted $\ell_2$ norms of $\boldsymbol{\rho}$. Also, the weights in the norm are *dependent* on the initialization direction $\bar{\boldsymbol{\eta}}$ unless $L = 2$ and $\alpha \to 0$. In general, $Q_{L,\alpha,\bar{\boldsymbol{\eta}}}$ interpolates the standard $\ell_1$ and $\ell_2$ norms only if $|[\bar{\boldsymbol{\eta}}]_j|$ is the same for all $j \in [m]$. This result is similar to the observations made in Woodworth et al. (2020) which considers a diagonal network with a "differential" structure $f(\boldsymbol{x}; \boldsymbol{w}_+, \boldsymbol{w}_-) = \boldsymbol{x}^T(\boldsymbol{w}_+^{\odot L} - \boldsymbol{w}_-^{\odot L})$. In contrast, our results apply to a more general class of networks, without the need to have the differential structure. In Appendix A.4, we state corollaries of Theorem 5 for linear diagonal networks and linear full-length convolutional networks with even data points. There, we also show that deep matrix sensing with commutative sensor matrices (Arora et al., 2019b) is a special case of our setting.

Next, we present the regression counterpart of Theorem 3, for 2-layer linear tensor networks with a single data point. For this extremely overparametrized setup, we can fully characterize the limit points as functions of initialization $\boldsymbol{v}_1(0) = \alpha \bar{\boldsymbol{v}}_1$ and $\boldsymbol{v}_2(0) = \alpha \bar{\boldsymbol{v}}_2$, for *any* linear tensor networks including linear convolutional networks with filter size smaller than input dimension.

**Theorem 6.** *Suppose we have a 2-layer linear tensor network (6) and a single data point $(\boldsymbol{x}, y)$. Consider the compact SVD $\boldsymbol{M}(\boldsymbol{x}) = \boldsymbol{U}_1 \text{diag}(\boldsymbol{s})\boldsymbol{U}_2^T$, where $\boldsymbol{U}_1 \in \mathbb{R}^{k_1 \times m}$, $\boldsymbol{U}_2 \in \mathbb{R}^{k_2 \times m}$, and $\boldsymbol{s} \in \mathbb{R}^m$ for $m \leq \min\{k_1, k_2\}$. Assume that there exists $\lambda > 0$ such that the initial directions $\bar{\boldsymbol{v}}_1, \bar{\boldsymbol{v}}_2$ of the network parameters satisfy $[\boldsymbol{U}_1^T \bar{\boldsymbol{v}}_1]_j^2 - [\boldsymbol{U}_2^T \bar{\boldsymbol{v}}_2]_j^2 \geq \lambda$ for all $j \in [m]$. Then, gradient flow converges to a global minimizer of the loss $\mathcal{L}$, and $\boldsymbol{v}_1(t)$ and $\boldsymbol{v}_2(t)$ converge to the limit points:*

$$\boldsymbol{v}_1^\infty = \alpha \boldsymbol{U}_1\left(\boldsymbol{U}_1^T \bar{\boldsymbol{v}}_1 \odot \cosh\left(g^{-1}\left(\frac{y}{\alpha^2}\right)\boldsymbol{s}\right) + \boldsymbol{U}_2^T \bar{\boldsymbol{v}}_2 \odot \sinh\left(g^{-1}\left(\frac{y}{\alpha^2}\right)\boldsymbol{s}\right)\right) + \alpha(\boldsymbol{I}_{k_1} - \boldsymbol{U}_1\boldsymbol{U}_1^T)\bar{\boldsymbol{v}}_1,$$

$$\boldsymbol{v}_2^\infty = \alpha \boldsymbol{U}_2\left(\boldsymbol{U}_1^T \bar{\boldsymbol{v}}_1 \odot \sinh\left(g^{-1}\left(\frac{y}{\alpha^2}\right)\boldsymbol{s}\right) + \boldsymbol{U}_2^T \bar{\boldsymbol{v}}_2 \odot \cosh\left(g^{-1}\left(\frac{y}{\alpha^2}\right)\boldsymbol{s}\right)\right) + \alpha(\boldsymbol{I}_{k_2} - \boldsymbol{U}_2\boldsymbol{U}_2^T)\bar{\boldsymbol{v}}_2,$$

*where $g^{-1}$ is the inverse of the following strictly increasing function*

$$g(\nu) = \sum_{j=1}^m [\boldsymbol{s}]_j\left(\frac{[\boldsymbol{U}_1^T \bar{\boldsymbol{v}}_1]_j^2 + [\boldsymbol{U}_2^T \bar{\boldsymbol{v}}_2]_j^2}{2}\sinh(2[\boldsymbol{s}]_j\nu) + [\boldsymbol{U}_1^T \bar{\boldsymbol{v}}_1]_j[\boldsymbol{U}_2^T \bar{\boldsymbol{v}}_2]_j \cosh(2[\boldsymbol{s}]_j\nu)\right).$$

The proof can be found in Appendix G. We can observe that as $\alpha \to 0$, we have $g^{-1}\left(\frac{y}{\alpha^2}\right) \to \infty$, which results in exponentially faster growth of the $\sinh(\cdot)$ and $\cosh(\cdot)$ for the top singular values. As a result, the top singular vectors dominate the limit points $\boldsymbol{v}_1^\infty$ and $\boldsymbol{v}_2^\infty$ as $\alpha \to 0$, and they do not depend on the initial directions $\bar{\boldsymbol{v}}_1, \bar{\boldsymbol{v}}_2$. Experiment results in Section 6 support this observation.

## 5.2 IMPLICIT BIAS IN FULLY-CONNECTED NETWORKS: THE $\alpha \to 0$ LIMIT

We state our last theoretical element of this paper, which proves that the linear coefficients $\boldsymbol{\beta}_{\text{fc}}(\boldsymbol{\Theta}_{\text{fc}})$ of deep linear fully-connected networks converge to the minimum $\ell_2$ norm solution as $\alpha \to 0$. We assume for simplicity that $d_1 = d_2 = \cdots = d_L = d$ in this section, but we can extend it for $d_l \geq d$ without too much difficulty. Recall $f_{\text{fc}}(\boldsymbol{x}; \boldsymbol{\Theta}_{\text{fc}}) = \boldsymbol{x}^T \boldsymbol{W}_1 \cdots \boldsymbol{W}_{L-1} \boldsymbol{w}_L$. We minimize the training loss $\mathcal{L}$ with initialization $\boldsymbol{W}_l(0) = \alpha \bar{\boldsymbol{W}}_l$ for $l \in [L-1]$ and $\boldsymbol{w}_L(0) = \alpha \bar{\boldsymbol{w}}_L$.

**Theorem 7.** *Assume that initial directions $\bar{\boldsymbol{W}}_1, \ldots, \bar{\boldsymbol{W}}_{L-1}, \bar{\boldsymbol{w}}_L$ satisfy (1) $\bar{\boldsymbol{W}}_l^T \bar{\boldsymbol{W}}_l \succeq \bar{\boldsymbol{W}}_{l+1} \bar{\boldsymbol{W}}_{l+1}^T$ for $l \in [L-2]$, and (2) there exists $\lambda > 0$ such that $\bar{\boldsymbol{W}}_{L-1}^T \bar{\boldsymbol{W}}_{L-1} - \bar{\boldsymbol{w}}_L \bar{\boldsymbol{w}}_L^T \succeq \lambda \boldsymbol{I}_d$. Then, the gradient flow converges to a global minimum, and $\lim_{\alpha \to 0} \lim_{t \to \infty} \boldsymbol{\beta}_{\text{fc}}(\boldsymbol{\Theta}_{\text{fc}}(t)) = \boldsymbol{X}^T(\boldsymbol{X}\boldsymbol{X}^T)^{-1}\boldsymbol{y}$.*

The proof is presented in Appendix H. Theorem 7 shows that in the limit $\alpha \to 0$, linear fully-connected networks have bias towards the minimum $\ell_2$ norm solution, regardless of the depth. This is consistent with the results shown for classification. We also note that the convergence to a global minimum holds for any $\alpha > 0$, and our sufficient conditions ($\bar{\boldsymbol{W}}_l^T \bar{\boldsymbol{W}}_l \succeq \bar{\boldsymbol{W}}_{l+1} \bar{\boldsymbol{W}}_{l+1}^T$ and $\bar{\boldsymbol{W}}_{L-1}^T \bar{\boldsymbol{W}}_{L-1} - \bar{\boldsymbol{w}}_L \bar{\boldsymbol{w}}_L^T \succeq \lambda \boldsymbol{I}_d$) for global convergence is a generalization of the zero-asymmetric initialization scheme ($\bar{\boldsymbol{W}}_1 = \cdots = \bar{\boldsymbol{W}}_{L-1} = \boldsymbol{I}_d$ and $\bar{\boldsymbol{w}}_L = \boldsymbol{0}$) proposed in Wu et al. (2019).

## 6 EXPERIMENTS

**Regression.** To fully visualize the trajectory of linear coefficients, we run simple experiments with 2-layer linear fully-connected/diagonal/convolutional networks with a single 2-dimensional data point $(\boldsymbol{x}, y) = ([1\ \ 2], 1)$. For this dataset, the minimum $\ell_2$ norm solution (corresponding to fully-connected networks) of the regression problem is $[0.2\ \ 0.4]$, whereas the minimum $\ell_1$ norm solution (corresponding to diagonal) is $[0\ \ 0.5]$ and the minimum DFT-domain $\ell_1$ norm solution (corresponding to convolutional) is $[0.33\ \ 0.33]$. We randomly pick four directions $\bar{\boldsymbol{z}}_1, \ldots \bar{\boldsymbol{z}}_4 \in \mathbb{R}^2$, and choose initial directions of the network parameters in a way that their linear coefficients at initialization are exactly $\boldsymbol{\beta}(\boldsymbol{\Theta}(0)) = \alpha^2 \bar{\boldsymbol{z}}_j$. With varying initial scales $\alpha \in \{0.01, 0.5, 1\}$, we run GD with small step size $\eta = 10^{-3}$ for large enough number of iterations $T = 5 \times 10^3$. Figures 1 and 2 plot the trajectories of $\boldsymbol{\beta}(\boldsymbol{\Theta})$ (appropriately clipped for visual clarity) as well as the predicted limit points (Theorem 6). We observe that even though the networks start at the same linear coefficients $\alpha^2 \bar{\boldsymbol{z}}_j$, they evolve differently due to different architectures. Note that the prediction of limit points is accurate, and the solution found by GD is less dependent on initial directions when $\alpha$ is small.

**Classification.** It is shown in the existing works as well as in Section 4 that the limit directions of linear coefficients are independent of the initialization. Is this also true in practice? To see this, we run a set of toy experiments on classification with two data points $(\boldsymbol{x}_1, y_1) = ([1\ \ 2], +1)$ and $(\boldsymbol{x}_2, y_2) = ([0\ \ {-3}], -1)$. One can check that the max-margin classifiers for this problem are in the same directions to the corresponding min-norm solutions in the regression problem above. We use the same networks as in regression, and the same set of initial directions satisfying $\boldsymbol{\beta}(\boldsymbol{\Theta}(0)) = \alpha^2 \bar{\boldsymbol{z}}_j$. With initial scales $\alpha \in \{0.01, 0.5, 1\}$, we run GD with step size $\eta = 5 \times 10^{-4}$ for $T = 2 \times 10^6$ iterations. All experiments reached $\mathcal{L}(\boldsymbol{\Theta}) \lesssim 10^{-5}$ at the end. The trajectories are plotted in Figure 2 in the Appendix. We find that, in contrast to our theoretical characterization, the actual coefficients are quite dependent on initialization, because we do not train the network all the way to zero loss. This observation is also consistent with a recent analysis (Moroshko et al., 2020) for diagonal networks, and suggests that understanding the behavior of iterates after a finite number of steps is an important future work.

## 7 CONCLUSION

This paper studies the implicit bias of gradient flow on training linear tensor networks. Under a general tensor formulation of linear networks, we provide theorems characterizing how the network architectures and initializations affect the limit directions/points of gradient flow. Our work provides a unified framework that connects multiple existing results on implicit bias of gradient flow as special cases.

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

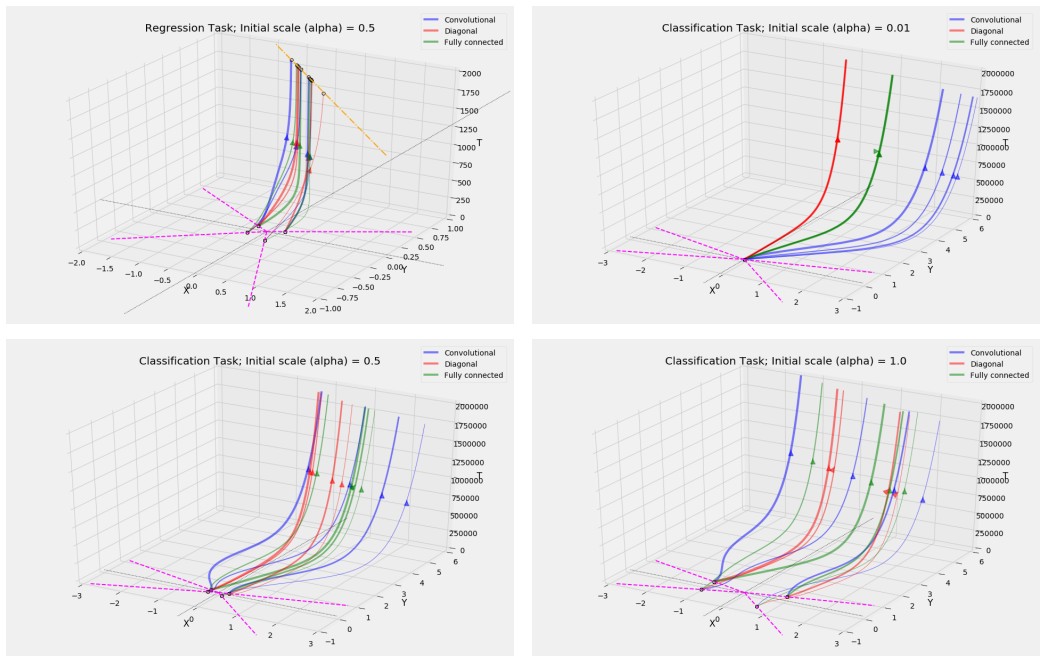

Figure 2: Gradient descent trajectories of linear coefficients of linear fully-connected, diagonal, and convolutional networks on a regression task with initial scale $\alpha = 0.5$ (top left), and networks on a classification task with initial scales $\alpha = 0.01, 0.5, 1$ (rest). Networks are initialized at the same coefficients (circles on purple lines), but follow different trajectories due to different implicit biases of networks induced from their architecture. The top left figure shows that our theoretical predictions on limit points (circles on yellow line, the set of global minima) agree with the solution found by GD. For details of the experimental setup, please refer to Section 6.

## A  COROLLARIES ON SPECIFIC NETWORK ARCHITECTURES

We present corollaries obtained by specializing the theorems in the main text to specific network architectures. We briefly review the linear neural network architectures studied in this section.

**Linear fully-connected networks.**  An $L$-layer linear fully-connected network is defined as

$$f_{\text{fc}}(\boldsymbol{x}; \boldsymbol{\Theta}_{\text{fc}}) = \boldsymbol{x}^T \boldsymbol{W}_1 \cdots \boldsymbol{W}_{L-1} \boldsymbol{w}_L, \qquad (10)$$

where $\boldsymbol{W}_l \in \mathbb{R}^{d_l \times d_{l+1}}$ for $l \in [L-1]$ (we use $d_1 = d$) and $\boldsymbol{w}_L \in \mathbb{R}^{d_L}$.

**Linear diagonal networks.**  An $L$-layer linear diagonal network is written as

$$f_{\text{diag}}(\boldsymbol{x}; \boldsymbol{\Theta}_{\text{diag}}) = (\boldsymbol{x} \odot \boldsymbol{w}_1 \odot \cdots \odot \boldsymbol{w}_{L-1})^T \boldsymbol{w}_L, \qquad (11)$$

where $\boldsymbol{w}_l \in \mathbb{R}^d$ for $l \in [L]$.

**Linear (circular) convolutional networks.**  An $L$-layer linear convolutional network is written as

$$f_{\text{conv}}(\boldsymbol{x}; \boldsymbol{\Theta}_{\text{conv}}) = (\cdots ((\boldsymbol{x} \star \boldsymbol{w}_1) \star \boldsymbol{w}_2) \cdots \star \boldsymbol{w}_{L-1})^T \boldsymbol{w}_L, \qquad (12)$$

where $\boldsymbol{w}_l \in \mathbb{R}^{k_l}$ with $k_l \leq d$ and $k_L = d$, and $\star$ defines the circular convolution: for any $\boldsymbol{a} \in \mathbb{R}^d$ and $\boldsymbol{b} \in \mathbb{R}^k$ ($k \leq d$), we have $\boldsymbol{a} \star \boldsymbol{b} \in \mathbb{R}^d$ defined as $[\boldsymbol{a} \star \boldsymbol{b}]_i = \sum_{j=1}^{k} [\boldsymbol{a}]_{(i+j-1) \bmod d} [\boldsymbol{b}]_j$, for $i \in [d]$. In case of $k_l = d$ for all $l \in [L]$, we refer to this network as *full-length* convolutional networks.

**Deep matrix sensing.**  The deep matrix sensing problem considered in Gunasekar et al. (2017); Arora et al. (2019b) aims to minimize the following problem

$$\underset{\boldsymbol{W}_1,\dots,\boldsymbol{W}_L \in \mathbb{R}^{d \times d}}{\text{minimize}} \quad \mathcal{L}_{\text{ms}}(\boldsymbol{W}_1 \cdots \boldsymbol{W}_L) := \sum_{i=1}^{n} (y_i - \langle \boldsymbol{A}_i, \boldsymbol{W}_1 \cdots \boldsymbol{W}_L \rangle)^2, \qquad (13)$$

where the sensor matrices $\boldsymbol{A}_1, \dots, \boldsymbol{A}_n \in \mathbb{R}^{d \times d}$ are symmetric. Following Gunasekar et al. (2017); Arora et al. (2019b), we consider sensor matrices $\boldsymbol{A}_1, \dots, \boldsymbol{A}_n \in \mathbb{R}^{d \times d}$ that commute. To make the problem underdetermined, we assume that $n \leq d$, and $\boldsymbol{A}_i$'s are linearly independent.

### A.1 COROLLARY OF THEOREM 1

**Corollary 1.** *Consider an L-layer linear fully-connected network* (10). *If the training loss satisfies* $\mathcal{L}(\mathbf{\Theta}_{\mathrm{fc}}(t_0)) < 1$ *for some* $t_0 \geq 0$*, then* $\boldsymbol{\beta}_{\mathrm{fc}}(\mathbf{\Theta}_{\mathrm{fc}}(t))$ *converges in a direction that aligns with the solution of the following optimization problem*

$$\text{minimize}_{\boldsymbol{z} \in \mathbb{R}^d} \quad \|\boldsymbol{z}\|_2^2 \quad \text{subject to} \quad y_i \boldsymbol{x}_i^T \boldsymbol{z} \geq 1, \ \forall i \in [n].$$

Corollary 1 shows that whenever the network separates the data correctly, the linear coefficients $\boldsymbol{\beta}_{\mathrm{fc}}(\mathbf{\Theta}_{\mathrm{fc}})$ convergence in direction to the $\ell_2$ max-margin classifier. Note that this corollary does not require the directional convergence of $\boldsymbol{X}^T \boldsymbol{r}$, which is different from Theorem 1. In fact, this corollary also appears in Ji & Telgarsky (2020), but we provide an alternative proof based on our tensor formulation. The proof of Corollary 1 can be found in Appendix C.

### A.2 COROLLARIES OF THEOREM 2

**Corollary 2.** *Consider an L-layer linear diagonal network* (11). *If there exists* $\lambda > 0$ *such that the initial directions* $\bar{\boldsymbol{w}}_1, \ldots, \bar{\boldsymbol{w}}_L$ *of the network parameters satisfy* $[\bar{\boldsymbol{w}}_l]_j^2 - [\bar{\boldsymbol{w}}_L]_j^2 \geq \lambda$ *for all* $l \in [L-1]$ *and* $j \in [d]$*, then* $\boldsymbol{\beta}_{\mathrm{diag}}(\mathbf{\Theta}_{\mathrm{diag}}(t))$ *converges in a direction that aligns with a stationary point* $\boldsymbol{z}^\infty$ *of*

$$\text{minimize}_{\boldsymbol{z} \in \mathbb{R}^d} \quad \|\boldsymbol{z}\|_{2/L} \quad \text{subject to} \quad y_i \boldsymbol{x}_i^T \boldsymbol{z} \geq 1, \ \forall i \in [n].$$

For full-length convolutional networks, we define $\boldsymbol{F} \in \mathbb{C}^{d \times d}$ to be the matrix of discrete Fourier transform basis $[\boldsymbol{F}]_{j,k} = \frac{1}{\sqrt{d}} \exp(-\frac{\sqrt{-1} \cdot 2\pi (j-1)(k-1)}{d})$. Note that $\boldsymbol{F}^* = \boldsymbol{F}^{-1}$, and both $\boldsymbol{F}$ and $\boldsymbol{F}^*$ are symmetric, but not Hermitian.

**Corollary 3.** *Consider an L-layer linear full-length convolutional network* (12). *If there exists* $\lambda > 0$ *such that the initial directions* $\bar{\boldsymbol{w}}_1, \ldots, \bar{\boldsymbol{w}}_L$ *of the network parameters satisfy* $|[\boldsymbol{F}\bar{\boldsymbol{w}}_l]_j|^2 - |[\boldsymbol{F}\bar{\boldsymbol{w}}_L]_j|^2 \geq \lambda$ *for all* $l \in [L-1]$ *and* $j \in [d]$*, then* $\boldsymbol{\beta}_{\mathrm{conv}}(\mathbf{\Theta}_{\mathrm{conv}}(t))$ *converges in a direction that aligns with a stationary point* $\boldsymbol{z}^\infty$ *of*

$$\text{minimize}_{\boldsymbol{z} \in \mathbb{R}^d} \quad \|\boldsymbol{F}\boldsymbol{z}\|_{2/L} \quad \text{subject to} \quad y_i \boldsymbol{x}_i^T \boldsymbol{z} \geq 1, \ \forall i \in [n].$$

Corollary 2 shows that in the limit, linear diagonal network finds a sparse solution $\boldsymbol{z}$ that is a stationary point of the $\ell_{2/L}$ max-margin classification problem. Corollary 3 has a similar conclusion except that the standard $\ell_{2/L}$ norm is replaced with DFT-domain $\ell_{2/L}$ norm. By specifying mild conditions on initialization (see Remark 1), these corollaries remove the convergence assumptions required in Gunasekar et al. (2018b). The proofs of Corollaries 2 and 3 are in Appendix D.

### A.3 COROLLARY OF THEOREM 3

Recall that Theorem 3 can be applied to any 2-layer networks that can be represented as linear tensor networks. Examples include the convolutional networks that are not full-length (i.e., filter size $k_1 < d$), which are not covered by the previous result (Gunasekar et al., 2018b). Here, we present the characterization of convergence directions of $\boldsymbol{\beta}_{\mathrm{conv}}(\mathbf{\Theta}_{\mathrm{conv}}(t))$ for small-filter cases: $k_1 = 1$ and $k_1 = 2$.

**Corollary 4.** *Consider a 2-layer linear convolutional network* (12) *with* $k_1 = 1$ *and a single data point* $(\boldsymbol{x}, y)$*. If there exists* $\lambda > 0$ *such that the initial directions* $\bar{\boldsymbol{w}}_1$ *and* $\bar{\boldsymbol{w}}_2$ *of the network parameters satisfy* $\|\boldsymbol{x}\|^2 \bar{\boldsymbol{v}}_1^2 - (\boldsymbol{x}^T \bar{\boldsymbol{v}}_2)^2 \geq \|\boldsymbol{x}\|^2 \lambda$*, then* $\boldsymbol{\beta}_{\mathrm{conv}}(\mathbf{\Theta}_{\mathrm{conv}}(t))$ *converges in direction that aligns with* $y\boldsymbol{x}$*.*

*Consider a 2-layer linear convolutional network* (12) *with* $k_1 = 2$ *and a single data point* $(\boldsymbol{x}, y)$*. Let* $\overleftarrow{\boldsymbol{x}} := [[\boldsymbol{x}]_2 \ \cdots \ [\boldsymbol{x}]_d \ [\boldsymbol{x}]_1]$*, and* $\overrightarrow{\boldsymbol{x}} := [[\boldsymbol{x}]_d \ [\boldsymbol{x}]_1 \ \cdots \ [\boldsymbol{x}]_{d-1}]$*. If there exists* $\lambda > 0$ *such that the initial directions* $\bar{\boldsymbol{w}}_1$ *and* $\bar{\boldsymbol{w}}_2$ *of the network parameters satisfy*

$$([\bar{\boldsymbol{v}}_1]_1 + [\bar{\boldsymbol{v}}_1]_2)^2 - \frac{((\boldsymbol{x} + \overleftarrow{\boldsymbol{x}})^T \bar{\boldsymbol{v}}_2)^2}{\|\boldsymbol{x}\|_2^2 + \boldsymbol{x}^T \overleftarrow{\boldsymbol{x}}} \geq \lambda, \quad \text{and} \quad ([\bar{\boldsymbol{v}}_1]_1 - [\bar{\boldsymbol{v}}_1]_2)^2 - \frac{((\boldsymbol{x} - \overleftarrow{\boldsymbol{x}})^T \bar{\boldsymbol{v}}_2)^2}{\|\boldsymbol{x}\|_2^2 - \boldsymbol{x}^T \overleftarrow{\boldsymbol{x}}} \geq \lambda,$$

*then* $\boldsymbol{\beta}_{\mathrm{conv}}(\mathbf{\Theta}_{\mathrm{conv}}(t))$ *converges in a direction that aligns with a filtered version of* $\boldsymbol{x}$*:*

$$\lim_{t \to \infty} \frac{\boldsymbol{\beta}_{\mathrm{conv}}(\mathbf{\Theta}_{\mathrm{conv}}(t))}{\|\boldsymbol{\beta}_{\mathrm{conv}}(\mathbf{\Theta}_{\mathrm{conv}}(t))\|_2} \propto \begin{cases} 2y\boldsymbol{x} + y\overleftarrow{\boldsymbol{x}} + y\overrightarrow{\boldsymbol{x}} & \text{if } \boldsymbol{x}^T \overleftarrow{\boldsymbol{x}} > 0, \\ 2y\boldsymbol{x} - y\overleftarrow{\boldsymbol{x}} - y\overrightarrow{\boldsymbol{x}} & \text{if } \boldsymbol{x}^T \overleftarrow{\boldsymbol{x}} < 0. \end{cases}$$

Corollary 4 shows that if the filter size is $k_1 = 1$, then the limit direction of $\boldsymbol{\beta}_{\text{conv}}(\boldsymbol{\Theta}_{\text{conv}})$ is always the $\ell_2$ max-margin classifier. Note that this is quite different from the case $k_1 = d$ which converges to the DFT-domain $\ell_1$ max-margin classifier. However, for $1 < k_1 < d$, it is difficult to characterize the limit direction as the max-margin classifier of some common norms. Rather, the limit directions of $\boldsymbol{\beta}_{\text{conv}}(\boldsymbol{\Theta}_{\text{conv}})$ correspond to a "filtered" version of the data point, and the weights of the filter depend on the data point $\boldsymbol{x}$. For $k_1 = 2$, the filter is a low-pass filter if the autocorrelation $\boldsymbol{x}^T \overleftarrow{\boldsymbol{x}}$ of $\boldsymbol{x}$ is positive, and high-pass if the autocorrelation is negative. For $k_1 > 2$, the filter weights are more complicated to characterize in terms of $\boldsymbol{x}$, and the filter length increases as $k_1$ increases. We prove Corollary 4 in Appendix E.

## A.4 COROLLARIES OF THEOREM 5

In this subsection, we apply Theorem 5 to linear diagonal networks, linear full-length convolutional networks with even data, and deep matrix sensing. The proofs of the corollaries can be found in Appendix F.

**Corollary 5.** *Consider an L-layer linear diagonal network* (11). *For some $\lambda > 0$, choose any vector $\bar{\boldsymbol{w}} \in \mathbb{R}^d$ satisfying $[\bar{\boldsymbol{w}}]_j^2 \geq \lambda$ for all $j \in [d]$, and choose initial directions $\bar{\boldsymbol{w}}_l = \bar{\boldsymbol{w}}$ for $l \in [L-1]$ and $\bar{\boldsymbol{w}}_L = \boldsymbol{0}$. Then, the linear coefficients $\boldsymbol{\beta}_{\text{diag}}(\boldsymbol{\Theta}_{\text{diag}}(t))$ converge to the solution $\boldsymbol{z}^\infty$ of*

$$\text{minimize}_{\boldsymbol{z} \in \mathbb{R}^d} \quad Q_{L,\alpha,\bar{\boldsymbol{w}}}(\boldsymbol{z}) := \alpha^2 \sum_{j=1}^d [\bar{\boldsymbol{w}}]_j^2 H_L\left(\frac{[\boldsymbol{z}]_j}{\alpha^L |[\bar{\boldsymbol{w}}]_j|^L}\right) \quad \text{subject to} \quad \boldsymbol{X}\boldsymbol{z} = \boldsymbol{y}.$$

Recall that the original statement of Assumption 1 allows the matrices $\boldsymbol{S}, \boldsymbol{U}_1, \ldots, \boldsymbol{U}_L$ to be complex, but Theorem 5 poses another assumption that these matrices are real. In applying Theorem 2 to convolutional networks to get Corollary 3, we used the fact that the data tensor $\boldsymbol{\mathsf{M}}_{\text{conv}}(\boldsymbol{x})$ of a linear full-length convolutional network satisfies Assumption 1 with $\boldsymbol{S} = d^{\frac{L-1}{2}}\boldsymbol{F}$ and $\boldsymbol{U}_1 = \cdots = \boldsymbol{U}_L = \boldsymbol{F}^*$, where $\boldsymbol{F} \in \mathbb{C}^{d \times d}$ is the matrix of discrete Fourier transform basis $[\boldsymbol{F}]_{j,k} = \frac{1}{\sqrt{d}}\exp(-\frac{\sqrt{-1}\cdot 2\pi(j-1)(k-1)}{d})$ and $\boldsymbol{F}^*$ is the complex conjugate of $\boldsymbol{F}$. Note that these are complex matrices, so one cannot directly apply Theorem 5 to convolutional networks. However, it turns out that if the data and initialization are even, we can derive a corollary for convolutional networks.

We say that a vector is *even* when it satisfies the even symmetry, as in even functions. More concretely, a vector $\boldsymbol{x} \in \mathbb{R}^d$ is even if $[\boldsymbol{x}]_{j+2} = [\boldsymbol{x}]_{d-j}$ for $j = 0, \ldots, \lfloor\frac{d-3}{2}\rfloor$; i.e., the vector has the even symmetry around its "origin" $[\boldsymbol{x}]_1$. From the definition of the matrix $\boldsymbol{F} \in \mathbb{C}^{d \times d}$, it is straightforward to check that if $\boldsymbol{x}$ is real and even, then its DFT $\boldsymbol{F}\boldsymbol{x}$ is also real and even (see Appendix F.4 for details).

**Corollary 6.** *Consider an L-layer linear full-length convolutional network* (12). *Assume that the data points $\{\boldsymbol{x}_i\}_{i=1}^n$ are all even. For some $\lambda > 0$, choose any even vector $\bar{\boldsymbol{w}}$ satisfying $[\boldsymbol{F}\bar{\boldsymbol{w}}]_j^2 \geq \lambda$ for all $j \in [d]$, and choose initial directions $\bar{\boldsymbol{w}}_l = \bar{\boldsymbol{w}}$ for $l \in [L-1]$ and $\bar{\boldsymbol{w}}_L = \boldsymbol{0}$. Then, the linear coefficients $\boldsymbol{\beta}_{\text{conv}}(\boldsymbol{\Theta}_{\text{conv}}(t))$ converge to the solution $\boldsymbol{z}^\infty$ of*

$$\underset{\boldsymbol{z} \in \mathbb{R}^d,\, \text{even}}{\text{minimize}} \quad Q_{L,\alpha,\boldsymbol{F}\bar{\boldsymbol{w}}}(\boldsymbol{F}\boldsymbol{z}) := \alpha^2 \sum_{j=1}^d [\boldsymbol{F}\bar{\boldsymbol{w}}]_j^2 H_L\left(\frac{[\boldsymbol{F}\boldsymbol{z}]_j}{\alpha^L |[\boldsymbol{F}\bar{\boldsymbol{w}}]_j|^L}\right) \quad \text{subject to} \quad \boldsymbol{X}\boldsymbol{z} = \boldsymbol{y}.$$

Corollaries 5 and 6 show that the interpolation between minimum weighted $\ell_1$ and weighted $\ell_2$ solutions occurs for diagonal networks, and also for convolutional networks (in DFT domain, with the restriction of even symmetry). The conclusion of Corollary 5 is similar to the results in Woodworth et al. (2020), but the network architecture (11) considered in our corollary is a slightly different from the "differential" network $f(\boldsymbol{x}; \boldsymbol{w}_+, \boldsymbol{w}_-) = \boldsymbol{x}^T(\boldsymbol{w}_+^{\odot L} - \boldsymbol{w}_-^{\odot L})$ in Woodworth et al. (2020).

As mentioned in the main text, we can actually show that the matrix sensing result in Arora et al. (2019b) is a special case of our Theorem 5. Given any symmetric matrix $\boldsymbol{M} \in \mathbb{R}^{d \times d}$, let $\text{eig}(\boldsymbol{M}) \in \mathbb{R}^d$ be the $d$-dimensional vector of eigenvalues of $\boldsymbol{M}$.

**Corollary 7.** *Consider the depth-L deep matrix sensing problem* (13). *Let $\boldsymbol{A}_i$'s be symmetric, and assume $\boldsymbol{A}_1, \ldots, \boldsymbol{A}_n$ commute. For $\alpha > 0$, choose initialization $\boldsymbol{W}_l(0) = \alpha\boldsymbol{I}_d$ for $l \in [L-1]$ and $\boldsymbol{W}_L(0) = \boldsymbol{0}$. Then, the product $\boldsymbol{W}_L(t) \cdots \boldsymbol{W}_1(t)$ converge to the solution $\boldsymbol{M}^\infty$ of*

$$\underset{\boldsymbol{M} \in \mathbb{R}^{d \times d},\, \text{symmetric}}{\text{minimize}} \quad Q_{L,\alpha}(\text{eig}(\boldsymbol{M})) := \alpha^2 \sum_{j=1}^d H_L\left(\frac{[\text{eig}(\boldsymbol{M})]_j}{\alpha^L}\right) \quad \text{subject to} \quad \mathcal{L}_{\text{ms}}(\boldsymbol{M}) = 0.$$

Under an additional assumption that $\boldsymbol{A}_i$'s are positive semidefinite, Theorem 2 in Arora et al. (2019b) studies the initialization $\boldsymbol{W}_l(0) = \alpha \boldsymbol{I}_d$ for all $l \in [L]$, and shows that the limit point of $\boldsymbol{W}_L \dots \boldsymbol{W}_1$ converges to the minimum nuclear norm solution as $\alpha \to 0$. We remove the assumption of positive definiteness of $\boldsymbol{A}_i$'s and let $\boldsymbol{W}_L(0) = \boldsymbol{0}$, to show a complete characterization of the solution found by gradient flow, which interpolates between the minimum nuclear norm (i.e., Schatten 1-norm) solution (when $\alpha \to 0$) and the minimum Frobenius norm (i.e., Schatten 2-norm) solution (when $\alpha \to \infty$).

## B    TENSOR REPRESENTATION OF FULLY-CONNECTED NETWORKS

In Section 3, we only defined the data tensor $\mathsf{M}_{\mathrm{fc}}(\boldsymbol{x})$ of fully-connected networks for $L = 2$. Here, we describe an iterative procedure constructing the data tensor for deep fully-connected networks.

We start with $\mathsf{T}_1(\boldsymbol{x}) := \boldsymbol{x} \in \mathbb{R}^{d_1}$. Next, define a block diagonal matrix $\mathsf{T}_2(\boldsymbol{x}) \in \mathbb{R}^{d_1 d_2 \times d_2}$ where the "diagonals" $[\mathsf{T}_2(\boldsymbol{x})]_{d_1(j-1)+1:d_1 j, j} = \mathsf{T}_1(\boldsymbol{x})$ for $j \in [d_2]$, while all the other entries are filled with 0. We continue this "block diagonal" procedure, as the following. Having defined $\mathsf{T}_{l-1}(\boldsymbol{x}) \in \mathbb{R}^{d_1 d_2 \times \cdots \times d_{l-2} d_{l-1} \times d_{l-1}}$,

1. Define $\mathsf{T}_l(\boldsymbol{x}) \in \mathbb{R}^{d_1 d_2 \times \cdots \times d_{l-1} d_l \times d_l}$.
2. Set $[\mathsf{T}_l(\boldsymbol{x})]_{\cdot,\dots,\cdot,d_{l-1}(j-1)+1:d_{l-1}j, j} = \mathsf{T}_{l-1}(\boldsymbol{x}), \forall j \in [d_l]$.
3. Set all the remaining entries of $\mathsf{T}_l(\boldsymbol{x})$ to zero.

We repeat this process for $l = 2, \dots, L$, and set $\mathsf{M}_{\mathrm{fc}}(\boldsymbol{x}) := \mathsf{T}_L(\boldsymbol{x})$. By defining the parameters of the tensor formulation $\boldsymbol{v}_l = \mathrm{vec}(\boldsymbol{W}_l)$ for $l \in [L-1]$ and $\boldsymbol{v}_L = \boldsymbol{w}_L$, and using the tensor $\mathsf{M}(\boldsymbol{x}) = \mathsf{M}_{\mathrm{fc}}(\boldsymbol{x})$, we can check the equivalence of (2) and (5).

## C    PROOFS OF THEOREM 1 AND COROLLARY 1

### C.1    PROOF OF THEOREM 1

The proof of Theorem 1 is outlined as follows. First, using the directional convergence and alignment results in Ji & Telgarsky (2020), we prove that each of our network parameters $\boldsymbol{v}_l$ converges in direction, and it aligns with its corresponding negative gradient $-\nabla_{\boldsymbol{v}_l}\mathcal{L}$. Then, we prove that the directions of $\boldsymbol{v}_l$'s are actually singular vectors of $\mathsf{M}(-\boldsymbol{u}^\infty)$, where $\boldsymbol{u}^\infty := \lim_{t \to \infty} \frac{\boldsymbol{X}^T \boldsymbol{r}(t)}{\|\boldsymbol{X}^T \boldsymbol{r}(t)\|_2}$.

Since a linear tensor network is an $L$-homogeneous polynomial of $\boldsymbol{v}_1, \dots, \boldsymbol{v}_L$, it satisfies the assumptions required for Theorems 3.1 and 4.1 in Ji & Telgarsky (2020). These theorems imply that if the gradient flow satisfies $\mathcal{L}(\boldsymbol{\Theta}(t_0)) < 1$ for some $t_0 \geq 0$, then $\boldsymbol{\Theta}(t)$ converges in direction, and the direction aligns with $-\nabla_{\boldsymbol{\Theta}}\mathcal{L}(\boldsymbol{\Theta}(t))$; that is,

$$\lim_{t \to \infty} \|\boldsymbol{\Theta}(t)\|_2 = \infty, \quad \lim_{t \to \infty} \frac{\boldsymbol{\Theta}(t)}{\|\boldsymbol{\Theta}(t)\|_2} = \boldsymbol{\Theta}^\infty, \quad \lim_{t \to \infty} \frac{\boldsymbol{\Theta}(t)^T \nabla_{\boldsymbol{\Theta}}\mathcal{L}(\boldsymbol{\Theta}(t))}{\|\boldsymbol{\Theta}(t)\|_2 \|\nabla_{\boldsymbol{\Theta}}\mathcal{L}(\boldsymbol{\Theta}(t))\|_2} = -1. \quad (14)$$

For linear tensor networks (6), the parameter $\boldsymbol{\Theta}$ is the concatenation of all parameter vectors $\boldsymbol{v}_1, \dots, \boldsymbol{v}_L$, so (14) holds for $\boldsymbol{\Theta} = \begin{bmatrix} \boldsymbol{v}_1^T & \dots & \boldsymbol{v}_L^T \end{bmatrix}^T$.

Now, recall that by the definition of the linear tensor network, we have the following gradient flow

$$\dot{\boldsymbol{v}}_l = \mathsf{M}(-\boldsymbol{X}^T \boldsymbol{r}) \circ (\boldsymbol{v}_1, \dots, \boldsymbol{v}_{l-1}, \boldsymbol{I}_{k_l}, \boldsymbol{v}_{l+1}, \dots, \boldsymbol{v}_L).$$

Note that we can apply this to calculate the rate of growth of $\|\boldsymbol{v}_l\|_2^2$:

$$\begin{aligned}
\frac{d}{dt}\|\boldsymbol{v}_l\|_2^2 &= 2\boldsymbol{v}_l^T \dot{\boldsymbol{v}}_l = 2\boldsymbol{v}_l^T \mathsf{M}(-\boldsymbol{X}^T \boldsymbol{r}) \circ (\boldsymbol{v}_1, \dots, \boldsymbol{v}_{l-1}, \boldsymbol{I}_{k_l}, \boldsymbol{v}_{l+1}, \dots, \boldsymbol{v}_L) \\
&= 2\mathsf{M}(-\boldsymbol{X}^T \boldsymbol{r}) \circ (\boldsymbol{v}_1, \dots, \boldsymbol{v}_{l-1}, \boldsymbol{v}_l, \boldsymbol{v}_{l+1}, \dots, \boldsymbol{v}_L) \\
&= \frac{d}{dt}\|\boldsymbol{v}_{l'}\|_2^2 \quad \text{for any } l' \in [L],
\end{aligned}$$

so the rate at which $\|\boldsymbol{v}_l\|_2^2$ grows over time is the same for all layers $l \in [L]$. By the definition of $\boldsymbol{\Theta}$ and (14), we have

$$\|\boldsymbol{\Theta}\|_2^2 = \sum_{l=1}^{L} \|\boldsymbol{v}_l\|_2^2 \to \infty,$$

which then implies

$$\lim_{t\to\infty} \|\boldsymbol{v}_l(t)\|_2 \to \infty, \quad \lim_{t\to\infty} \frac{\|\boldsymbol{\Theta}(t)\|_2}{\|\boldsymbol{v}_l(t)\|_2} = \sqrt{\frac{\|\boldsymbol{\Theta}(t)\|_2^2}{\|\boldsymbol{v}_l(t)\|_2^2}} = \sqrt{L},$$

for all $l \in [L]$. Now, let $\mathcal{I}_l$ be the set of indices that correspond to the components of $\boldsymbol{v}_l$ in $\boldsymbol{\Theta}$. It follows from (14) that

$$\lim_{t\to\infty} \frac{\boldsymbol{v}_l(t)}{\|\boldsymbol{v}_l(t)\|_2} = \lim_{t\to\infty} \frac{\boldsymbol{v}_l(t)}{\|\boldsymbol{\Theta}(t)\|_2} \frac{\|\boldsymbol{\Theta}(t)\|_2}{\|\boldsymbol{v}_l(t)\|_2} = \lim_{t\to\infty} \frac{[\boldsymbol{\Theta}(t)]_{\mathcal{I}_l}}{\|\boldsymbol{\Theta}(t)\|_2} \frac{\|\boldsymbol{\Theta}(t)\|_2}{\|\boldsymbol{v}_l(t)\|_2} = \sqrt{L}[\boldsymbol{\Theta}^\infty]_{\mathcal{I}_l},$$

thus showing the directional convergence of $\boldsymbol{v}_l$'s.

Next, it follows from directional convergence of $\boldsymbol{\Theta}$ and its alignment with $-\nabla_{\boldsymbol{\Theta}}\mathcal{L}(\boldsymbol{\Theta})$ (14) that $\nabla_{\boldsymbol{\Theta}}\mathcal{L}(\boldsymbol{\Theta})$ also converges in direction, in the opposite direction of $\boldsymbol{\Theta}$. By comparing the components in $\mathcal{I}_l$'s, we get that $\nabla_{\boldsymbol{v}_l}\mathcal{L}(\boldsymbol{\Theta})$ converges in the opposite direction of $\boldsymbol{v}_l$.

For any $l \in [L]$, now let $\boldsymbol{v}_l^\infty := \lim_{t\to\infty} \frac{\boldsymbol{v}_l(t)}{\|\boldsymbol{v}_l(t)\|_2}$. Also recall the assumption that $\boldsymbol{X}^T \boldsymbol{r}(t)$ converges in direction; let the unit vector $\boldsymbol{u}^\infty := \lim_{t\to\infty} \frac{\boldsymbol{X}^T \boldsymbol{r}(t)}{\|\boldsymbol{X}^T \boldsymbol{r}(t)\|_2}$ be the limit direction. By the gradient flow dynamics of $\boldsymbol{v}_l$, we have

$$\boldsymbol{v}_l^\infty \propto -\nabla_{\boldsymbol{v}_l}\mathcal{L}(\boldsymbol{\Theta}^\infty) = \mathbf{M}(-\boldsymbol{u}^\infty) \circ (\boldsymbol{v}_1^\infty, \dots, \boldsymbol{v}_{l-1}^\infty, \boldsymbol{I}_{k_l}, \boldsymbol{v}_{l+1}^\infty, \dots, \boldsymbol{v}_L^\infty),$$

for all $l \in [L]$. Note that this equation has the same form as (8), the definition of singular vectors in tensors. So this proves that $(\boldsymbol{v}_1^\infty, \dots, \boldsymbol{v}_L^\infty)$ are singular vectors of $\mathbf{M}(-\boldsymbol{u}^\infty)$.

## C.2 Proof of Corollary 1

The proof proceeds as follows. First, we will show using the structure of the data tensor $\mathbf{M}_{\text{fc}}$ that the limit direction of linear coefficients $\boldsymbol{\beta}_{\text{fc}}(\boldsymbol{\Theta}_{\text{fc}}^\infty)$ is proportional to $c\boldsymbol{u}^\infty$, where $c$ is a nonzero scalar and $\boldsymbol{u}^\infty$ is the limit direction of $\boldsymbol{X}^T \boldsymbol{r}$. Then, through a closer look at $\boldsymbol{u}^\infty$ and $c$, we will prove that $\boldsymbol{\beta}_{\text{fc}}(\boldsymbol{\Theta}_{\text{fc}}^\infty)$ is in fact a conic combination of the support vectors (i.e., the data points with the minimum margins). Finally, we will compare $\boldsymbol{\beta}_{\text{fc}}(\boldsymbol{\Theta}_{\text{fc}}^\infty)$ with the KKT conditions of the $\ell_2$ max-margin classification problem and conclude that $\boldsymbol{\beta}_{\text{fc}}(\boldsymbol{\Theta}_{\text{fc}}^\infty)$ must be in the same direction as the $\ell_2$ max-margin classifier.

Due to the way how the data tensor $\mathbf{M}_{\text{fc}}$ is constructed for fully-connected networks (Appendix B), we always have

$$-\nabla_{\boldsymbol{v}_1}\mathcal{L}(\boldsymbol{\Theta}_{\text{fc}}) = \mathbf{M}_{\text{fc}}(-\boldsymbol{X}^T \boldsymbol{r}) \circ (\boldsymbol{I}_{k_1}, \boldsymbol{v}_2, \dots, \boldsymbol{v}_L) \in \text{span}\left\{ \begin{bmatrix} \boldsymbol{X}^T \boldsymbol{r} \\ \boldsymbol{0} \\ \vdots \\ \boldsymbol{0} \end{bmatrix}, \begin{bmatrix} \boldsymbol{0} \\ \boldsymbol{X}^T \boldsymbol{r} \\ \vdots \\ \boldsymbol{0} \end{bmatrix}, \dots, \begin{bmatrix} \boldsymbol{0} \\ \boldsymbol{0} \\ \vdots \\ \boldsymbol{X}^T \boldsymbol{r} \end{bmatrix} \right\}.$$

From Theorem 1, we have directional convergence of $\boldsymbol{v}_1$ and its alignment with $-\nabla_{\boldsymbol{v}_1}\mathcal{L}(\boldsymbol{\Theta}_{\text{fc}})$. This means that the limit direction $\boldsymbol{v}_1^\infty$, which is a fixed vector, must be also in the span of vectors written above. This implies that $\boldsymbol{X}^T \boldsymbol{r}$ must also converge to some direction, say $\boldsymbol{u}^\infty := \lim_{t\to\infty} \frac{\boldsymbol{X}^T \boldsymbol{r}(t)}{\|\boldsymbol{X}^T \boldsymbol{r}(t)\|_2}$.

Now recall the definition of $\boldsymbol{v}_1$ in case of the fully-connected network: $\boldsymbol{v}_1 = \text{vec}(\boldsymbol{W}_1)$. So, by reshaping $\boldsymbol{v}_1^\infty$ into its original $d_1 \times d_2$ matrix form $\boldsymbol{W}_1^\infty$, we have

$$\boldsymbol{W}_1^\infty \propto \boldsymbol{u}^\infty \boldsymbol{q}^T,$$

for some $\boldsymbol{q} \in \mathbb{R}^{d_2}$. This implies that the linear coefficients $\boldsymbol{\beta}_{\text{fc}}(\boldsymbol{\Theta}_{\text{fc}})$ of the network converge in direction to

$$\boldsymbol{\beta}_{\text{fc}}(\boldsymbol{\Theta}_{\text{fc}}^\infty) = \boldsymbol{W}_1^\infty \boldsymbol{W}_2^\infty \dots \boldsymbol{W}_{L-1}^\infty \boldsymbol{w}_L^\infty \propto \boldsymbol{u}^\infty \boldsymbol{q}^T \boldsymbol{W}_2^\infty \dots \boldsymbol{W}_{L-1}^\infty \boldsymbol{w}_L^\infty = c\boldsymbol{u}^\infty, \qquad (15)$$

where $c$ is some nonzero real number.

Let us now take a closer look at the vector $\boldsymbol{u}^\infty$, the limit direction of $\boldsymbol{X}^T\boldsymbol{r}$. Recall from Section 2.1 that for any $i \in [n]$,

$$[\boldsymbol{r}]_i = -y_i \exp(-y_i f_{\text{fc}}(\boldsymbol{x}_i; \boldsymbol{\Theta}_{\text{fc}})) = -y_i \exp(-y_i \boldsymbol{x}_i^T \boldsymbol{\beta}_{\text{fc}}(\boldsymbol{\Theta}_{\text{fc}})),$$

in case of classification. Recall that $\|\boldsymbol{\beta}_{\text{fc}}(\boldsymbol{\Theta}_{\text{fc}}(t))\|_2 \to \infty$ while converging to a certain direction $\boldsymbol{\beta}_{\text{fc}}(\boldsymbol{\Theta}_{\text{fc}}^\infty)$. This means that if

$$y_j \boldsymbol{x}_j^T \boldsymbol{\beta}_{\text{fc}}(\boldsymbol{\Theta}_{\text{fc}}^\infty) > y_i \boldsymbol{x}_i^T \boldsymbol{\beta}_{\text{fc}}(\boldsymbol{\Theta}_{\text{fc}}^\infty)$$

for any $i, j \in [n]$, then

$$\lim_{t \to \infty} \frac{\exp(-y_j \boldsymbol{x}_j^T \boldsymbol{\beta}_{\text{fc}}(\boldsymbol{\Theta}_{\text{fc}}(t)))}{\exp(-y_i \boldsymbol{x}_i^T \boldsymbol{\beta}_{\text{fc}}(\boldsymbol{\Theta}_{\text{fc}}(t)))} = 0. \tag{16}$$

Take $i$ to be the index of any support vector, i.e., any $i$ that attains the minimum $y_i x_i^T \boldsymbol{\beta}_{\text{fc}}(\boldsymbol{\Theta}_{\text{fc}}^\infty)$ among all data points. Using such an $i$, the observation (16) implies that $\lim_{t \to \infty}[\boldsymbol{r}(t)]_j = 0$ for any $\boldsymbol{x}_j$ that is not a support vector. Thus, by the argument above, $\boldsymbol{u}^\infty$ can in fact be written as

$$\boldsymbol{u}^\infty = \lim_{t \to \infty} \frac{\sum_{i=1}^n \boldsymbol{x}_i [\boldsymbol{r}(t)]_i}{\|\sum_{i=1}^n \boldsymbol{x}_i [\boldsymbol{r}(t)]_i\|_2} = -\sum_{i=1}^n \nu_i y_i \boldsymbol{x}_i, \tag{17}$$

where $\nu_i \geq 0$ for all $i \in [n]$, and $\nu_j = 0$ for $\boldsymbol{x}_j$'s that are *not* support vectors. Combining (17) and (15),

$$\boldsymbol{\beta}_{\text{fc}}(\boldsymbol{\Theta}_{\text{fc}}^\infty) \propto -c \sum_{i=1}^n \nu_i y_i \boldsymbol{x}_i. \tag{18}$$

Recall that we do not yet know whether $c$, introduced in (15), is positive or negative; we will now show that $c$ has to be negative. From Lyu & Li (2020), we know that $\mathcal{L}(\boldsymbol{\Theta}_{\text{fc}}(t)) \to 0$, which implies that $y_i \boldsymbol{x}_i^T \boldsymbol{\beta}_{\text{fc}}(\boldsymbol{\Theta}_{\text{fc}}^\infty) > 0$ for all $i \in [n]$. However, if $c > 0$, then (18) implies that $\boldsymbol{\beta}_{\text{fc}}(\boldsymbol{\Theta}_{\text{fc}}^\infty)$ is inside a cone $\mathcal{K}$ defined as

$$\mathcal{K} := \left\{ \sum_{i=1}^n \gamma_i y_i \boldsymbol{x}_i \mid \gamma_i \leq 0, \forall i \in [n] \right\}.$$

Note that the polar cone of $\mathcal{K}$, denoted as $\mathcal{K}^\circ$, is

$$\mathcal{K}^\circ := \left\{ \boldsymbol{z} \mid \boldsymbol{\beta}^T \boldsymbol{z} \leq 0, \forall \boldsymbol{\beta} \in \mathcal{K} \right\} = \left\{ \boldsymbol{z} \mid y_i \boldsymbol{x}_i^T \boldsymbol{z} \geq 0, \forall i \in [n] \right\}.$$

It is known that $\mathcal{K} \cap \mathcal{K}^\circ = \{\boldsymbol{0}\}$ for any convex cone $\mathcal{K}$ and its polar cone $\mathcal{K}^\circ$. Therefore, having $c > 0$ implies that $\boldsymbol{\beta}_{\text{fc}}(\boldsymbol{\Theta}_{\text{fc}}^\infty) \in \mathcal{K} \setminus \mathcal{K}^\circ$, which means that there exists some $i \in [n]$ such that $y_i \boldsymbol{x}_i^T \boldsymbol{\beta}_{\text{fc}}(\boldsymbol{\Theta}_{\text{fc}}^\infty) < 0$; this contradicts the fact that the loss goes to zero as $t \to \infty$. Therefore, $c$ in (15) and (18) must be negative:

$$\boldsymbol{\beta}_{\text{fc}}(\boldsymbol{\Theta}_{\text{fc}}^\infty) \propto \sum_{i=1}^n \nu_i y_i \boldsymbol{x}_i, \tag{19}$$

for $\nu_i \geq 0$ for all $i \in [n]$ and $\nu_j = 0$ for all $\boldsymbol{x}_j$'s that are not suport vectors.

Finally, compare (19) with the KKT conditions of the following optimization problem:

$$\underset{\boldsymbol{z}}{\text{minimize}} \quad \|\boldsymbol{z}\|_2^2 \quad \text{subject to} \quad y_i \boldsymbol{x}_i^T \boldsymbol{z} \geq 1, \ \forall i \in [n].$$

The KKT conditions of this problem are

$$\boldsymbol{z} = \sum_{i=1}^n \mu_i y_i \boldsymbol{x}_i, \ \text{ and } \ \mu_i \geq 0, \ \mu_i(1 - y_i \boldsymbol{x}_i^T \boldsymbol{z}) = 0 \text{ for all } i \in [n],$$

where $\mu_1, \ldots, \mu_n$ are the dual variables. Note that this is (up to scaling) satisfied by $\boldsymbol{\beta}_{\text{fc}}(\boldsymbol{\Theta}_{\text{fc}}^\infty)$ (19), if we replace $\mu_i$'s with $\nu_i$'s. This finishes the proof that $\boldsymbol{\beta}_{\text{fc}}(\boldsymbol{\Theta}_{\text{fc}}^\infty)$ is aligned with the $\ell_2$ max-margin classifier.

# D   PROOFS OF THEOREM 2 AND COROLLARIES 2 & 3

## D.1   PROOF OF THEOREM 2

### D.1.1   CONVERGENCE OF LOSS TO ZERO

Since Theorem 2 does not assume the existence of $t_0 \geq 0$ satisfying $\mathcal{L}(\boldsymbol{\Theta}(t_0)) < 1$, we need to first show that given the conditions on initialization, the training loss $\mathcal{L}(\boldsymbol{\Theta}(t))$ converges to zero. Recall from Section 2.1 that

$$\dot{\boldsymbol{v}}_l = -\nabla_{\boldsymbol{v}_l}\mathcal{L}(\boldsymbol{\Theta}) = \mathsf{M}(-\boldsymbol{X}^T\boldsymbol{r}) \circ (\boldsymbol{v}_1, \ldots, \boldsymbol{v}_{l-1}, \boldsymbol{I}_{k_l}, \boldsymbol{v}_{l+1}, \ldots, \boldsymbol{v}_L).$$

Applying the structure (9) in Assumption 1, we get

$$\dot{\boldsymbol{v}}_l = \mathsf{M}(-\boldsymbol{X}^T\boldsymbol{r}) \circ (\boldsymbol{v}_1, \ldots, \boldsymbol{v}_{l-1}, \boldsymbol{I}_{k_l}, \boldsymbol{v}_{l+1}, \ldots, \boldsymbol{v}_L)$$

$$= -\sum_{j=1}^{m}[\boldsymbol{S}\boldsymbol{X}^T\boldsymbol{r}]_j(\boldsymbol{v}_1^T[\boldsymbol{U}_1]_{\cdot,j} \otimes \cdots \otimes \boldsymbol{v}_{l-1}^T[\boldsymbol{U}_{l-1}]_{\cdot,j} \otimes [\boldsymbol{U}_l]_{\cdot,j} \otimes \boldsymbol{v}_{l+1}^T[\boldsymbol{U}_{l+1}]_{\cdot,j} \otimes \cdots \otimes \boldsymbol{v}_L^T[\boldsymbol{U}_L]_{\cdot,j})$$

$$= -\sum_{j=1}^{m}[\boldsymbol{S}\boldsymbol{X}^T\boldsymbol{r}]_j\left(\prod_{k \neq l}[\boldsymbol{U}_k^T\boldsymbol{v}_k]_j\right)[\boldsymbol{U}_l]_{\cdot,j}.$$

Left-multiplying $\boldsymbol{U}_l^H$ (the conjugate transpose of $\boldsymbol{U}_l$) to both sides, we get

$$\boldsymbol{U}_l^H\dot{\boldsymbol{v}}_l = -\boldsymbol{S}\boldsymbol{X}^T\boldsymbol{r} \odot \prod_{k \neq l}^{\odot}\boldsymbol{U}_k^T\boldsymbol{v}_k, \tag{20}$$

where $\prod^{\odot}$ denotes the product using entry-wise multiplication $\odot$.

Now consider the rate of growth for the absolute value squared of the $j$-th component of $\boldsymbol{U}_l^T\boldsymbol{v}_l$:

$$\frac{d}{dt}|[\boldsymbol{U}_l^T\boldsymbol{v}_l]_j|^2 = \frac{d}{dt}[\boldsymbol{U}_l^T\boldsymbol{v}_l]_j[\boldsymbol{U}_l^T\boldsymbol{v}_l]_j^* = \frac{d}{dt}[\boldsymbol{U}_l^T\boldsymbol{v}_l]_j[\boldsymbol{U}_l^H\boldsymbol{v}_l]_j$$

$$= [\boldsymbol{U}_l^T\dot{\boldsymbol{v}}_l]_j[\boldsymbol{U}_l^H\boldsymbol{v}_l]_j + [\boldsymbol{U}_l^H\dot{\boldsymbol{v}}_l]_j[\boldsymbol{U}_l^T\boldsymbol{v}_l]_j$$

$$= 2\operatorname{Re}\left([\boldsymbol{U}_l^H\dot{\boldsymbol{v}}_l]_j[\boldsymbol{U}_l^T\boldsymbol{v}_l]_j\right)$$

$$= 2\operatorname{Re}\left(-[\boldsymbol{S}\boldsymbol{X}^T\boldsymbol{r}]_j\prod_{k=1}^{L}[\boldsymbol{U}_k^T\boldsymbol{v}_k]_j\right)$$

$$= \frac{d}{dt}|[\boldsymbol{U}_{l'}^T\boldsymbol{v}_{l'}]_j|^2 \quad \text{for any } l' \in [L],$$

so for any $j \in [m]$, the squared absolute value of the $j$-th components in $\boldsymbol{U}_l^T\boldsymbol{v}_l$ grow at the same rate for each layer $l \in [L]$. This means that the gap between any two different layers stays constant for all $t \geq 0$. Combining this with our conditions on initial directions, we have

$$|[\boldsymbol{U}_l^T\boldsymbol{v}_l(t)]_j|^2 - |[\boldsymbol{U}_L^T\boldsymbol{v}_L(t)]_j|^2 = |[\boldsymbol{U}_l^T\boldsymbol{v}_l(0)]_j|^2 - |[\boldsymbol{U}_L^T\boldsymbol{v}_L(0)]_j|^2$$

$$= \alpha^2|[\boldsymbol{U}_l^T\bar{\boldsymbol{v}}_l]_j|^2 - \alpha^2|[\boldsymbol{U}_L^T\bar{\boldsymbol{v}}_L]_j|^2 \geq \alpha^2\lambda, \tag{21}$$

for any $j \in [m]$, $l \in [L-1]$, and $t \geq 0$. This inequality also implies

$$|[\boldsymbol{U}_l^T\boldsymbol{v}_l(t)]_j|^2 \geq |[\boldsymbol{U}_L^T\boldsymbol{v}_L(t)]_j|^2 + \alpha^2\lambda \geq \alpha^2\lambda. \tag{22}$$

Let us now consider the time derivative of $\mathcal{L}(\boldsymbol{\Theta}(t))$. We have the following chain of upper bounds on the time derivative:

$$\frac{d}{dt}\mathcal{L}(\boldsymbol{\Theta}(t)) = \nabla_{\boldsymbol{\Theta}}\mathcal{L}(\boldsymbol{\Theta}(t))^T\dot{\boldsymbol{\Theta}}(t) = -\|\nabla_{\boldsymbol{\Theta}}\mathcal{L}(\boldsymbol{\Theta}(t))\|_2^2$$

$$\leq -\|\nabla_{\boldsymbol{v}_L}\mathcal{L}(\boldsymbol{\Theta}(t))\|_2^2 = -\|\dot{\boldsymbol{v}}_L(t)\|_2^2$$

$$\overset{(a)}{\leq} -\|\boldsymbol{U}_L^H\dot{\boldsymbol{v}}_L(t)\|_2^2 \overset{(b)}{=} -\left\|\boldsymbol{S}\boldsymbol{X}^T\boldsymbol{r}(t) \odot \prod_{k \neq L}^{\odot}\boldsymbol{U}_k^T\boldsymbol{v}_k(t)\right\|_2^2$$

$$= -\sum_{j=1}^{m}|[\boldsymbol{S}\boldsymbol{X}^T\boldsymbol{r}(t)]_j|^2\prod_{k \neq L}|[\boldsymbol{U}_k^T\boldsymbol{v}_k(t)]_j|^2$$

$$\stackrel{(c)}{\leq} -\alpha^{2L-2}\lambda^{L-1}\sum_{j=1}^{m}|[\boldsymbol{S}\boldsymbol{X}^T\boldsymbol{r}(t)]_j|^2$$

$$= -\alpha^{2L-2}\lambda^{L-1}\|\boldsymbol{S}\boldsymbol{X}^T\boldsymbol{r}(t)\|_2^2$$

$$\stackrel{(d)}{\leq} -\alpha^{2L-2}\lambda^{L-1}s_{\min}(\boldsymbol{S})^2\|\boldsymbol{X}^T\boldsymbol{r}(t)\|_2^2, \tag{23}$$

where (a) used the fact that $\|\dot{\boldsymbol{v}}_L(t)\|_2^2 \geq \|\boldsymbol{U}_L\boldsymbol{U}_L^H\dot{\boldsymbol{v}}_L(t)\|_2^2$ because it is a projection onto a subspace, and $\|\boldsymbol{U}_L\boldsymbol{U}_L^H\dot{\boldsymbol{v}}_L(t)\|_2^2 = \|\boldsymbol{U}_L^H\dot{\boldsymbol{v}}_L(t)\|_2^2$ because $\boldsymbol{U}_L^H\boldsymbol{U}_L = \boldsymbol{I}_{k_L}$; (b) is due to (20); (c) is due to (22); and (d) used the fact that $\boldsymbol{S} \in \mathbb{C}^{m \times d}$ is a matrix that has full column rank, so for any $\boldsymbol{z} \in \mathbb{C}^d$, we can use $\|\boldsymbol{S}\boldsymbol{z}\|_2 \geq s_{\min}(\boldsymbol{S})\|\boldsymbol{z}\|_2$ where $s_{\min}(\boldsymbol{S})$ is the minimum singular value of $\boldsymbol{S}$.

We now prove a lower bound on the quantity $\|\boldsymbol{X}^T\boldsymbol{r}(t)\|_2^2$. Recall from Section 2.1 the definition of $[\boldsymbol{r}(t)]_i = -y_i\exp(-y_i f(\boldsymbol{x}_i; \boldsymbol{\Theta}(t)))$ for classification problems. Also, recall the assumption that the dataset is linearly separable, which means that there exists a unit vector $\boldsymbol{z} \in \mathbb{R}^d$ such that

$$y_i\boldsymbol{x}_i^T\boldsymbol{z} \geq \gamma > 0$$

holds for all $i \in [n]$, for some $\gamma > 0$. Using these,

$$\|\boldsymbol{X}^T\boldsymbol{r}(t)\|_2^2 = \|\sum_{i=1}^{n}y_i\boldsymbol{x}_i\exp(-y_i f(\boldsymbol{x}_i; \boldsymbol{\Theta}(t)))\|_2^2$$

$$\geq [\boldsymbol{z}^T\sum_{i=1}^{n}y_i\boldsymbol{x}_i\exp(-y_i f(\boldsymbol{x}_i; \boldsymbol{\Theta}(t)))]^2$$

$$\geq \gamma^2[\sum_{i=1}^{n}\exp(-y_i f(\boldsymbol{x}_i; \boldsymbol{\Theta}(t)))]^2 = \gamma^2\mathcal{L}(\boldsymbol{\Theta}(t))^2.$$

Combining this with (23), we get

$$\frac{d}{dt}\mathcal{L}(\boldsymbol{\Theta}(t)) \leq -\alpha^{2L-2}\lambda^{L-1}s_{\min}(\boldsymbol{S})^2\gamma^2\mathcal{L}(\boldsymbol{\Theta}(t))^2,$$

which implies

$$\mathcal{L}(\boldsymbol{\Theta}(t)) \leq \frac{\mathcal{L}(\boldsymbol{\Theta}(0))}{1 + \alpha^{2L-2}\lambda^{L-1}s_{\min}(\boldsymbol{S})^2\gamma^2 t}.$$

Therefore, $\mathcal{L}(\boldsymbol{\Theta}(t)) \to 0$ as $t \to \infty$.

### D.1.2 CHARACTERIZING THE LIMIT DIRECTION

Since we proved that $\mathcal{L}(\boldsymbol{\Theta}(t)) \to 0$, the argument in the proof of Theorem 1 applies to this case, and shows that the parameters $\boldsymbol{v}_l$ converge in direction and align with $\dot{\boldsymbol{v}}_l = -\nabla_{\boldsymbol{v}_l}\mathcal{L}(\boldsymbol{\Theta})$. Let $\boldsymbol{v}_l^\infty := \lim_{t\to\infty}\frac{\boldsymbol{v}_l(t)}{\|\boldsymbol{v}_l(t)\|_2}$ be the limit direction of $\boldsymbol{v}_l$.

The remaining steps of the proof are as follows. We first prove that $\boldsymbol{S}\boldsymbol{X}^T\boldsymbol{r}(t)$ converges in direction $\boldsymbol{u}^\infty$. Using this $\boldsymbol{u}^\infty$, we derive a number of conditions that has to be satisfied by the limit directions of the parameters. Finally, we compare these conditions with the KKT conditions of the minimization problem, and finish the proof.

By Assumption 1, we have

$$f(\boldsymbol{x}; \boldsymbol{\Theta}) = \mathbf{M}(\boldsymbol{x}) \circ (\boldsymbol{v}_1, \dots, \boldsymbol{v}_L) = \sum_{j=1}^{m}[\boldsymbol{S}\boldsymbol{x}]_j\prod_{l=1}^{L}[\boldsymbol{U}_l^T\boldsymbol{v}_l]_j$$

$$= \Big[\sum_{j=1}^{m}\Big(\prod_{l=1}^{L}[\boldsymbol{U}_l^T\boldsymbol{v}_l]_j\Big)[\boldsymbol{S}]_{j,\cdot}\Big]\boldsymbol{x} = \boldsymbol{x}^T\boldsymbol{S}^T\Big(\prod_{l\in[L]}^{\odot}\boldsymbol{U}_l^T\boldsymbol{v}_l\Big) = \boldsymbol{x}^T\boldsymbol{S}^T\boldsymbol{\rho}.$$

Here, we defined $\boldsymbol{\rho} := \prod_{l\in[L]}^{\odot}\boldsymbol{U}_l^T\boldsymbol{v}_l \in \mathbb{C}^m$. Since the linear coefficients must be real, we have $\boldsymbol{S}^T\boldsymbol{\rho} \in \mathbb{R}^d$ for any real $\boldsymbol{v}_l$'s. Since $\boldsymbol{v}_l$'s converge in direction, $\boldsymbol{\rho}$ also converges in direction, to $\boldsymbol{\rho}^\infty := \prod_{l\in[L]}^{\odot}\boldsymbol{U}_l^T\boldsymbol{v}_l^\infty$. So we can express the limit direction of $\boldsymbol{\beta}(\boldsymbol{\Theta})$ as

$$\boldsymbol{\beta}(\boldsymbol{\Theta}^\infty) \propto \boldsymbol{S}^T\Big(\prod_{l\in[L]}^{\odot}\boldsymbol{U}_l^T\boldsymbol{v}_l^\infty\Big) = \boldsymbol{S}^T\boldsymbol{\rho}^\infty. \tag{24}$$

From (20) and alignment of $\boldsymbol{v}_l$ and $\dot{\boldsymbol{v}}_l$, we have

$$\lim_{t \to \infty} \boldsymbol{U}_l^H \boldsymbol{v}_l(t) = \lim_{t \to \infty} (\boldsymbol{U}_l^T \boldsymbol{v}_l(t))^* \propto - \lim_{t \to \infty} \boldsymbol{S} \boldsymbol{X}^T \boldsymbol{r}(t) \odot \prod_{k \neq l}^{\odot} \boldsymbol{U}_k^T \boldsymbol{v}_k(t). \tag{25}$$

Since all vectors $\boldsymbol{U}_l^T \boldsymbol{v}_l(t)$ converge in direction, the term $\boldsymbol{S} \boldsymbol{X}^T \boldsymbol{r}(t)$ should also converge in direction. Let $\boldsymbol{u}^\infty := \lim_{t \to \infty} \frac{\boldsymbol{S} \boldsymbol{X}^T \boldsymbol{r}(t)}{\|\boldsymbol{S} \boldsymbol{X}^T \boldsymbol{r}(t)\|_2}$. One can use the same argument as in Appendix C.2, more specifically (16) and (17), to show that $\boldsymbol{u}^\infty$ can be written as

$$\boldsymbol{u}^\infty = \lim_{t \to \infty} \frac{\boldsymbol{S} \sum_{i=1}^n \boldsymbol{x}_i [\boldsymbol{r}(t)]_i}{\|\boldsymbol{S} \sum_{i=1}^n \boldsymbol{x}_i [\boldsymbol{r}(t)]_i\|_2} = -\boldsymbol{S} \sum_{i=1}^n \nu_i y_i \boldsymbol{x}_i, \tag{26}$$

where $\nu_i \geq 0$ for all $i \in [n]$, and $\nu_j = 0$ for $\boldsymbol{x}_j$'s that are *not* support vectors, i.e., those satisfying $y_j \boldsymbol{x}_j^T \boldsymbol{S}^T \boldsymbol{\rho}^\infty > \min_{i \in [n]} y_i \boldsymbol{x}_i^T \boldsymbol{S}^T \boldsymbol{\rho}^\infty$.

Using $\boldsymbol{u}^\infty$, we can rewrite (25) as

$$\boldsymbol{U}_l^H \boldsymbol{v}_l^\infty \propto -\boldsymbol{u}^\infty \odot \prod_{k \neq l}^{\odot} \boldsymbol{U}_k^T \boldsymbol{v}_k^\infty,$$

for all $l \in [L]$. Element-wise multiplying $\boldsymbol{U}_l^T \boldsymbol{v}_l^\infty$ to both sides gives

$$\boldsymbol{U}_l^T \boldsymbol{v}_l^\infty \odot \boldsymbol{U}_l^H \boldsymbol{v}_l^\infty = |\boldsymbol{U}_l^T \boldsymbol{v}_l^\infty|^{\odot 2} \propto -\boldsymbol{u}^\infty \odot \prod_{k \in [L]}^{\odot} \boldsymbol{U}_k^T \boldsymbol{v}_k^\infty = -\boldsymbol{u}^\infty \odot \boldsymbol{\rho}^\infty, \tag{27}$$

where $\boldsymbol{a}^{\odot b}$ denotes element-wise $b$-th power of the vector $\boldsymbol{a}$. Since the LHS of (27) is a positive real number, we have

$$\arg(|[\boldsymbol{U}_l^T \boldsymbol{v}_l]_j|^2) = 0 = \arg([-\boldsymbol{u}^\infty]_j) + \arg([\boldsymbol{\rho}^\infty]_j), \tag{28}$$

so using this, (27) becomes

$$|\boldsymbol{U}_l^T \boldsymbol{v}_l^\infty|^{\odot 2} \propto |\boldsymbol{u}^\infty| \odot |\boldsymbol{\rho}^\infty|. \tag{29}$$

Now element-wise multiply (29) for all $l \in [L]$, then we get

$$|\boldsymbol{\rho}^\infty|^{\odot 2} \propto |\boldsymbol{u}^\infty|^{\odot L} \odot |\boldsymbol{\rho}^\infty|^{\odot L}. \tag{30}$$

A close look at (30) reveals that if $L \geq 2$, $\boldsymbol{\rho}^\infty$ and $\boldsymbol{u}^\infty$ must satisfy that

$$|[\boldsymbol{\rho}^\infty]_j| \neq 0 \implies |[\boldsymbol{u}^\infty]_j| \propto |[\boldsymbol{\rho}^\infty]_j|^{\frac{2}{L} - 1}, \tag{31}$$

for all $j \in [m]$. There is another condition that has to be satisfied when $L = 2$:

$$|[\boldsymbol{\rho}^\infty]_j| = 0, |[\boldsymbol{\rho}^\infty]_{j'}| \neq 0 \implies |[\boldsymbol{u}^\infty]_j| \leq |[\boldsymbol{u}^\infty]_{j'}|, \tag{32}$$

for any $j, j' \in [m]$; let us prove why. First, consider the time derivative of $[\boldsymbol{\rho}]_j = [\boldsymbol{U}_1^T \boldsymbol{v}_1]_j [\boldsymbol{U}_2^T \boldsymbol{v}_2]_j$.

$$\frac{d}{dt}[\boldsymbol{\rho}(t)]_j = [\boldsymbol{U}_1^T \boldsymbol{v}_1(t)]_j \frac{d}{dt}[\boldsymbol{U}_2^T \boldsymbol{v}_2(t)]_j + [\boldsymbol{U}_2^T \boldsymbol{v}_2(t)]_j \frac{d}{dt}[\boldsymbol{U}_1^T \boldsymbol{v}_1(t)]_j$$

$$\overset{(a)}{=} -[\boldsymbol{S} \boldsymbol{X}^T \boldsymbol{r}(t)]_j^* (|[\boldsymbol{U}_1^T \boldsymbol{v}_1(t)]_j|^2 + |[\boldsymbol{U}_2^T \boldsymbol{v}_2(t)]_j|^2), \tag{33}$$

where (a) used (20). Now consider

$$\frac{|\frac{d}{dt}[\boldsymbol{\rho}(t)]_j|}{\|\boldsymbol{S} \boldsymbol{X}^T \boldsymbol{r}(t)\|_2 |[\boldsymbol{\rho}(t)]_j|} = \frac{|[\boldsymbol{S} \boldsymbol{X}^T \boldsymbol{r}(t)]_j|}{\|\boldsymbol{S} \boldsymbol{X}^T \boldsymbol{r}(t)\|_2} \frac{|[\boldsymbol{U}_1^T \boldsymbol{v}_1(t)]_j|^2 + |[\boldsymbol{U}_2^T \boldsymbol{v}_2(t)]_j|^2}{|[\boldsymbol{\rho}(t)]_j|}. \tag{34}$$

We want to compare this quantity for different $j, j' \in [m]$. Before we do that, we take a look at the last term in the RHS of (34). Recall from (21) that

$$|[\boldsymbol{U}_1^T \boldsymbol{v}_1(t)]_j|^2 = |[\boldsymbol{U}_2^T \boldsymbol{v}_2(t)]_j|^2 + |[\boldsymbol{U}_1^T \boldsymbol{v}_1(0)]_j|^2 - |[\boldsymbol{U}_2^T \boldsymbol{v}_2(0)]_j|^2. \tag{35}$$

For simplicity, let $\delta_j := |[\boldsymbol{U}_1^T \boldsymbol{v}_1(0)]_j|^2 - |[\boldsymbol{U}_2^T \boldsymbol{v}_2(0)]_j|^2$, which is a positive number due to our assumption on initialization. Then, we can use (35) and $|[\boldsymbol{\rho}(t)]_j| = |[\boldsymbol{U}_1^T \boldsymbol{v}_1(t)]_j| |[\boldsymbol{U}_2^T \boldsymbol{v}_2(t)]_j|$ to show that

$$\frac{|[\boldsymbol{U}_1^T \boldsymbol{v}_1(t)]_j|^2 + |[\boldsymbol{U}_2^T \boldsymbol{v}_2(t)]_j|^2}{|[\boldsymbol{\rho}(t)]_j|} = \frac{2|[\boldsymbol{U}_2^T \boldsymbol{v}_2(t)]_j|^2 + \delta_j}{|[\boldsymbol{U}_2^T \boldsymbol{v}_2(t)]_j| \sqrt{|[\boldsymbol{U}_2^T \boldsymbol{v}_2(t)]_j|^2 + \delta_j}} \geq 2,$$

$$\lim_{t \to \infty} \frac{|[\boldsymbol{U}_1^T \boldsymbol{v}_1(t)]_j|^2 + |[\boldsymbol{U}_2^T \boldsymbol{v}_2(t)]_j|^2}{|[\boldsymbol{\rho}(t)]_j|} = 2 \ \text{ if } \lim_{t \to \infty} |[\boldsymbol{U}_2^T \boldsymbol{v}_2(t)]_j| = \infty.$$

Recall that we want to prove that (32) should necessarily hold. For the sake of contradiction, suppose that there exists $j \in [m]$ that satisfies $|[\boldsymbol{\rho}^\infty]_j| = 0$ but $|[\boldsymbol{u}^\infty]_j| > |[\boldsymbol{u}^\infty]_{j'}|$, for some $j' \in [m]$ satisfying $|[\boldsymbol{\rho}^\infty]_{j'}| \neq 0$. Note that having $|[\boldsymbol{\rho}^\infty]_j| = 0$ and $|[\boldsymbol{\rho}^\infty]_{j'}| \neq 0$ implies that $|[\boldsymbol{\rho}(t)]_{j'}| \to \infty$ and $\frac{|[\boldsymbol{\rho}(t)]_j|}{|[\boldsymbol{\rho}(t)]_{j'}|} \to 0$. We now want to compare the ratio of (34) for $j$ and $j'$. First, note that

$$\lim_{t \to \infty} \frac{|[\boldsymbol{S} \boldsymbol{X}^T \boldsymbol{r}(t)]_j| / \|\boldsymbol{S} \boldsymbol{X}^T \boldsymbol{r}(t)\|_2}{|[\boldsymbol{S} \boldsymbol{X}^T \boldsymbol{r}(t)]_{j'}| / \|\boldsymbol{S} \boldsymbol{X}^T \boldsymbol{r}(t)\|_2} = \frac{|[\boldsymbol{u}^\infty]_j|}{|[\boldsymbol{u}^\infty]_{j'}|} > 1. \tag{36}$$

Next, using $\frac{|[\boldsymbol{\rho}(t)]_j|}{|[\boldsymbol{\rho}(t)]_{j'}|} \to 0$ and the fact that $x \mapsto \frac{2x^2 + \delta}{x\sqrt{x^2 + \delta}}$ is a decreasing function of $x \geq 0$ for any $\delta > 0$, we have

$$\frac{(|[\boldsymbol{U}_1^T \boldsymbol{v}_1(t)]_j|^2 + |[\boldsymbol{U}_2^T \boldsymbol{v}_2(t)]_j|^2) / |[\boldsymbol{\rho}(t)]_j|}{(|[\boldsymbol{U}_1^T \boldsymbol{v}_1(t)]_{j'}|^2 + |[\boldsymbol{U}_2^T \boldsymbol{v}_2(t)]_{j'}|^2)) / |[\boldsymbol{\rho}(t)]_{j'}|} \geq 1, \tag{37}$$

for any $t \geq t_0$, when $t_0$ is large enough. Combining (36) and (37) to compare the ratio of (34) for $j$ and $j'$, we get that there exists some $t_0' \geq 0$ such that for any $t \geq t_0'$, we have

$$\frac{\left| \frac{d}{dt}[\boldsymbol{\rho}(t)]_j \right| / |[\boldsymbol{\rho}(t)]_j|}{\left| \frac{d}{dt}[\boldsymbol{\rho}(t)]_{j'} \right| / |[\boldsymbol{\rho}(t)]_{j'}|} > 1. \tag{38}$$

This implies that the ratio of the absolute value of time derivative of $[\boldsymbol{\rho}(t)]_j$ to the absolute value of current value of $[\boldsymbol{\rho}(t)]_j$ is strictly bigger than that of $[\boldsymbol{\rho}(t)]_{j'}$. Moreover, we saw in (33) that the phase of $\frac{d}{dt}[\boldsymbol{\rho}(t)]_j$ converges to that of $-[\boldsymbol{u}^\infty]_j^*$. Since this holds for all $t \geq t_0'$, (38) results in a growth of $|[\boldsymbol{\rho}(t)]_j|$ that is exponentially faster than that of $|[\boldsymbol{\rho}(t)]_{j'}|$, so $[\boldsymbol{\rho}(t)]_j$ becomes a dominant component in $\boldsymbol{\rho}(t)$ as $t \to \infty$. This contradicts that $[\boldsymbol{\rho}^\infty]_j = 0$, hence the condition (32) has to be satisfied.

So far, we have characterized a number of conditions (26), (28), (31), (32) that have to be satisfied by the limit directions $\boldsymbol{u}^\infty$ and $\boldsymbol{\rho}^\infty$ of $\boldsymbol{X}^T \boldsymbol{r}$ and $\boldsymbol{\rho}$. We now consider the following optimization problem and prove that these conditions are in fact the KKT conditions of the optimization problem. Consider

$$\underset{\boldsymbol{\rho} \in \mathbb{C}^m}{\text{minimize}} \quad \|\boldsymbol{\rho}\|_{2/L} \quad \text{subject to} \quad y_i \boldsymbol{x}_i^T \boldsymbol{S}^T \boldsymbol{\rho} \geq 1, \ \forall i \in [n]. \tag{39}$$

The KKT conditions of this problem are

$$\partial \|\boldsymbol{\rho}\|_{2/L} \ni \boldsymbol{S}^* \sum_{i=1}^n \mu_i y_i \boldsymbol{x}_i, \ \text{ and } \ \mu_i \geq 0, \ \mu_i(1 - y_i \boldsymbol{x}_i^T \boldsymbol{S}^T \boldsymbol{\rho}) = 0 \text{ for all } i \in [n],$$

where $\mu_1, \ldots, \mu_n$ are the dual variables. The symbol $\partial \|\cdot\|_{2/L}$ denotes the (local) subdifferential of the $\ell_{2/L}$ norm[1], which can be written as

$$\partial \|\boldsymbol{\rho}\|_1 = \{\boldsymbol{u} \in \mathbb{C}^m \mid |[\boldsymbol{u}]_j| \leq 1 \text{ for all } j \in [m], \text{ and } [\boldsymbol{\rho}]_j \neq 0 \implies [\boldsymbol{u}]_j = \exp(\sqrt{-1} \arg([\boldsymbol{\rho}]_j))\},$$

if $L = 2$ (in this case $\partial \|\boldsymbol{\rho}\|_1$ is the global subdifferential), and

$$\partial \|\boldsymbol{\rho}\|_{2/L} = \left\{ \boldsymbol{u} \in \mathbb{C}^m \mid [\boldsymbol{\rho}]_j \neq 0 \implies [\boldsymbol{u}]_j = \frac{2}{L} |[\boldsymbol{\rho}]_j|^{\frac{2}{L} - 1} \exp(\sqrt{-1} \arg([\boldsymbol{\rho}]_j)) \right\},$$

if $L > 2$. By replacing $\mu_i$'s with $\nu_i$'s defined in (26), we can check from (26), (28), (31), (32) that the that $\boldsymbol{\rho}^\infty$ and $\boldsymbol{u}^\infty$ satisfy the KKT conditions up to scaling. Therefore, by (24), $\boldsymbol{\beta}(\boldsymbol{\Theta}(t))$ converges in direction aligned with $\boldsymbol{S}^T \boldsymbol{\rho}^\infty$, where $\boldsymbol{\rho}^\infty$ is again aligned with a stationary point (global minimum in case of $L = 2$) of the optimization problem (39).

If $\boldsymbol{S}$ is invertible, we can get $\boldsymbol{S}^{-T} \boldsymbol{\beta}(\boldsymbol{\Theta}^\infty) \propto \boldsymbol{\rho}^\infty$. Plugging this into the optimization problem (39) gives the last statement of the theorem.

---

[1] the definition of subdifferentials used here is taken from Gunasekar et al. (2018b).

### D.2 PROOF OF COROLLARY 2

It suffices to prove that linear diagonal networks satisfy Assumption 1, with $S = I_d$. The proof is very straightforward, since $\mathbf{M}_{\mathrm{diag}}(x) \in \mathbb{R}^{d \times \cdots \times d}$ has $[\mathbf{M}_{\mathrm{diag}}(x)]_{j,j,\ldots,j} = [x]_j$ while all the remaining entries are zero. It is straightforward to verify that $\mathbf{M}_{\mathrm{diag}}(x)$ satisfies Assumption 1 with $S = U_1 = \cdots = U_L = I_d$. A direct substitution into Theorem 2 gives the corollary.

### D.3 PROOF OF COROLLARY 3

For full-length convolutional networks ($k_1 = \cdots = k_L = d$), we will prove that they satisfy Assumption 1 with $S = d^{\frac{L-1}{2}} F$ and $U_1 = \cdots = U_L = F^*$, where $F \in \mathbb{C}^{d \times d}$ is the matrix of discrete Fourier transform basis $[F]_{j,k} = \frac{1}{\sqrt{d}} \exp(-\frac{\sqrt{-1} \cdot 2\pi(j-1)(k-1)}{d})$ and $F^*$ is the complex conjugate of $F$.

For simplicity of notation, define $\psi = \exp(-\frac{\sqrt{-1} \cdot 2\pi}{d})$. With such matrices $S$ and $U_1, \ldots, U_L$, we can write $M(x)$ as

$$\mathbf{M}(x) = \sum_{j=1}^{d} [Sx]_j ([U_1]_{\cdot,j} \otimes [U_2]_{\cdot,j} \otimes \cdots \otimes [U_L]_{\cdot,j})$$

$$= \sum_{j=1}^{d} \left[ d^{\frac{L-2}{2}} \sum_{k=1}^{d} [x]_k \psi^{(j-1)(k-1)} \right] \begin{bmatrix} \psi^0/\sqrt{d} \\ \psi^{-(j-1)}/\sqrt{d} \\ \psi^{-2(j-1)}/\sqrt{d} \\ \vdots \\ \psi^{-(d-1)(j-1)}/\sqrt{d} \end{bmatrix}^{\otimes L},$$

where $a^{\otimes L}$ denotes the $L$-times tensor product of $a$. We will show that $\mathbf{M}(x) = \mathbf{M}_{\mathrm{conv}}(x)$. For any $j_1, \ldots, j_L \in [d]$,

$$[\mathbf{M}(x)]_{j_1,\ldots,j_L} = \frac{1}{d} \sum_{l=1}^{d} \left[ \sum_{k=1}^{d} [x]_k \psi^{(l-1)(k-1)} \right] \psi^{-(l-1)(\sum_{q=1}^{L} j_q - L)}$$

$$= \frac{1}{d} \sum_{k=1}^{d} [x]_k \sum_{l=1}^{d} \psi^{(l-1)(k-1-\sum_{q=1}^{L} j_q + L)}.$$

Recall that

$$\sum_{l=1}^{d} \psi^{(l-1)(k-1-\sum_{q=1}^{L} j_q + L)} = \begin{cases} d & \text{if } k-1-\sum_{q=1}^{L} j_q + L \text{ is a multiple of } d, \\ 0 & \text{otherwise.} \end{cases}$$

Using this, we have

$$[\mathbf{M}(x)]_{j_1,\ldots,j_L} = \frac{1}{d} \sum_{k=1}^{d} [x]_k \sum_{l=1}^{d} \psi^{(l-1)(k-1-\sum_{q=1}^{L} j_q + L)}$$

$$= [x]_{\sum_{q=1}^{L} j_q - L + 1 \bmod d} = [\mathbf{M}_{\mathrm{conv}}(x)]_{j_1,\ldots,j_L}.$$

Hence, linear full-length convolutional networks satisfy Assumption 1 with $S = d^{\frac{L-1}{2}} F$. A direct substitution into Theorem 2 and then using the fact that $|[Fz]_j| = |[F^*z]_j|$ for any real vector $z \in \mathbb{R}^d$ gives the corollary.

## E PROOFS OF THEOREM 3 AND COROLLARY 4

### E.1 PROOF OF THEOREM 3

#### E.1.1 CONVERGENCE OF LOSS TO ZERO

Since Theorem 3 does not assume the existence of $t_0 \geq 0$ satisfying $\mathcal{L}(\Theta(t_0)) < 1$, we need to first show that given the conditions on initialization, the training loss $\mathcal{L}(\Theta(t))$ converges to zero. Since

$L = 2$ and $\mathbf{M}(\boldsymbol{x}) = \boldsymbol{U}_1 \operatorname{diag}(\boldsymbol{s}) \boldsymbol{U}_2^T$, we can write the gradient flow dynamics from Section 2.1 as

$$\begin{aligned}
\dot{\boldsymbol{v}}_1 &= -\mathbf{M}(\boldsymbol{X}^T \boldsymbol{r}) \circ (\boldsymbol{I}_{k_1}, \boldsymbol{v}_2) = -r \boldsymbol{U}_1 \operatorname{diag}(\boldsymbol{s}) \boldsymbol{U}_2^T \boldsymbol{v}_2, \\
\dot{\boldsymbol{v}}_2 &= -\mathbf{M}(\boldsymbol{X}^T \boldsymbol{r}) \circ (\boldsymbol{v}_1, \boldsymbol{I}_{k_2}) = -r \boldsymbol{U}_2 \operatorname{diag}(\boldsymbol{s}) \boldsymbol{U}_1^T \boldsymbol{v}_1,
\end{aligned} \tag{40}$$

where $r(t) = -y \exp(-y f(\boldsymbol{x}; \boldsymbol{\Theta}(t)))$ is the residual of the data point $(\boldsymbol{x}, y)$. From (40) we get

$$\boldsymbol{U}_l^T \dot{\boldsymbol{v}}_1 = -r \boldsymbol{s} \odot \boldsymbol{U}_2^T \boldsymbol{v}_2, \ \ \boldsymbol{U}_2^T \dot{\boldsymbol{v}}_2 = -r \boldsymbol{s} \odot \boldsymbol{U}_1^T \boldsymbol{v}_1. \tag{41}$$

Now consider the rate of growth for the $j$-th component of $\boldsymbol{U}_1^T \boldsymbol{v}_1$ squared:

$$\frac{d}{dt} [\boldsymbol{U}_1^T \boldsymbol{v}_1]_j^2 = 2[\boldsymbol{U}_1^T \boldsymbol{v}_1]_j [\boldsymbol{U}_1^T \dot{\boldsymbol{v}}_1]_j = -2r[\boldsymbol{s}]_j [\boldsymbol{U}_1^T \boldsymbol{v}_1]_j [\boldsymbol{U}_2^T \boldsymbol{v}_2]_j = \frac{d}{dt} [\boldsymbol{U}_2^T \boldsymbol{v}_2]_j^2. \tag{42}$$

So for any $j \in [m]$, $[\boldsymbol{U}_1^T \boldsymbol{v}_1]_j^2$ and $[\boldsymbol{U}_2^T \boldsymbol{v}_2]_j^2$ grow at the same rate. This means that the gap between the two layers stays constant for all $t \geq 0$. Combining this with our conditions on initial directions,

$$\begin{aligned}
[\boldsymbol{U}_1^T \boldsymbol{v}_1(t)]_j^2 - [\boldsymbol{U}_2^T \boldsymbol{v}_2(t)]_j^2 &= [\boldsymbol{U}_1^T \boldsymbol{v}_1(0)]_j^2 - [\boldsymbol{U}_2^T \boldsymbol{v}_2(0)]_j^2 \\
&= \alpha^2 [\boldsymbol{U}_1^T \bar{\boldsymbol{v}}_1]_j^2 - \alpha^2 [\boldsymbol{U}_2^T \bar{\boldsymbol{v}}_2]_j^2 \geq \alpha^2 \lambda,
\end{aligned} \tag{43}$$

for any $j \in [m]$ and $t \geq 0$. This inequality implies

$$[\boldsymbol{U}_1^T \boldsymbol{v}_1(t)]_j^2 \geq [\boldsymbol{U}_2^T \boldsymbol{v}_2(t)]_j^2 + \alpha^2 \lambda \geq \alpha^2 \lambda. \tag{44}$$

Let us now consider the time derivative of $\mathcal{L}(\boldsymbol{\Theta}(t))$. We have the following chain of upper bounds on the time derivative:

$$\begin{aligned}
\frac{d}{dt} \mathcal{L}(\boldsymbol{\Theta}(t)) &= \nabla_{\boldsymbol{\Theta}} \mathcal{L}(\boldsymbol{\Theta}(t))^T \dot{\boldsymbol{\Theta}}(t) = -\|\nabla_{\boldsymbol{\Theta}} \mathcal{L}(\boldsymbol{\Theta}(t))\|_2^2 \\
&\leq -\|\nabla_{\boldsymbol{v}_2} \mathcal{L}(\boldsymbol{\Theta}(t))\|_2^2 = -\|\dot{\boldsymbol{v}}_2(t)\|_2^2 \\
&\overset{(a)}{\leq} -\|\boldsymbol{U}_2^T \dot{\boldsymbol{v}}_2(t)\|_2^2 \overset{(b)}{=} -r(t)^2 \left\| \boldsymbol{s} \odot \boldsymbol{U}_1^T \boldsymbol{v}_1(t) \right\|_2^2 \\
&= -r(t)^2 \sum_{j=1}^m [\boldsymbol{s}]_j^2 [\boldsymbol{U}_1^T \boldsymbol{v}_1(t)]_j^2 \\
&\overset{(c)}{\leq} -\alpha^2 \lambda r(t)^2 \sum_{j=1}^m [\boldsymbol{s}]_j^2 \\
&= -\alpha^2 \lambda \|\boldsymbol{s}\|_2^2 \mathcal{L}(\boldsymbol{\Theta}(t))^2,
\end{aligned}$$

where (a) used the fact that $\|\dot{\boldsymbol{v}}_2(t)\|_2^2 \geq \|\boldsymbol{U}_2 \boldsymbol{U}_2^T \dot{\boldsymbol{v}}_2(t)\|_2^2$ because it is a projection onto a subspace, and $\|\boldsymbol{U}_2 \boldsymbol{U}_2^T \dot{\boldsymbol{v}}_L(t)\|_2^2 = \|\boldsymbol{U}_2^T \dot{\boldsymbol{v}}_2(t)\|_2^2$ because $\boldsymbol{U}_2^T \boldsymbol{U}_2 = \boldsymbol{I}_{k_2}$; (b) is due to (41); (c) is due to (44). From this, we get

$$\mathcal{L}(\boldsymbol{\Theta}(t)) \leq \frac{\mathcal{L}(\boldsymbol{\Theta}(0))}{1 + \alpha^2 \lambda \|\boldsymbol{s}\|_2^2 t}.$$

Therefore, $\mathcal{L}(\boldsymbol{\Theta}(t)) \to 0$ as $t \to \infty$.

### E.1.2 CHARACTERIZING THE LIMIT DIRECTION

Since we proved that $\mathcal{L}(\boldsymbol{\Theta}(t)) \to 0$, the argument in the proof of Theorem 1 applies to this case, and shows that the parameters $\boldsymbol{v}_l$ converge in direction and align with $\dot{\boldsymbol{v}}_l = -\nabla_{\boldsymbol{v}_l} \mathcal{L}(\boldsymbol{\Theta})$. Let $\boldsymbol{v}_l^\infty := \lim_{t \to \infty} \frac{\boldsymbol{v}_l(t)}{\|\boldsymbol{v}_l(t)\|_2}$ be the limit direction of $\boldsymbol{v}_l$. As done in the proof of Theorem 2, define $\boldsymbol{\rho}(t) = \boldsymbol{U}_1^T \boldsymbol{v}_1(t) \odot \boldsymbol{U}_2^T \boldsymbol{v}_2(t)$ and $\boldsymbol{\rho}^\infty = \boldsymbol{U}_1^T \boldsymbol{v}_1^\infty \odot \boldsymbol{U}_2^T \boldsymbol{v}_2^\infty$.

It follows from $r(t) = -y \exp(-y f(\boldsymbol{x}; \boldsymbol{\Theta}(t)))$ that we have $\operatorname{sign}(r(t)) = -\operatorname{sign}(y)$. Using this, (41), and alignment of $\boldsymbol{v}_l$ and $\dot{\boldsymbol{v}}_l$, we have

$$\boldsymbol{U}_1^T \boldsymbol{v}_1^\infty \propto y \boldsymbol{s} \odot \boldsymbol{U}_2^T \boldsymbol{v}_2^\infty, \ \ \boldsymbol{U}_2^T \boldsymbol{v}_2^\infty \propto y \boldsymbol{s} \odot \boldsymbol{U}_1^T \boldsymbol{v}_1^\infty. \tag{45}$$

Element-wise multiplying $\boldsymbol{U}_l^T \boldsymbol{v}_l^\infty$ to both sides gives

$$(\boldsymbol{U}_1^T \boldsymbol{v}_1^\infty)^{\odot 2} \propto y \boldsymbol{s} \odot \boldsymbol{\rho}^\infty, \ \ (\boldsymbol{U}_2^T \boldsymbol{v}_2^\infty)^{\odot 2} \propto y \boldsymbol{s} \odot \boldsymbol{\rho}^\infty. \tag{46}$$

Since the LHSs are positive and $s$ is positive, the following equations have to be satisfied for all $j \in [m]$:

$$\text{sign}(y) = \text{sign}([\boldsymbol{\rho}^\infty]_j). \tag{47}$$

Now, multiplying both sides of the two equations (46), we get

$$(\boldsymbol{\rho}^\infty)^{\odot 2} \propto \boldsymbol{s}^{\odot 2} \odot (\boldsymbol{\rho}^\infty)^{\odot 2}. \tag{48}$$

From (48), $\boldsymbol{\rho}^\infty$ must satisfy that

$$[\boldsymbol{\rho}^\infty]_j \neq 0, [\boldsymbol{\rho}^\infty]_{j'} \neq 0 \implies |[\boldsymbol{s}]_j| = |[\boldsymbol{s}]_{j'}|, \tag{49}$$

for all $j, j' \in [m]$. As in the proof of Theorem 2, there is another condition that has to be satisfied:

$$[\boldsymbol{\rho}^\infty]_j = 0, [\boldsymbol{\rho}^\infty]_{j'} \neq 0 \implies |[\boldsymbol{s}]_j| \leq |[\boldsymbol{s}]_{j'}|, \tag{50}$$

for any $j, j' \in [m]$; let us prove why. First, consider the time derivative of $[\boldsymbol{\rho}]_j = [\boldsymbol{U}_1^T \boldsymbol{v}_1]_j [\boldsymbol{U}_2^T \boldsymbol{v}_2]_j$.

$$\frac{d}{dt}[\boldsymbol{\rho}(t)]_j = [\boldsymbol{U}_1^T \boldsymbol{v}_1(t)]_j \frac{d}{dt}[\boldsymbol{U}_2^T \boldsymbol{v}_2(t)]_j + [\boldsymbol{U}_2^T \boldsymbol{v}_2(t)]_j \frac{d}{dt}[\boldsymbol{U}_1^T \boldsymbol{v}_1(t)]_j$$

$$\stackrel{(a)}{=} -r(t)[\boldsymbol{s}]_j([\boldsymbol{U}_1^T \boldsymbol{v}_1(t)]_j^2 + [\boldsymbol{U}_2^T \boldsymbol{v}_2(t)]_j^2),$$

where (a) used (41). Now consider

$$\frac{\left|\frac{d}{dt}[\boldsymbol{\rho}(t)]_j\right|}{|r(t)||[\boldsymbol{\rho}(t)]_j|} = |[\boldsymbol{s}]_j| \frac{[\boldsymbol{U}_1^T \boldsymbol{v}_1(t)]_j^2 + [\boldsymbol{U}_2^T \boldsymbol{v}_2(t)]_j^2}{|[\boldsymbol{\rho}(t)]_j|}. \tag{51}$$

We want to compare this quantity for different $j, j' \in [m]$. Before we do that, we take a look at the last term in the RHS of (51). Recall from (43) that

$$[\boldsymbol{U}_1^T \boldsymbol{v}_1(t)]_j^2 = [\boldsymbol{U}_2^T \boldsymbol{v}_2(t)]_j^2 + [\boldsymbol{U}_1^T \boldsymbol{v}_1(0)]_j^2 - [\boldsymbol{U}_2^T \boldsymbol{v}_2(0)]_j^2. \tag{52}$$

For simplicity, let $\delta_j := [\boldsymbol{U}_1^T \boldsymbol{v}_1(0)]_j^2 - [\boldsymbol{U}_2^T \boldsymbol{v}_2(0)]_j^2$, which is a positive number due to our assumption on initialization. Then, we can use (52) and $|[\boldsymbol{\rho}(t)]_j| = |[\boldsymbol{U}_1^T \boldsymbol{v}_1(t)]_j||[\boldsymbol{U}_2^T \boldsymbol{v}_2(t)]_j|$ to show that

$$\frac{[\boldsymbol{U}_1^T \boldsymbol{v}_1(t)]_j^2 + [\boldsymbol{U}_2^T \boldsymbol{v}_2(t)]_j^2}{|[\boldsymbol{\rho}(t)]_j|} = \frac{2[\boldsymbol{U}_2^T \boldsymbol{v}_2(t)]_j^2 + \delta_j}{|[\boldsymbol{U}_2^T \boldsymbol{v}_2(t)]_j|\sqrt{[\boldsymbol{U}_2^T \boldsymbol{v}_2(t)]_j^2 + \delta_j}} \geq 2,$$

$$\lim_{t \to \infty} \frac{[\boldsymbol{U}_1^T \boldsymbol{v}_1(t)]_j^2 + [\boldsymbol{U}_2^T \boldsymbol{v}_2(t)]_j^2}{|[\boldsymbol{\rho}(t)]_j|} = 2 \quad \text{if } \lim_{t \to \infty} |[\boldsymbol{U}_2^T \boldsymbol{v}_2(t)]_j| = \infty.$$

Recall that we want to prove that (50) should necessarily hold. For the sake of contradiction, suppose that there exists $j \in [m]$ that satisfies $[\boldsymbol{\rho}^\infty]_j = 0$ but $|[\boldsymbol{s}]_j| > |[\boldsymbol{s}]_{j'}|$, for some $j' \in [m]$ satisfying $[\boldsymbol{\rho}^\infty]_{j'} \neq 0$. Note that having $[\boldsymbol{\rho}^\infty]_j = 0$ and $[\boldsymbol{\rho}^\infty]_{j'} \neq 0$ implies that $|[\boldsymbol{\rho}(t)]_{j'}| \to \infty$ and $\frac{|[\boldsymbol{\rho}(t)]_j|}{|[\boldsymbol{\rho}(t)]_{j'}|} \to 0$. We now want to compare the ratio of (51) for $j$ and $j'$. Using $\frac{|[\boldsymbol{\rho}(t)]_j|}{|[\boldsymbol{\rho}(t)]_{j'}|} \to 0$ and the fact that $x \mapsto \frac{2x^2+\delta}{x\sqrt{x^2+\delta}}$ is a decreasing function of $x \geq 0$ for any $\delta > 0$, we have

$$\frac{([\boldsymbol{U}_1^T \boldsymbol{v}_1(t)]_j^2 + [\boldsymbol{U}_2^T \boldsymbol{v}_2(t)]_j^2)/|[\boldsymbol{\rho}(t)]_j|}{([\boldsymbol{U}_1^T \boldsymbol{v}_1(t)]_{j'}^2 + [\boldsymbol{U}_2^T \boldsymbol{v}_2(t)]_{j'}^2))/|[\boldsymbol{\rho}(t)]_{j'}|} \geq 1, \tag{53}$$

for any $t \geq t_0$, when $t_0$ is large enough. Combining $\frac{|[\boldsymbol{s}]_j|}{|[\boldsymbol{s}]_{j'}|} > 1$ and (53) to compare the ratio of (51) for $j$ and $j'$, there exists some $t_0 \geq 0$ such that for any $t \geq t_0$, we have

$$\frac{\left|\frac{d}{dt}[\boldsymbol{\rho}(t)]_j\right|/|[\boldsymbol{\rho}(t)]_j|}{\left|\frac{d}{dt}[\boldsymbol{\rho}(t)]_{j'}\right|/|[\boldsymbol{\rho}(t)]_{j'}|} > 1. \tag{54}$$

This implies that the ratio of the absolute value of time derivative of $[\boldsymbol{\rho}(t)]_j$ to the absolute value of current value of $[\boldsymbol{\rho}(t)]_j$ is strictly bigger than that of $[\boldsymbol{\rho}(t)]_{j'}$. Moreover, by the definition of $r(t)$, $\frac{d}{dt}[\boldsymbol{\rho}(t)]_j$ does not change sign over time. Since this holds for all $t \geq t_0$, (54) results in a growth of $|[\boldsymbol{\rho}(t)]_j|$ that is exponentially faster than that of $|[\boldsymbol{\rho}(t)]_{j'}|$, so $[\boldsymbol{\rho}(t)]_j$ becomes a dominant component in $\boldsymbol{\rho}(t)$ as $t \to \infty$. This contradicts that $[\boldsymbol{\rho}^\infty]_j = 0$, hence the condition (50) has to be satisfied.

So far, we have characterized some conditions (47), (49), (50) that have to be satisfied by the limit direction $\rho^\infty$ of $\rho$. We now consider the following optimization problem and prove that these conditions are in fact the KKT conditions of the optimization problem. Consider

$$\underset{\rho \in \mathbb{R}^m}{\text{minimize}} \quad \|\rho\|_1 \quad \text{subject to} \quad y s^T \rho \geq 1. \tag{55}$$

The KKT condition of this problem is

$$\partial \|\rho\|_1 \ni y s,$$

where the global subdifferential $\partial \|\cdot\|_1$ is defined as

$$\partial \|\rho\|_1 = \{u \in \mathbb{R}^m \mid |[u]_j| \leq 1 \text{ for all } j \in [m], \text{ and } [\rho]_j \neq 0 \implies [u]_j = \text{sign}([\rho]_j)\}.$$

We can check from (47), (49), (50) that the that $\rho^\infty$ satisfies the KKT condition up to scaling.

Now, how do we characterize $v_1^\infty$ and $v_2^\infty$ in terms of $\rho^\infty$? Let $\eta_1^\infty := U_1^T v_1^\infty$ and $\eta_2^\infty := U_2^T v_2^\infty$. Then, $v_l^\infty = U_l \eta_l^\infty = U_l U_l^T v_l^\infty$ holds because any component orthogonal to the column space of $U_l$ stays unchanged while the component in the column space of $U_l$ diverges to infinity. By (42), $|\eta_1^\infty| = |\eta_2^\infty| = |\rho^\infty|^{\odot 1/2}$. By (45), we have $\text{sign}(\eta_1^\infty) = \text{sign}(y) \odot \text{sign}(\eta_2^\infty)$.

### E.2 PROOF OF COROLLARY 4

The proof of Corollary 4 boils down to characterizing the SVD of $\mathbf{M}_{\text{conv}}(x)$.

#### E.2.1 THE $k_1 = 1$ CASE

First, it is straightforward to check that for $L = 2$ and $k_1 = 1$, we have

$$\beta_{\text{conv}}(\Theta_{\text{conv}}) = v_1 v_2.$$

For $k_1 = 1$, the data tensor is simply $\mathbf{M}_{\text{conv}}(x) = x^T$. Thus, we have $U_1 = 1$, $U_2 = \frac{x}{\|x\|_2}$, and $s = \|x\|_2$. Substituting $U_1$ and $U_2$ to the theorem gives the condition on initial directions in Corollary 4. Also, the theorem implies us that the limit direction $v_2^\infty$ of $v_2$ satisfies $v_2^\infty \propto y v_1^\infty x$. Using this, it is easy to check that

$$\beta_{\text{conv}}(\Theta_{\text{conv}}^\infty) \propto v_1^\infty v_2^\infty \propto y x.$$

#### E.2.2 THE $k_1 = 2$ CASE

First, it is straightforward to check that for $L = 2$ and $k_1 = 2$, we have

$$\beta_{\text{conv}}(\Theta_{\text{conv}}) = \begin{bmatrix} [v_1]_1 & 0 & 0 & \cdots & 0 & [v_1]_2 \\ [v_1]_2 & [v_1]_1 & 0 & \cdots & 0 & 0 \\ 0 & [v_1]_2 & [v_1]_1 & \cdots & 0 & 0 \\ \vdots & \vdots & \vdots & \ddots & \vdots & \vdots \\ 0 & 0 & 0 & \cdots & [v_1]_1 & 0 \\ 0 & 0 & 0 & \cdots & [v_1]_2 & [v_1]_1 \end{bmatrix} v_2. \tag{56}$$

For $k_1 = 2$, by definition, the data tensor is

$$\mathbf{M}_{\text{conv}}(x) = \begin{bmatrix} x^T \\ \overleftarrow{x}^T \end{bmatrix},$$

and it is straightforward to check that the SVD of this matrix is

$$\mathbf{M}_{\text{conv}}(x) = \begin{bmatrix} x^T \\ \overleftarrow{x}^T \end{bmatrix} = \begin{bmatrix} 1/\sqrt{2} & 1/\sqrt{2} \\ 1/\sqrt{2} & -1/\sqrt{2} \end{bmatrix} \begin{bmatrix} \sqrt{\|x\|_2^2 + x^T \overleftarrow{x}} & 0 \\ 0 & \sqrt{\|x\|_2^2 - x^T \overleftarrow{x}} \end{bmatrix} \begin{bmatrix} \frac{x^T + \overleftarrow{x}^T}{\sqrt{2}\sqrt{\|x\|_2^2 + x^T \overleftarrow{x}}} \\ \frac{x^T - \overleftarrow{x}^T}{\sqrt{2}\sqrt{\|x\|_2^2 - x^T \overleftarrow{x}}} \end{bmatrix},$$

so

$$U_1 = \begin{bmatrix} 1/\sqrt{2} & 1/\sqrt{2} \\ 1/\sqrt{2} & -1/\sqrt{2} \end{bmatrix}, U_2 = \begin{bmatrix} \frac{x + \overleftarrow{x}}{\sqrt{2}\sqrt{\|x\|_2^2 + x^T \overleftarrow{x}}} & \frac{x - \overleftarrow{x}}{\sqrt{2}\sqrt{\|x\|_2^2 - x^T \overleftarrow{x}}} \end{bmatrix}, s = \begin{bmatrix} \sqrt{\|x\|_2^2 + x^T \overleftarrow{x}} \\ \sqrt{\|x\|_2^2 - x^T \overleftarrow{x}} \end{bmatrix}.$$

Substituting $U_1$ and $U_2$ to the theorem gives the conditions on initial directions. Also, note that the maximum singular value depends on the sign of $x^T \overleftarrow{x}$. Consider the optimization problem in the theorem statement:

$$\text{minimize}_{\rho \in \mathbb{R}^m} \quad \|\rho\|_1 \quad \text{subject to} \quad y s^T \rho \geq 1.$$

If $x^T \overleftarrow{x} > 0$, then the solution $\rho^\infty$ to this problem is in the direction of $[y \ \ 0]$. Therefore, the limit directions $v_1^\infty$ and $v_2^\infty$ will be of the form

$$v_1^\infty \propto c_1 \begin{bmatrix} 1 \\ 1 \end{bmatrix}, \quad v_2^\infty \propto c_2(x + \overleftarrow{x}),$$

where $\text{sign}(c_1)\text{sign}(c_2) = \text{sign}(y)$. Using (56), it is straightforward to check that

$$\beta_{\text{conv}}(\Theta_{\text{conv}}^\infty) \propto y \begin{bmatrix} 1 & 0 & 0 & \cdots & 0 & 1 \\ 1 & 1 & 0 & \cdots & 0 & 0 \\ 0 & 1 & 1 & \cdots & 0 & 0 \\ \vdots & \vdots & \vdots & \ddots & \vdots & \vdots \\ 0 & 0 & 0 & \cdots & 1 & 0 \\ 0 & 0 & 0 & \cdots & 1 & 1 \end{bmatrix} (x + \overleftarrow{x}) = y(2x + \overleftarrow{x} + \overrightarrow{x}).$$

Similarly, if $x^T \overleftarrow{x} < 0$, then the solution $\rho^\infty$ is in the direction of $[0 \ \ y]$. Using (56), we have

$$\beta_{\text{conv}}(\Theta_{\text{conv}}^\infty) \propto y \begin{bmatrix} 1 & 0 & 0 & \cdots & 0 & -1 \\ -1 & 1 & 0 & \cdots & 0 & 0 \\ 0 & -1 & 1 & \cdots & 0 & 0 \\ \vdots & \vdots & \vdots & \ddots & \vdots & \vdots \\ 0 & 0 & 0 & \cdots & 1 & 0 \\ 0 & 0 & 0 & \cdots & -1 & 1 \end{bmatrix} (x - \overleftarrow{x}) = y(2x - \overleftarrow{x} - \overrightarrow{x}).$$

# F  PROOFS OF THEOREM 5, COROLLARIES 5, 6 & 7, AND LEMMA 4

## F.1  PROOF OF LEMMA 4

In this subsection, we restate Lemma 4 and prove it.

**Lemma 4.** *Consider the system of ODEs, where $p, q : \mathbb{R} \to \mathbb{R}$:*

$$\dot{p} = p^{L-2}q, \quad \dot{q} = p^{L-1}, \quad p(0) = 1, \quad q(0) = 0.$$

*Then, the solutions $p_L(t)$ and $q_L(t)$ are continuous on their maximal interval of existence of the form $(-c, c) \subset \mathbb{R}$ for some $c \in (0, \infty]$. Define $h_L(t) = p_L(t)^{L-1}q_L(t)$; then, $h_L(t)$ is odd and strictly increasing, satisfying $\lim_{t \uparrow c} h_L(t) = \infty$ and $\lim_{t \downarrow -c} h_L(t) = -\infty$.*

**Proof**   First, continuity (and also continuous differentiability) of $p(t)$ and $q(t)$ is straightforward because the RHSs of the ODEs are differentiable in $p$ and $q$. Next, define $\tilde{p}(t) = p(-t)$ and $\tilde{q}(t) = -q(-t)$. Then, one can show that $\tilde{p}$ and $\tilde{q}$ are also the solution of the ODE because

$$\frac{d}{dt}\tilde{p}(t) = \frac{d}{dt}p(-t) = -\dot{p}(-t) = -p(-t)^{L-2}q(-t) = \tilde{p}(t)^{L-2}\tilde{q}(t),$$

$$\frac{d}{dt}\tilde{q}(t) = -\frac{d}{dt}q(-t) = \dot{q}(-t) = p(-t)^{L-1} = \tilde{p}(t)^{L-1}.$$

However, by the Picard-Lindelöf theorem, the solution has to be unique; this means that $p(t) = \tilde{p}(t) = p(-t)$ and $q(t) = \tilde{q}(t) = -q(-t)$, which proves that $p$ is even and $q$ is odd and also implies that the domain of $p$ and $q$ has to be of the form $(-c, c)$ (i.e. symmetric around the origin) and $h = p^{L-1}q$ is odd.

To show that $h$ is strictly increasing, it suffices to show that $p$ and $q$ are both strictly increasing on $[0, c)$. To this end, we show that $p(t) \geq 1$ for all $t \in [0, c)$. First, due to the initial condition $p(0) = 1$ and continuity of $p$, there exists $\epsilon_1 > 0$ such that $p(t) > 0$ for all $t \in [0, \epsilon_1) =: I_1$. This implies that $\dot{q}(t) = p(t)^{L-1} > 0$ for $t \in I_1 \setminus \{0\}$, so $q$ is strictly increasing on $I_1$. Since $q(0) = 0$, we have $q(t) > 0$ for $t \in I_1 \setminus \{0\}$, which then implies that $\dot{p}(t) = p(t)^{L-2}q(t) > 0$. Therefore, $p$ is

also strictly increasing on $I_1$; this then means $p(t) \geq 1$ for $t \in [0, \epsilon_1]$ because $p(0) = 1$. Now, due to $p(\epsilon_1) \geq 1$ and continuity of $p$, there exists $\epsilon_2 > \epsilon_1$ such that $p(t) > 0$ for all $t \in [\epsilon_1, \epsilon_2) =: I_2$. Using the argument above for $I_2$ results in $p(t) \geq 1$ for $t \in [0, \epsilon_2]$. Repeating this until the end of the domain, we can show that $p(t) \geq 1$ holds for all $t \in [0, c)$. By $p \geq 1$, we have $\dot{q} = p^{L-1} \geq 1$ on $[0, c)$, so $q$ is strictly increasing on $[0, c)$. Also, $q(t) > 0$ on $(0, c)$, so $\dot{p} = p^{L-2}q > 0$ on $(0, c)$ and $p$ is also strictly increasing on $[0, c)$. This proves that $h$ is strictly increasing on $[0, c)$, and also on $(-c, c)$ by oddity of $h$.

Finally, it is left to show $\lim_{t \uparrow c} h(t) = \infty$ and $\lim_{t \downarrow -c} h(t) = -\infty$. If $c < \infty$, then this together with monotonicity implies that the limits hold. To see why, suppose $c < \infty$ and $\lim_{t \uparrow c} h(t) < \infty$. Then, $p$ and $q$ can be extended beyond $t \geq c$, which contradicts the fact that $(-c, c)$ is the maximal interval of existence of the solution. Next, consider the case $c = \infty$. From $p(t) \geq 1$, we have $\dot{q}(t) \geq 1$ for $t \geq 0$. This implies that $q(t) \geq t$ for $t \geq 0$. Now, $\dot{p}(t) \geq p(t)^{L-2}q(t) \geq t$, which gives $p(t) \geq \frac{t^2}{2} + 1$ for $t \geq 0$. Therefore, we have

$$\lim_{t \to \infty} h(t) = \lim_{t \to \infty} p(t)^{L-1} q(t) \geq \lim_{t \to \infty} \left( \frac{t^2}{2} + 1 \right)^{L-1} t = \infty,$$

hence finishing the proof. □

## F.2 PROOF OF THEOREM 5

### F.2.1 CONVERGENCE OF LOSS TO ZERO

We first show that given the conditions on initialization, the training loss $\mathcal{L}(\boldsymbol{\Theta}(t))$ converges to zero. Recall from Section 2.1 that

$$\dot{\boldsymbol{v}}_l = -\nabla_{\boldsymbol{v}_l} \mathcal{L}(\boldsymbol{\Theta}) = \mathsf{M}(-\boldsymbol{X}^T \boldsymbol{r}) \circ (\boldsymbol{v}_1, \ldots, \boldsymbol{v}_{l-1}, \boldsymbol{I}_{k_l}, \boldsymbol{v}_{l+1}, \ldots, \boldsymbol{v}_L).$$

Applying the structure (9) in Assumption 1, we get

$$\dot{\boldsymbol{v}}_l = \mathsf{M}(-\boldsymbol{X}^T \boldsymbol{r}) \circ (\boldsymbol{v}_1, \ldots, \boldsymbol{v}_{l-1}, \boldsymbol{I}_{k_l}, \boldsymbol{v}_{l+1}, \ldots, \boldsymbol{v}_L)$$

$$= -\sum_{j=1}^m [\boldsymbol{S}\boldsymbol{X}^T \boldsymbol{r}]_j (\boldsymbol{v}_1^T [\boldsymbol{U}_1]_{\cdot,j} \otimes \cdots \otimes \boldsymbol{v}_{l-1}^T [\boldsymbol{U}_{l-1}]_{\cdot,j} \otimes [\boldsymbol{U}_l]_{\cdot,j} \otimes \boldsymbol{v}_{l+1}^T [\boldsymbol{U}_{l+1}]_{\cdot,j} \otimes \cdots \otimes \boldsymbol{v}_L^T [\boldsymbol{U}_L]_{\cdot,j})$$

$$= -\sum_{j=1}^m [\boldsymbol{S}\boldsymbol{X}^T \boldsymbol{r}]_j \left( \prod_{k \neq l} [\boldsymbol{U}_k^T \boldsymbol{v}_k]_j \right) [\boldsymbol{U}_l]_{\cdot,j}.$$

Left-multiplying $\boldsymbol{U}_l^T$ to both sides, we get

$$\boldsymbol{U}_l^T \dot{\boldsymbol{v}}_l = -\boldsymbol{S}\boldsymbol{X}^T \boldsymbol{r} \odot \prod_{k \neq l}^{\odot} \boldsymbol{U}_k^T \boldsymbol{v}_k, \tag{57}$$

where $\prod^{\odot}$ denotes the product using entry-wise multiplication $\odot$.

Now consider the rate of growth for the second power of the $j$-th component of $\boldsymbol{U}_l^T \boldsymbol{v}_l$:

$$\frac{d}{dt} [\boldsymbol{U}_l^T \boldsymbol{v}_l]_j^2 = 2[\boldsymbol{U}_l^T \dot{\boldsymbol{v}}_l]_j [\boldsymbol{U}_l^T \boldsymbol{v}_l]_j = -2[\boldsymbol{S}\boldsymbol{X}^T \boldsymbol{r}]_j \prod_{k=1}^L [\boldsymbol{U}_k^T \boldsymbol{v}_k]_j = \frac{d}{dt} |[\boldsymbol{U}_{l'}^T \boldsymbol{v}_{l'}]_j|^2$$

for any $l' \in [L]$. Thus, for any $j \in [m]$, the second power of the $j$-th components in $\boldsymbol{U}_l^T \boldsymbol{v}_l$ grow at the same rate for each layer $l \in [L]$. This means that the gap between any two different layers stays constant for all $t \geq 0$. Combining this with our conditions on initial directions, we have

$$[\boldsymbol{U}_l^T \boldsymbol{v}_l(t)]_j^2 - [\boldsymbol{U}_L^T \boldsymbol{v}_L(t)]_j^2 = [\boldsymbol{U}_l^T \boldsymbol{v}_l(0)]_j^2 - [\boldsymbol{U}_L^T \boldsymbol{v}_L(0)]_j^2 = \alpha^2 [\bar{\boldsymbol{\eta}}]_j^2 \geq \alpha^2 \lambda,$$

for any $j \in [m]$, $l \in [L-1]$, and $t \geq 0$. This inequality also implies

$$[\boldsymbol{U}_l^T \boldsymbol{v}_l(t)]_j^2 \geq [\boldsymbol{U}_L^T \boldsymbol{v}_L(t)]_j^2 + \alpha^2 \lambda \geq \alpha^2 \lambda. \tag{58}$$

Let us now consider the time derivative of $\mathcal{L}(\boldsymbol{\Theta}(t))$. We have the following chain of upper bounds on the time derivative:

$$
\begin{aligned}
\frac{d}{dt}\mathcal{L}(\boldsymbol{\Theta}(t)) &= \nabla_{\boldsymbol{\Theta}}\mathcal{L}(\boldsymbol{\Theta}(t))^T\dot{\boldsymbol{\Theta}}(t) = -\|\nabla_{\boldsymbol{\Theta}}\mathcal{L}(\boldsymbol{\Theta}(t))\|_2^2 \\
&\leq -\|\nabla_{\boldsymbol{v}_L}\mathcal{L}(\boldsymbol{\Theta}(t))\|_2^2 = -\|\dot{\boldsymbol{v}}_L(t)\|_2^2 \\
&\overset{(a)}{\leq} -\|\boldsymbol{U}_L^T\dot{\boldsymbol{v}}_L(t)\|_2^2 \overset{(b)}{=} -\left\|\boldsymbol{S}\boldsymbol{X}^T\boldsymbol{r}(t)\odot\prod_{k\neq L}^{\odot}\boldsymbol{U}_k^T\boldsymbol{v}_k(t)\right\|_2^2 \\
&= -\sum_{j=1}^m [\boldsymbol{S}\boldsymbol{X}^T\boldsymbol{r}(t)]_j^2\prod_{k\neq L}[\boldsymbol{U}_k^T\boldsymbol{v}_k(t)]_j^2 \\
&\overset{(c)}{\leq} -\alpha^{2L-2}\lambda^{L-1}\sum_{j=1}^m [\boldsymbol{S}\boldsymbol{X}^T\boldsymbol{r}(t)]_j^2 \\
&= -\alpha^{2L-2}\lambda^{L-1}\|\boldsymbol{S}\boldsymbol{X}^T\boldsymbol{r}(t)\|_2^2 \\
&\overset{(d)}{\leq} -\alpha^{2L-2}\lambda^{L-1}s_{\min}(\boldsymbol{S})^2 s_{\min}(\boldsymbol{X})^2\|\boldsymbol{r}(t)\|_2^2, \\
&= -2\alpha^{2L-2}\lambda^{L-1}s_{\min}(\boldsymbol{S})^2 s_{\min}(\boldsymbol{X})^2\mathcal{L}(\boldsymbol{\Theta}(t)), \quad\quad (59)
\end{aligned}
$$

where (a) used the fact that $\|\dot{\boldsymbol{v}}_L(t)\|_2^2 \geq \|\boldsymbol{U}_L\boldsymbol{U}_L^T\dot{\boldsymbol{v}}_L(t)\|_2^2$ because it is a projection onto a subspace, and $\|\boldsymbol{U}_L\boldsymbol{U}_L^T\dot{\boldsymbol{v}}_L(t)\|_2^2 = \|\boldsymbol{U}_L^T\dot{\boldsymbol{v}}_L(t)\|_2^2$ because $\boldsymbol{U}_L^T\boldsymbol{U}_L = \boldsymbol{I}_{k_L}$; (b) is due to (57); (c) is due to (58); and (d) used the fact that $\boldsymbol{S}\in\mathbb{R}^{m\times d}$ and $\boldsymbol{X}^T\in\mathbb{R}^{d\times n}$ are matrices that have full column rank, so for any $\boldsymbol{z}\in\mathbb{C}^n$, we can use $\|\boldsymbol{S}\boldsymbol{X}^T\boldsymbol{z}\|_2 \geq s_{\min}(\boldsymbol{S})s_{\min}(\boldsymbol{X})\|\boldsymbol{z}\|_2$ where $s_{\min}(\cdot)$ denotes the minimum singular value of a matrix.

From (59), we get

$$
\mathcal{L}(\boldsymbol{\Theta}(t)) \leq \mathcal{L}(\boldsymbol{\Theta}(0))\exp(-2\alpha^{2L-2}\lambda^{L-1}s_{\min}(\boldsymbol{S})^2 s_{\min}(\boldsymbol{X})^2 t), \quad\quad (60)
$$

so that $\mathcal{L}(\boldsymbol{\Theta}(t)) \to 0$ as $t\to\infty$.

### F.2.2 Characterizing the limit point

Now, we move on to characterize the limit points of the gradient flow. First, by defining a "transformed" version of the parameters $\boldsymbol{\eta}_l(t) := \boldsymbol{U}_l^T\boldsymbol{v}_l(t)$ and using (57), one can define an equivalent system of ODEs:

$$
\begin{aligned}
\dot{\boldsymbol{\eta}}_l &= -\boldsymbol{S}\boldsymbol{X}^T\boldsymbol{r}\odot\prod_{k\neq l}^{\odot}\boldsymbol{\eta}_k \text{ for } l\in[L], \\
\boldsymbol{\eta}_l(0) &= \alpha\bar{\boldsymbol{\eta}} \text{ for } l\in[L-1], \quad \boldsymbol{\eta}_L(0) = \boldsymbol{0}.
\end{aligned} \quad\quad (61)
$$

Using Lemma 4, it is straightforward to verify that the solution to (61) has the following form. For odd $L$, we have

$$
\begin{aligned}
\boldsymbol{\eta}_l(t) &= \alpha\bar{\boldsymbol{\eta}}\odot p_L\left(-\alpha^{L-2}|\bar{\boldsymbol{\eta}}|^{\odot L-2}\odot\boldsymbol{S}\boldsymbol{X}^T\int_0^t\boldsymbol{r}(\tau)d\tau\right) \text{ for } l\in[L-1], \\
\boldsymbol{\eta}_L(t) &= \alpha|\bar{\boldsymbol{\eta}}|\odot q_L\left(-\alpha^{L-2}|\bar{\boldsymbol{\eta}}|^{\odot L-2}\odot\boldsymbol{S}\boldsymbol{X}^T\int_0^t\boldsymbol{r}(\tau)d\tau\right).
\end{aligned} \quad\quad (62)
$$

Similarly, for even $L$, the solution for (61) satisfies

$$
\begin{aligned}
\boldsymbol{\eta}_l(t) &= \alpha\bar{\boldsymbol{\eta}}\odot p_L\left(-\alpha^{L-2}\bar{\boldsymbol{\eta}}^{\odot L-2}\odot\boldsymbol{S}\boldsymbol{X}^T\int_0^t\boldsymbol{r}(\tau)d\tau\right) \text{ for } l\in[L-1], \\
\boldsymbol{\eta}_L(t) &= \alpha\bar{\boldsymbol{\eta}}\odot q_L\left(-\alpha^{L-2}\bar{\boldsymbol{\eta}}^{\odot L-2}\odot\boldsymbol{S}\boldsymbol{X}^T\int_0^t\boldsymbol{r}(\tau)d\tau\right).
\end{aligned} \quad\quad (63)
$$

Now that we know how the solutions $\boldsymbol{\eta}_l$ look like, let us see how these relate to the linear coefficients of the network. By Assumption 1, we have

$$
f(\boldsymbol{x};\boldsymbol{\Theta}) = \mathbf{M}(\boldsymbol{x})\circ(\boldsymbol{v}_1,\ldots,\boldsymbol{v}_L) = \sum_{j=1}^m [\boldsymbol{S}\boldsymbol{x}]_j\prod_{l=1}^L[\boldsymbol{U}_l^T\boldsymbol{v}_l]_j
$$

$$= \left[ \sum_{j=1}^{m} \left( \prod_{l=1}^{L} [\boldsymbol{\eta}_l]_j \right) [\boldsymbol{S}]_{j,\cdot} \right] \boldsymbol{x} = \boldsymbol{x}^T \boldsymbol{S}^T \left( \prod_{l \in [L]}^{\odot} \boldsymbol{\eta}_l \right) = \boldsymbol{x}^T \boldsymbol{S}^T \boldsymbol{\rho}.$$

Here, we defined $\boldsymbol{\rho} := \prod_{l \in [L]}^{\odot} \boldsymbol{\eta}_l \in \mathbb{R}^m$. Therefore, the linear coefficients of the network can be written as $\boldsymbol{\beta}(\boldsymbol{\Theta}(t)) = \boldsymbol{S}^T \boldsymbol{\rho}(t)$. From the solutions (62) and (63), we can write

$$\boldsymbol{\rho}(t) = \prod_{i=1}^{L} \boldsymbol{\eta}_l(t) = \alpha^L |\bar{\boldsymbol{\eta}}|^{\odot L} \odot h_L \left( -\alpha^{L-2} |\bar{\boldsymbol{\eta}}|^{\odot L-2} \odot \boldsymbol{S} \boldsymbol{X}^T \int_0^t \boldsymbol{r}(\tau) d\tau \right),$$

where $h_L := p_L^{L-1} q_L$, defined in Lemma 4. By the convergence of the loss to zero (60), we have $\lim_{t \to \infty} \boldsymbol{X} \boldsymbol{\beta}(\boldsymbol{\Theta}(t)) = \boldsymbol{y}$. Therefore,

$$\boldsymbol{X} \boldsymbol{S}^T \underbrace{\left( \alpha^L |\bar{\boldsymbol{\eta}}|^{\odot L} \odot h_L \left( -\alpha^{L-2} |\bar{\boldsymbol{\eta}}|^{\odot L-2} \odot \boldsymbol{S} \boldsymbol{X}^T \int_0^{\infty} \boldsymbol{r}(\tau) d\tau \right) \right)}_{=:\boldsymbol{\rho}^{\infty}} = \boldsymbol{y}. \tag{64}$$

Next, we will show that $\boldsymbol{\rho}^{\infty}$ is in fact the solution of the following optimization problem

$$\underset{\boldsymbol{\rho} \in \mathbb{R}^m}{\text{minimize}} \quad Q_{L,\alpha,\bar{\boldsymbol{\eta}}}(\boldsymbol{\rho}) \quad \text{subject to} \quad \boldsymbol{X} \boldsymbol{S}^T \boldsymbol{\rho} = \boldsymbol{y}, \tag{65}$$

where $Q_{L,\alpha,\bar{\boldsymbol{\eta}}} : \mathbb{R}^m \to \mathbb{R}$ is a norm-like function defined using $H_L(t) := \int_0^t h_L^{-1}(\tau) d\tau$:

$$Q_{L,\alpha,\bar{\boldsymbol{\eta}}}(\boldsymbol{\rho}) = \alpha^2 \sum_{j=1}^{m} [\bar{\boldsymbol{\eta}}]_j^2 H_L \left( \frac{[\boldsymbol{\rho}]_j}{\alpha^L |[\bar{\boldsymbol{\eta}}]_j|^L} \right).$$

Note that the KKT conditions for (65) are

$$\boldsymbol{X} \boldsymbol{S}^T \boldsymbol{\rho} = \boldsymbol{y}, \quad \nabla_{\boldsymbol{\rho}} Q_{L,\alpha,\bar{\boldsymbol{\eta}}}(\boldsymbol{\rho}) = \boldsymbol{S} \boldsymbol{X}^T \boldsymbol{\nu},$$

for some $\boldsymbol{\nu} \in \mathbb{R}^n$. It is clear from (64) that $\boldsymbol{\rho}^{\infty}$ satisfies the first condition (primal feasibility), so let us check the other one. Through a straightforward calculation, we get

$$\nabla_{\boldsymbol{\rho}} Q_{L,\alpha,\bar{\boldsymbol{\eta}}}(\boldsymbol{\rho}) = \alpha^{2-L} |\bar{\boldsymbol{\eta}}|^{\odot 2-L} \odot h_L^{-1} \left( \alpha^{-L} |\bar{\boldsymbol{\eta}}|^{\odot(-L)} \odot \boldsymbol{\rho} \right).$$

Equating this with $\boldsymbol{S} \boldsymbol{X}^T \boldsymbol{\nu}$ gives

$$\alpha^{2-L} |\bar{\boldsymbol{\eta}}|^{\odot 2-L} \odot h_L^{-1} \left( \alpha^{-L} |\bar{\boldsymbol{\eta}}|^{\odot(-L)} \odot \boldsymbol{\rho} \right) = \boldsymbol{S} \boldsymbol{X}^T \boldsymbol{\nu}$$

$$\Leftrightarrow h_L^{-1} \left( \alpha^{-L} |\bar{\boldsymbol{\eta}}|^{\odot(-L)} \odot \boldsymbol{\rho} \right) = \alpha^{L-2} |\bar{\boldsymbol{\eta}}|^{\odot L-2} \odot \boldsymbol{S} \boldsymbol{X}^T \boldsymbol{\nu}$$

$$\Leftrightarrow \boldsymbol{\rho} = \alpha^L |\bar{\boldsymbol{\eta}}|^{\odot L} \odot h_L \left( \alpha^{L-2} |\bar{\boldsymbol{\eta}}|^{\odot L-2} \odot \boldsymbol{S} \boldsymbol{X}^T \boldsymbol{\nu} \right).$$

Hence, by setting $\boldsymbol{\nu} = -\int_0^{\infty} \boldsymbol{r}(\tau) d\tau$, $\boldsymbol{\rho}^{\infty}$ satisfies this condition as well. Also, if $\boldsymbol{S}$ is invertible, we can substitute $\boldsymbol{\rho} = \boldsymbol{S}^{-T} \boldsymbol{z}$ to (65) to get the last statement of the theorem. This finishes the proof.

### F.3 PROOF OF COROLLARY 5

The proof is a direct consequence of the fact that Assumption 1 holds with $\boldsymbol{S} = \boldsymbol{U}_1 = \cdots = \boldsymbol{U}_L = \boldsymbol{I}_d$ for linear diagonal networks. Hence, the proof is the same as Corollary 2, proved in Appendix D.2.

### F.4 PROOF OF COROLLARY 6

We start by showing the DFT of a real and even vector is also real and even. Suppose that $\boldsymbol{x} \in \mathbb{R}^d$ is real and even. First,

$$[\boldsymbol{F} \boldsymbol{x}]_j = \frac{1}{\sqrt{d}} \sum_{k=1}^{d} [\boldsymbol{x}]_k \exp \left( -\frac{\sqrt{-1} \cdot 2\pi(j-1)(k-1)}{d} \right)$$

$$= \frac{1}{\sqrt{d}} \sum_{k=1}^{d} [\boldsymbol{x}]_k \cos\left(-\frac{2\pi(j-1)(k-1)}{d}\right) + \frac{\sqrt{-1}}{\sqrt{d}} \sum_{k=1}^{d} [\boldsymbol{x}]_k \sin\left(-\frac{2\pi(j-1)(k-1)}{d}\right)$$

$$= \frac{1}{\sqrt{d}} \sum_{k=1}^{d} [\boldsymbol{x}]_k \cos\left(-\frac{2\pi(j-1)(k-1)}{d}\right) \in \mathbb{R},$$

for all $j \in [d]$. To prove that $\boldsymbol{Fx}$ is even, for $j = 0, \ldots, \lfloor\frac{d-3}{2}\rfloor$, we have

$$\begin{aligned}
[\boldsymbol{Fx}]_{j+2} &= \frac{1}{\sqrt{d}} \sum_{k=1}^{d} [\boldsymbol{x}]_k \cos\left(-\frac{2\pi(j+1)(k-1)}{d}\right) \\
&= \frac{1}{\sqrt{d}} \sum_{k=1}^{d} [\boldsymbol{x}]_k \cos\left(2\pi(k-1) - \frac{2\pi(j+1)(k-1)}{d}\right) \\
&= \frac{1}{\sqrt{d}} \sum_{k=1}^{d} [\boldsymbol{x}]_k \cos\left(\frac{2\pi(d-j-1)(k-1)}{d}\right) \\
&= \frac{1}{\sqrt{d}} \sum_{k=1}^{d} [\boldsymbol{x}]_k \cos\left(-\frac{2\pi(d-j-1)(k-1)}{d}\right) \\
&= [\boldsymbol{Fx}]_{d-j}.
\end{aligned}$$

It is proved in Appendix D.3 that linear full-length convolutional networks ($k_1 = \cdots = k_L = d$) satisfy Assumption 1 with $\boldsymbol{S} = d^{\frac{L-1}{2}}\boldsymbol{F}$ and $\boldsymbol{U}_1 = \cdots = \boldsymbol{U}_L = \boldsymbol{F}^*$, where $\boldsymbol{F} \in \mathbb{C}^{d \times d}$ is the matrix of discrete Fourier transform basis $[\boldsymbol{F}]_{j,k} = \frac{1}{\sqrt{d}}\exp(-\frac{\sqrt{-1}\cdot 2\pi(j-1)(k-1)}{d})$ and $\boldsymbol{F}^*$ is the complex conjugate of $\boldsymbol{F}$.

The proof of convergence of loss to zero in Appendix F.2.1 is written for real matrices $\boldsymbol{S}, \boldsymbol{U}_1, \ldots, \boldsymbol{U}_L$, but we can actually apply the same argument as in Appendix D.1.1 and prove that the loss converges to zero, even in the case where $\boldsymbol{S}, \boldsymbol{U}_1, \ldots, \boldsymbol{U}_L$ are complex.

Next, since $\boldsymbol{U}_l$'s are complex, we can write the system of ODE as (see (20) for its derivation)

$$\boldsymbol{F}\dot{\boldsymbol{w}}_l = -d^{\frac{L-1}{2}}\boldsymbol{FX}^T\boldsymbol{r} \odot \prod_{k \neq l}^{\odot} \boldsymbol{F}^*\boldsymbol{w}_k, \tag{66}$$

Since all data points $\boldsymbol{x}_i$ and initialization $\boldsymbol{w}_l(0)$ are real and even, we have that $\boldsymbol{FX}^T\boldsymbol{r}$ is real and even, and $\boldsymbol{F}^*\boldsymbol{w}_l(0) = \boldsymbol{Fw}_l(0)$'s are real and even. By (66), we see that the time derivatives of $\boldsymbol{Fw}_l$ are also real and even. Thus, the parameters $\boldsymbol{w}_l(t)$ are all real and even for all $t \geq 0$. From this observation, we can define $\boldsymbol{\eta}_l(t) := \boldsymbol{Fw}_l(t)$, $\bar{\boldsymbol{\eta}} := \boldsymbol{F\bar{w}}$, and $\boldsymbol{S} := d^{\frac{L-1}{2}}\mathrm{Re}(\boldsymbol{F})$, which are all real by the even symmetry. Then, starting from (61), the proof goes through.

### F.5  PROOF OF COROLLARY 7

Since the sensor matrices $\boldsymbol{A}_1, \ldots, \boldsymbol{A}_n$ commute, they are simultaneously diagonalizable with a real unitary matrix $\boldsymbol{U} \in \mathbb{R}^{d \times d}$, i.e., $\boldsymbol{U}^T\boldsymbol{A}_i\boldsymbol{U}$'s are diagonal matrices. From the deep matrix sensing problem (13), we can compute $\nabla_{\boldsymbol{W}_l}\mathcal{L}_{\mathrm{ms}}$, which gives the gradient flow dynamics of $\boldsymbol{W}_l$.

$$\dot{\boldsymbol{W}}_l = -\nabla_{\boldsymbol{W}_l}\mathcal{L}_{\mathrm{ms}} = -\boldsymbol{W}_{l-1}^T \cdots \boldsymbol{W}_1^T \left(\sum_{i=1}^{n} r_i \boldsymbol{A}_i\right) \boldsymbol{W}_L^T \cdots \boldsymbol{W}_{l+1}^T,$$

where $r_i = \langle \boldsymbol{A}_i, \boldsymbol{W}_1 \cdots \boldsymbol{W}_L \rangle - y_i$ is the residual for the $i$-th sensor matrix. If we left-multiply $\boldsymbol{U}^T$ and right-multiply $\boldsymbol{U}$ to both sides, we get

$$\boldsymbol{U}^T\dot{\boldsymbol{W}}_l\boldsymbol{U} = -\boldsymbol{U}^T\boldsymbol{W}_{l-1}^T\boldsymbol{U} \cdots \boldsymbol{U}^T\boldsymbol{W}_1^T\boldsymbol{U}\left(\sum_{i=1}^{n} r_i \boldsymbol{U}^T\boldsymbol{A}_i\boldsymbol{U}\right)\boldsymbol{U}^T\boldsymbol{W}_L^T\boldsymbol{U} \cdots \boldsymbol{U}^T\boldsymbol{W}_{l+1}^T\boldsymbol{U}. \tag{67}$$

If $\boldsymbol{U}^T\boldsymbol{W}_k^T\boldsymbol{U}$ is a diagonal matrix for all $k \neq l$, then $\boldsymbol{U}^T\dot{\boldsymbol{W}}_l\boldsymbol{U}$ is also a diagonal matrix. Note also that, since $\boldsymbol{W}_l(0) = \alpha\boldsymbol{I}_d = \alpha\boldsymbol{UU}^T$ for $l \in [L-1]$, the product $\boldsymbol{U}^T\boldsymbol{W}_l\boldsymbol{U}$ is a diagonal matrix at initialization. These observations imply that $\boldsymbol{W}_l(t)$'s are all diagonalizable with $\boldsymbol{U}$ for all $t \geq 0$.

Now, define $\boldsymbol{v}_l(t) = \mathrm{eig}(\boldsymbol{W}_l(t))$, i.e., $\boldsymbol{U}^T \boldsymbol{W}_l \boldsymbol{U} = \mathrm{diag}(\boldsymbol{v}_l)$. Also, let $\boldsymbol{x}_i = \mathrm{eig}(\boldsymbol{A}_i)$. Then, (67) can be written as

$$\dot{\boldsymbol{v}}_l = -\Big(\sum_{i=1}^n r_i \boldsymbol{x}_i\Big) \odot \prod_{k \neq l}^{\odot} \boldsymbol{v}_k.$$

Therefore, this is equivalent to the regression problem with linear diagonal networks, initialized at $\boldsymbol{v}_l(0) = \alpha \mathbf{1}$ for $l \in [L-1]$ and $\boldsymbol{v}_L(0) = \boldsymbol{0}$. Given this equivalence, Corollary 7 can be implied from Corollary 5.

## G  PROOF OF THEOREM 6

### G.1  CONVERGENCE OF LOSS TO ZERO

We first show that given the conditions on initialization, the training loss $\mathcal{L}(\boldsymbol{\Theta}(t))$ converges to zero. Since $L = 2$ and $\mathbf{M}(\boldsymbol{x}) = \boldsymbol{U}_1 \mathrm{diag}(\boldsymbol{s}) \boldsymbol{U}_2^T$, we can write the gradient flow dynamics from Section 2.1 as

$$
\begin{aligned}
\dot{\boldsymbol{v}}_1 &= -\mathbf{M}(\boldsymbol{X}^T \boldsymbol{r}) \circ (\boldsymbol{I}_{k_1}, \boldsymbol{v}_2) = -r \boldsymbol{U}_1 \mathrm{diag}(\boldsymbol{s}) \boldsymbol{U}_2^T \boldsymbol{v}_2, \\
\dot{\boldsymbol{v}}_2 &= -\mathbf{M}(\boldsymbol{X}^T \boldsymbol{r}) \circ (\boldsymbol{v}_1, \boldsymbol{I}_{k_2}) = -r \boldsymbol{U}_2 \mathrm{diag}(\boldsymbol{s}) \boldsymbol{U}_1^T \boldsymbol{v}_1,
\end{aligned}
\tag{68}
$$

where $r(t) = f(\boldsymbol{x}; \boldsymbol{\Theta}(t)) - y$ is the residual of the data point $(\boldsymbol{x}, y)$. From (68) we get

$$\boldsymbol{U}_l^T \dot{\boldsymbol{v}}_1 = -r \boldsymbol{s} \odot \boldsymbol{U}_2^T \boldsymbol{v}_2, \ \ \boldsymbol{U}_2^T \dot{\boldsymbol{v}}_2 = -r \boldsymbol{s} \odot \boldsymbol{U}_1^T \boldsymbol{v}_1. \tag{69}$$

Now consider the rate of growth for the $j$-th component of $\boldsymbol{U}_1^T \boldsymbol{v}_1$ squared:

$$\frac{d}{dt}[\boldsymbol{U}_1^T \boldsymbol{v}_1]_j^2 = 2[\boldsymbol{U}_1^T \boldsymbol{v}_1]_j [\boldsymbol{U}_1^T \dot{\boldsymbol{v}}_1]_j = -2r[\boldsymbol{s}]_j [\boldsymbol{U}_1^T \boldsymbol{v}_1]_j [\boldsymbol{U}_2^T \boldsymbol{v}_2]_j = \frac{d}{dt}[\boldsymbol{U}_2^T \boldsymbol{v}_2]_j^2.$$

So for any $j \in [m]$, $[\boldsymbol{U}_1^T \boldsymbol{v}_1]_j^2$ and $[\boldsymbol{U}_2^T \boldsymbol{v}_2]_j^2$ grow at the same rate. This means that the gap between the two layers stays constant for all $t \geq 0$. Combining this with our conditions on initial directions,

$$
\begin{aligned}
[\boldsymbol{U}_1^T \boldsymbol{v}_1(t)]_j^2 - [\boldsymbol{U}_2^T \boldsymbol{v}_2(t)]_j^2 &= [\boldsymbol{U}_1^T \boldsymbol{v}_1(0)]_j^2 - [\boldsymbol{U}_2^T \boldsymbol{v}_2(0)]_j^2 \\
&= \alpha^2 [\boldsymbol{U}_1^T \bar{\boldsymbol{v}}_1]_j^2 - \alpha^2 [\boldsymbol{U}_2^T \bar{\boldsymbol{v}}_2]_j^2 \geq \alpha^2 \lambda,
\end{aligned}
$$

for any $j \in [m]$ and $t \geq 0$. This inequality implies

$$[\boldsymbol{U}_1^T \boldsymbol{v}_1(t)]_j^2 \geq [\boldsymbol{U}_2^T \boldsymbol{v}_2(t)]_j^2 + \alpha^2 \lambda \geq \alpha^2 \lambda. \tag{70}$$

Let us now consider the time derivative of $\mathcal{L}(\boldsymbol{\Theta}(t))$. We have the following chain of upper bounds on the time derivative:

$$
\begin{aligned}
\frac{d}{dt} \mathcal{L}(\boldsymbol{\Theta}(t)) &= \nabla_{\boldsymbol{\Theta}} \mathcal{L}(\boldsymbol{\Theta}(t))^T \dot{\boldsymbol{\Theta}}(t) = -\|\nabla_{\boldsymbol{\Theta}} \mathcal{L}(\boldsymbol{\Theta}(t))\|_2^2 \\
&\leq -\|\nabla_{\boldsymbol{v}_2} \mathcal{L}(\boldsymbol{\Theta}(t))\|_2^2 = -\|\dot{\boldsymbol{v}}_2(t)\|_2^2 \\
&\overset{(a)}{\leq} -\|\boldsymbol{U}_2^T \dot{\boldsymbol{v}}_2(t)\|_2^2 \overset{(b)}{=} -r(t)^2 \|\boldsymbol{s} \odot \boldsymbol{U}_1^T \boldsymbol{v}_1(t)\|_2^2 \\
&= -r(t)^2 \sum_{j=1}^m [\boldsymbol{s}]_j^2 [\boldsymbol{U}_1^T \boldsymbol{v}_1(t)]_j^2 \\
&\overset{(c)}{\leq} -\alpha^2 \lambda r(t)^2 \sum_{j=1}^m [\boldsymbol{s}]_j^2 \\
&= -2\alpha^2 \lambda \|\boldsymbol{s}\|_2^2 \mathcal{L}(\boldsymbol{\Theta}(t)),
\end{aligned}
$$

where (a) used the fact that $\|\dot{\boldsymbol{v}}_2(t)\|_2^2 \geq \|\boldsymbol{U}_2 \boldsymbol{U}_2^T \dot{\boldsymbol{v}}_2(t)\|_2^2$ because it is a projection onto a subspace, and $\|\boldsymbol{U}_2 \boldsymbol{U}_2^T \dot{\boldsymbol{v}}_L(t)\|_2^2 = \|\boldsymbol{U}_2^T \dot{\boldsymbol{v}}_2(t)\|_2^2$ because $\boldsymbol{U}_2^T \boldsymbol{U}_2 = \boldsymbol{I}_{k_2}$; (b) is due to (69); (c) is due to (70). From this, we get

$$\mathcal{L}(\boldsymbol{\Theta}(t)) \leq \mathcal{L}(\boldsymbol{\Theta}(0)) \exp(-2\alpha^2 \lambda \|\boldsymbol{s}\|_2^2 t). \tag{71}$$

Therefore, $\mathcal{L}(\boldsymbol{\Theta}(t)) \to 0$ as $t \to \infty$.

### G.2 CHARACTERIZING THE LIMIT POINT

Now, we move on to characterize the limit points of the gradient flow. First, note that any changes made in $\boldsymbol{v}_l$ over time are in the subspace spanned by the columns of $\boldsymbol{U}_l$. Therefore, any component in the initialization $\boldsymbol{v}_l(0) = \alpha \bar{\boldsymbol{v}}_l$ that is orthogonal to the column space of $\boldsymbol{U}_l$ stays constant.

So, we can focus on the evolution of $\boldsymbol{v}_l$ in the column space of $\boldsymbol{U}_l$; this can be done by defining a "transformed" version of the parameters $\boldsymbol{\eta}_l(t) := \boldsymbol{U}_l^T \boldsymbol{v}_l(t)$ and using (69), one can define an equivalent system of ODEs:

$$\dot{\boldsymbol{\eta}}_1 = -r\boldsymbol{s} \odot \boldsymbol{\eta}_2, \quad \dot{\boldsymbol{\eta}}_2 = -r\boldsymbol{s} \odot \boldsymbol{\eta}_1,$$
$$\boldsymbol{\eta}_1(0) = \alpha \bar{\boldsymbol{\eta}}_1, \quad \boldsymbol{\eta}_2(0) = \alpha \bar{\boldsymbol{\eta}}_2, \tag{72}$$

where $\bar{\boldsymbol{\eta}}_1 := \boldsymbol{U}_1^T \bar{\boldsymbol{v}}_1$, $\bar{\boldsymbol{\eta}}_2 := \boldsymbol{U}_2^T \bar{\boldsymbol{v}}_2$. It is straightforward to verify that the solution to (72) has the following form.

$$\boldsymbol{\eta}_1(t) = \alpha \bar{\boldsymbol{\eta}}_1 \odot \cosh\left(-\boldsymbol{s} \int_0^t r(\tau)d\tau\right) + \alpha \bar{\boldsymbol{\eta}}_2 \odot \sinh\left(-\boldsymbol{s} \int_0^t r(\tau)d\tau\right),$$
$$\boldsymbol{\eta}_2(t) = \alpha \bar{\boldsymbol{\eta}}_1 \odot \sinh\left(-\boldsymbol{s} \int_0^t r(\tau)d\tau\right) + \alpha \bar{\boldsymbol{\eta}}_2 \odot \cosh\left(-\boldsymbol{s} \int_0^t r(\tau)d\tau\right). \tag{73}$$

By the convergence of the loss to zero (71), we have $\lim_{t\to\infty} f(\boldsymbol{x}; \boldsymbol{\Theta}(t)) = y$. Note that $f(\boldsymbol{x}; \boldsymbol{\Theta}(t))$ can be written as

$$f(\boldsymbol{x}; \boldsymbol{\Theta}(t)) = \mathbf{M}(\boldsymbol{x}) \circ (\boldsymbol{v}_1(t), \boldsymbol{v}_2(t)) = \boldsymbol{v}_1(t)^T \mathbf{M}(\boldsymbol{x}) \boldsymbol{v}_2(t)$$
$$= \boldsymbol{v}_1(t)^T \boldsymbol{U}_1 \operatorname{diag}(\boldsymbol{s}) \boldsymbol{U}_2^T \boldsymbol{v}_2(t) = \boldsymbol{s}^T (\boldsymbol{\eta}_1(t) \odot \boldsymbol{\eta}_2(t)).$$

Therefore,

$$\lim_{t\to\infty} f(\boldsymbol{x}; \boldsymbol{\Theta}(t)) = \lim_{t\to\infty} \boldsymbol{s}^T (\boldsymbol{\eta}_1(t) \odot \boldsymbol{\eta}_2(t))$$
$$= \alpha^2 \boldsymbol{s}^T \left[ (\bar{\boldsymbol{\eta}}_1^{\odot 2} + \bar{\boldsymbol{\eta}}_2^{\odot 2}) \odot \cosh\left(-\boldsymbol{s} \int_0^\infty r(\tau)d\tau\right) \odot \sinh\left(-\boldsymbol{s} \int_0^\infty r(\tau)d\tau\right) \right.$$
$$\left. + (\bar{\boldsymbol{\eta}}_1 \odot \bar{\boldsymbol{\eta}}_2) \odot \left( \cosh^{\odot 2}\left(-\boldsymbol{s} \int_0^\infty r(\tau)d\tau\right) + \sinh^{\odot 2}\left(-\boldsymbol{s} \int_0^\infty r(\tau)d\tau\right) \right) \right]$$
$$= \alpha^2 \boldsymbol{s}^T \left[ \frac{\bar{\boldsymbol{\eta}}_1^{\odot 2} + \bar{\boldsymbol{\eta}}_2^{\odot 2}}{2} \odot \sinh\left(-2\boldsymbol{s} \int_0^\infty r(\tau)d\tau\right) + (\bar{\boldsymbol{\eta}}_1 \odot \bar{\boldsymbol{\eta}}_2) \odot \cosh\left(-2\boldsymbol{s} \int_0^\infty r(\tau)d\tau\right) \right]$$
$$= \alpha^2 \sum_{j=1}^m [\boldsymbol{s}]_j \left( \frac{[\bar{\boldsymbol{\eta}}_1]_j^2 + [\bar{\boldsymbol{\eta}}_2]_j^2}{2} \sinh\left(2[\boldsymbol{s}]_j \nu\right) + [\bar{\boldsymbol{\eta}}_1]_j [\bar{\boldsymbol{\eta}}_2]_j \cosh\left(2[\boldsymbol{s}]_j \nu\right) \right)$$
$$= y, \tag{74}$$

where we defined $\nu := -\int_0^\infty r(\tau)d\tau$. Consider the function $\nu \mapsto a \sinh(\nu) + b \cosh(\nu)$. This is a strictly increasing function if $a > |b|$. Note also that

$$\frac{[\bar{\boldsymbol{\eta}}_1]_j^2 + [\bar{\boldsymbol{\eta}}_2]_j^2}{2} \geq |[\bar{\boldsymbol{\eta}}_1]_j [\bar{\boldsymbol{\eta}}_2]_j|, \tag{75}$$

which holds with equality if and only if $|[\bar{\boldsymbol{\eta}}_1]_j| = |[\bar{\boldsymbol{\eta}}_2]_j|$. However, recall from our assumptions on initialization that $[\bar{\boldsymbol{\eta}}_1]_j^2 - [\bar{\boldsymbol{\eta}}_2]_j^2 \geq \lambda > 0$, so (75) can only hold with strict inequality. Therefore,

$$g(\nu) := \sum_{j=1}^m [\boldsymbol{s}]_j \left( \frac{[\bar{\boldsymbol{\eta}}_1]_j^2 + [\bar{\boldsymbol{\eta}}_2]_j^2}{2} \sinh(2[\boldsymbol{s}]_j \nu) + [\bar{\boldsymbol{\eta}}_1]_j [\bar{\boldsymbol{\eta}}_2]_j \cosh(2[\boldsymbol{s}]_j \nu) \right)$$

is a strictly increasing (hence invertible) function because it is a sum of $m$ strictly increasing function. Using this $g(\nu)$, (74) can be written as $\alpha^2 g(\nu) = y$, and by using the inverse of $g$, we have

$$\nu = -\int_0^\infty r(\tau)d\tau = g^{-1}\left(\frac{y}{\alpha^2}\right). \tag{76}$$

Plugging (76) into (73), we get

$$
\lim_{t \to \infty} \boldsymbol{v}_1(t)
$$
$$
= \boldsymbol{U}_1 \lim_{t \to \infty} \boldsymbol{\eta}_1(t) + \alpha(\boldsymbol{I}_{k_1} - \boldsymbol{U}_1 \boldsymbol{U}_1^T)\bar{\boldsymbol{v}}_1
$$
$$
= \alpha \boldsymbol{U}_1 \left( \bar{\boldsymbol{\eta}}_1 \odot \cosh\left( g^{-1}\left(\frac{y}{\alpha^2}\right) \boldsymbol{s} \right) + \bar{\boldsymbol{\eta}}_2 \odot \sinh\left( g^{-1}\left(\frac{y}{\alpha^2}\right) \boldsymbol{s} \right) \right) + \alpha(\boldsymbol{I}_{k_1} - \boldsymbol{U}_1 \boldsymbol{U}_1^T)\bar{\boldsymbol{v}}_1,
$$
$$
\lim_{t \to \infty} \boldsymbol{v}_2(t)
$$
$$
= \boldsymbol{U}_2 \lim_{t \to \infty} \boldsymbol{\eta}_2(t) + \alpha(\boldsymbol{I}_{k_2} - \boldsymbol{U}_2 \boldsymbol{U}_2^T)\bar{\boldsymbol{v}}_2
$$
$$
= \alpha \boldsymbol{U}_2 \left( \bar{\boldsymbol{\eta}}_1 \odot \sinh\left( g^{-1}\left(\frac{y}{\alpha^2}\right) \boldsymbol{s} \right) + \bar{\boldsymbol{\eta}}_2 \odot \cosh\left( g^{-1}\left(\frac{y}{\alpha^2}\right) \boldsymbol{s} \right) \right) + \alpha(\boldsymbol{I}_{k_2} - \boldsymbol{U}_2 \boldsymbol{U}_2^T)\bar{\boldsymbol{v}}_2.
$$

This finishes the proof.

## H    PROOF OF THEOREM 7

### H.1    CONVERGENCE OF LOSS TO ZERO

We first show that given the conditions on initialization, the training loss $\mathcal{L}(\boldsymbol{\Theta}(t))$ converges to zero. Recall from (10) that the linear fully-connected network can be written as

$$
f_{\mathrm{fc}}(\boldsymbol{x}; \boldsymbol{\Theta}_{\mathrm{fc}}) = \boldsymbol{x}^T \boldsymbol{W}_1 \boldsymbol{W}_2 \cdots \boldsymbol{W}_{L-1} \boldsymbol{w}_L.
$$

From the definition of the training loss $\mathcal{L}$, it is straightforward to check that the gradient flow dynamics read

$$
\begin{aligned}
\dot{\boldsymbol{W}}_l &= -\nabla_{\boldsymbol{W}_l} \mathcal{L}(\boldsymbol{\Theta}_{\mathrm{fc}}) = -\boldsymbol{W}_{l-1}^T \cdots \boldsymbol{W}_1^T \boldsymbol{X}^T \boldsymbol{r} \boldsymbol{w}_L^T \boldsymbol{W}_{L-1}^T \cdots \boldsymbol{W}_{l+1}^T \text{ for } l \in [L-1], \\
\dot{\boldsymbol{w}}_L &= -\nabla_{\boldsymbol{w}_L} \mathcal{L}(\boldsymbol{\Theta}_{\mathrm{fc}}) = -\boldsymbol{W}_{L-1}^T \cdots \boldsymbol{W}_1^T \boldsymbol{X}^T \boldsymbol{r}, \\
\boldsymbol{W}_l(0) &= \alpha \bar{\boldsymbol{W}}_l \text{ for } l \in [L-1], \\
\boldsymbol{w}_L(0) &= \alpha \bar{\boldsymbol{w}}_L,
\end{aligned}
\tag{77}
$$

where $\boldsymbol{r} \in \mathbb{R}^n$ is the residual vector satisfying $[\boldsymbol{r}]_i = f_{\mathrm{fc}}(\boldsymbol{x}_i; \boldsymbol{\Theta}_{\mathrm{fc}}) - y_i$, as defined in Section 2.1. From (77), we have

$$
\begin{aligned}
\boldsymbol{W}_l^T \dot{\boldsymbol{W}}_l &= \dot{\boldsymbol{W}}_{l+1} \boldsymbol{W}_{l+1}^T = -\boldsymbol{W}_l^T \cdots \boldsymbol{W}_1^T \boldsymbol{X}^T \boldsymbol{r} \boldsymbol{w}_L^T \boldsymbol{W}_{L-1}^T \cdots \boldsymbol{W}_{l+1}^T, \\
\dot{\boldsymbol{W}}_l^T \boldsymbol{W}_l &= \boldsymbol{W}_{l+1} \dot{\boldsymbol{W}}_{l+1}^T = -\boldsymbol{W}_{l+1} \cdots \boldsymbol{W}_{L-1} \boldsymbol{w}_L \boldsymbol{r}^T \boldsymbol{X} \boldsymbol{W}_1 \cdots \boldsymbol{W}_l,
\end{aligned}
$$

for any $l \in [L-2]$. From this, we have

$$
\frac{d}{dt} \boldsymbol{W}_l^T \boldsymbol{W}_l = \frac{d}{dt} \boldsymbol{W}_{l+1} \boldsymbol{W}_{l+1}^T,
$$

and thus

$$
\begin{aligned}
\boldsymbol{W}_l(t)^T \boldsymbol{W}_l(t) - \boldsymbol{W}_{l+1}(t) \boldsymbol{W}_{l+1}(t)^T &= \boldsymbol{W}_l(0)^T \boldsymbol{W}_l(0) - \boldsymbol{W}_{l+1}(0) \boldsymbol{W}_{l+1}(0)^T \\
&= \alpha^2 \bar{\boldsymbol{W}}_l^T \bar{\boldsymbol{W}}_l - \alpha^2 \bar{\boldsymbol{W}}_{l+1} \bar{\boldsymbol{W}}_{l+1}^T,
\end{aligned}
\tag{78}
$$

for any $l \in [L-2]$. Similarly, we have

$$
\begin{aligned}
\boldsymbol{W}_{L-1}(t)^T \boldsymbol{W}_{L-1}(t) - \boldsymbol{w}_L(t) \boldsymbol{w}_L(t)^T &= \boldsymbol{W}_{L-1}(0)^T \boldsymbol{W}_{L-1}(0) - \boldsymbol{w}_L(0) \boldsymbol{w}_L(0)^T \\
&= \alpha^2 \bar{\boldsymbol{W}}_{L-1}^T \bar{\boldsymbol{W}}_{L-1} - \alpha^2 \bar{\boldsymbol{w}}_L \bar{\boldsymbol{w}}_L^T.
\end{aligned}
\tag{79}
$$

Let us now consider the time derivative of $\mathcal{L}(\boldsymbol{\Theta}_{\mathrm{fc}}(t))$. We have the following chain of upper bounds on the time derivative:

$$
\begin{aligned}
\frac{d}{dt} \mathcal{L}(\boldsymbol{\Theta}_{\mathrm{fc}}(t)) = \nabla_{\boldsymbol{\Theta}_{\mathrm{fc}}} \mathcal{L}(\boldsymbol{\Theta}_{\mathrm{fc}}(t))^T \dot{\boldsymbol{\Theta}}_{\mathrm{fc}}(t) &= -\|\nabla_{\boldsymbol{\Theta}_{\mathrm{fc}}} \mathcal{L}(\boldsymbol{\Theta}_{\mathrm{fc}}(t))\|_2^2 \\
&\leq -\|\nabla_{\boldsymbol{w}_L} \mathcal{L}(\boldsymbol{\Theta}_{\mathrm{fc}}(t))\|_2^2 = -\|\dot{\boldsymbol{w}}_L(t)\|_2^2 \\
&= -\|\boldsymbol{W}_{L-1}^T \cdots \boldsymbol{W}_1^T \boldsymbol{X}^T \boldsymbol{r}\|_2^2.
\end{aligned}
\tag{80}
$$

Note from (80) that if $\boldsymbol{W}_{L-1}^T \cdots \boldsymbol{W}_1^T$ is full-rank, its minimum singular value is positive, and one can bound

$$\|\boldsymbol{W}_{L-1}^T \cdots \boldsymbol{W}_1^T \boldsymbol{X}^T \boldsymbol{r}\|_2 \geq \sigma_{\min}(\boldsymbol{W}_{L-1}^T \cdots \boldsymbol{W}_1^T)\|\boldsymbol{X}^T \boldsymbol{r}\|_2. \tag{81}$$

We now prove that the matrix $\boldsymbol{W}_{L-1}^T \cdots \boldsymbol{W}_1^T$ is full-rank, and its minimum singular value is bounded from below by $\alpha^{L-1}\lambda^{(L-1)/2}$ for any $t \geq 0$. To show this, it suffices to show that

$$\boldsymbol{W}_{L-1}^T \cdots \boldsymbol{W}_1^T \boldsymbol{W}_1 \cdots \boldsymbol{W}_{L-1} \succeq \alpha^{2L-2}\lambda^{L-1}\boldsymbol{I}_d. \tag{82}$$

Now,

$$\boldsymbol{W}_{L-1}^T \cdots \boldsymbol{W}_2^T \boldsymbol{W}_1^T \boldsymbol{W}_1 \boldsymbol{W}_2 \cdots \boldsymbol{W}_{L-1}$$
$$\overset{(a)}{=} \boldsymbol{W}_{L-1}^T \cdots \boldsymbol{W}_2^T (\boldsymbol{W}_2 \boldsymbol{W}_2^T + \alpha^2 \bar{\boldsymbol{W}}_1^T \bar{\boldsymbol{W}}_1 - \alpha^2 \bar{\boldsymbol{W}}_2 \bar{\boldsymbol{W}}_2^T)\boldsymbol{W}_2 \cdots \boldsymbol{W}_{L-1}$$
$$\overset{(b)}{\succeq} \boldsymbol{W}_{L-1}^T \cdots \boldsymbol{W}_3^T \boldsymbol{W}_2^T \boldsymbol{W}_2 \boldsymbol{W}_2^T \boldsymbol{W}_2 \boldsymbol{W}_3 \cdots \boldsymbol{W}_{L-1}$$
$$\overset{(a)}{=} \boldsymbol{W}_{L-1}^T \cdots \boldsymbol{W}_3^T (\boldsymbol{W}_3 \boldsymbol{W}_3^T + \alpha^2 \bar{\boldsymbol{W}}_2^T \bar{\boldsymbol{W}}_2 - \alpha^2 \bar{\boldsymbol{W}}_3 \bar{\boldsymbol{W}}_3^T)^2 \boldsymbol{W}_3 \cdots \boldsymbol{W}_{L-1}$$
$$\overset{(b)}{\succeq} \boldsymbol{W}_{L-1}^T \cdots \boldsymbol{W}_3^T (\boldsymbol{W}_3 \boldsymbol{W}_3^T)^2 \boldsymbol{W}_3 \cdots \boldsymbol{W}_{L-1}$$
$$= \cdots \succeq (\boldsymbol{W}_{L-1}^T \boldsymbol{W}_{L-1})^{L-1},$$

where equalities marked in (a) used (78), and inequalities marked in (b) used the initialization conditions $\bar{\boldsymbol{W}}_l^T \bar{\boldsymbol{W}}_l \succeq \bar{\boldsymbol{W}}_{l+1}\bar{\boldsymbol{W}}_{l+1}^T$. Next, it follows from (79) that

$$(\boldsymbol{W}_{L-1}^T \boldsymbol{W}_{L-1})^{L-1} = (\boldsymbol{w}_L \boldsymbol{w}_L^T + \alpha^2 \bar{\boldsymbol{W}}_{L-1}^T \bar{\boldsymbol{W}}_{L-1} - \alpha^2 \bar{\boldsymbol{w}}_L \bar{\boldsymbol{w}}_L^T)^{L-1}$$
$$\succeq \alpha^{2L-2}(\bar{\boldsymbol{W}}_{L-1}^T \bar{\boldsymbol{W}}_{L-1} - \bar{\boldsymbol{w}}_L \bar{\boldsymbol{w}}_L^T)^{L-1}$$
$$\overset{(c)}{\succeq} \alpha^{2L-2}\lambda^{L-1}\boldsymbol{I}_d.$$

where (c) used the assumption that $\bar{\boldsymbol{W}}_{L-1}^T \bar{\boldsymbol{W}}_{L-1} - \bar{\boldsymbol{w}}_L \bar{\boldsymbol{w}}_L^T \succeq \lambda \boldsymbol{I}_d$. This proves (82). Applying (82) to (80) then gives

$$\frac{d}{dt}\mathcal{L}(\boldsymbol{\Theta}_{\mathrm{fc}}(t)) \leq -\|\boldsymbol{W}_{L-1}^T \cdots \boldsymbol{W}_1^T \boldsymbol{X}^T \boldsymbol{r}\|_2^2$$
$$\leq -\sigma_{\min}(\boldsymbol{W}_{L-1}^T \cdots \boldsymbol{W}_1^T)^2 \|\boldsymbol{X}^T \boldsymbol{r}\|_2^2$$
$$\leq -\alpha^{2L-2}\lambda^{L-1}\|\boldsymbol{X}^T \boldsymbol{r}\|_2^2$$
$$\overset{(d)}{\leq} -\alpha^{2L-2}\lambda^{L-1}\sigma_{\min}(\boldsymbol{X})^2 \|\boldsymbol{r}\|_2^2$$
$$= -\alpha^{2L-2}\lambda^{L-1}\sigma_{\min}(\boldsymbol{X})^2 \mathcal{L}(\boldsymbol{\Theta}_{\mathrm{fc}}(t)),$$

where (d) used the fact that $\boldsymbol{X}^T$ is a full column rank matrix to apply a bound similar to (81). From this, we get

$$\mathcal{L}(\boldsymbol{\Theta}_{\mathrm{fc}}(t)) \leq \mathcal{L}(\boldsymbol{\Theta}_{\mathrm{fc}}(0)) \exp(-\alpha^{2L-2}\lambda^{L-1}\sigma_{\min}(\boldsymbol{X})^2 t),$$

hence proving $\mathcal{L}(\boldsymbol{\Theta}_{\mathrm{fc}}(t)) \to 0$ as $t \to \infty$.

## H.2 CHARACTERIZING THE LIMIT POINT: $\alpha \to 0$ CASE

Now, we move on to characterize the limit points of the gradient flow, for the "active regime" case $\alpha \to 0$. This part of the proof is motivated from the analysis in Ji & Telgarsky (2019a).

Let $\boldsymbol{u}_l$ and $\boldsymbol{v}_l$ be the top left and right singular vectors of $\boldsymbol{W}_l$, for $l \in [L-1]$. Note that since $\boldsymbol{W}_l$ varies over time, the singular vectors and singular value also vary over time. Similarly, let $s_l$ be the largest singular value of $\boldsymbol{W}_l$. We will show that the linear coefficients $\boldsymbol{\beta}_{\mathrm{fc}}(\boldsymbol{\Theta}_{\mathrm{fc}}) = \boldsymbol{W}_1 \cdots \boldsymbol{W}_{L-1}\boldsymbol{w}_L$ align with $\boldsymbol{u}_1$ as $\alpha \to 0$, and $\boldsymbol{u}_1$ is in the subspace of $\mathrm{row}(\boldsymbol{X})$ in the limit $\alpha \to 0$, hence proving that $\boldsymbol{\beta}_{\mathrm{fc}}(\boldsymbol{\Theta}_{\mathrm{fc}})$ is the minimum $\ell_2$ norm solution in the limit $\alpha \to 0$.

First, note from (78) and (79) that if we take trace of both sides, we get

$$\|\boldsymbol{W}_l\|_{\mathrm{F}}^2 - \|\boldsymbol{W}_{l+1}\|_{\mathrm{F}}^2 = \alpha^2(\|\bar{\boldsymbol{W}}_l\|_{\mathrm{F}}^2 - \|\bar{\boldsymbol{W}}_{l+1}\|_{\mathrm{F}}^2) \quad \text{for } l \in [L-2],$$

$$\|\boldsymbol{W}_{L-1}\|_{\mathrm{F}}^2 - \|\boldsymbol{w}_L\|_2^2 = \alpha^2(\|\bar{\boldsymbol{W}}_{L-1}\|_{\mathrm{F}}^2 - \|\bar{\boldsymbol{w}}_L\|_2^2).$$

Summing the equations above for $l, l+1, \ldots, L-1$, we get

$$\|\boldsymbol{W}_l\|_{\mathrm{F}}^2 - \|\boldsymbol{w}_L\|_2^2 = \alpha^2(\|\bar{\boldsymbol{W}}_l\|_{\mathrm{F}}^2 - \|\bar{\boldsymbol{w}}_L\|_2^2). \tag{83}$$

Next, consider the operator norms (i.e., the maximum singular values), denoted as $\|\cdot\|_2$, of the matrices.

$$
\begin{aligned}
\|\boldsymbol{W}_l\|_2^2 &\geq \boldsymbol{u}_{l+1}^T \boldsymbol{W}_l^T \boldsymbol{W}_l \boldsymbol{u}_{l+1} \\
&\overset{(e)}{=} \boldsymbol{u}_{l+1}^T \boldsymbol{W}_{l+1} \boldsymbol{W}_{l+1}^T \boldsymbol{u}_{l+1} + \alpha^2 \boldsymbol{u}_{l+1}^T (\bar{\boldsymbol{W}}_l^T \bar{\boldsymbol{W}}_l - \bar{\boldsymbol{W}}_{l+1} \bar{\boldsymbol{W}}_{l+1}^T) \boldsymbol{u}_{l+1} \\
&= \|\boldsymbol{W}_{l+1}\|_2^2 + \alpha^2 \boldsymbol{u}_{l+1}^T (\bar{\boldsymbol{W}}_l^T \bar{\boldsymbol{W}}_l - \bar{\boldsymbol{W}}_{l+1} \bar{\boldsymbol{W}}_{l+1}^T) \boldsymbol{u}_{l+1} \\
&\geq \|\boldsymbol{W}_{l+1}\|_2^2 - \alpha^2 \|\bar{\boldsymbol{W}}_l^T \bar{\boldsymbol{W}}_l - \bar{\boldsymbol{W}}_{l+1} \bar{\boldsymbol{W}}_{l+1}^T\|_2 \quad \text{for } l \in [L-2], \\
\|\boldsymbol{W}_{L-1}\|_2^2 &\geq \frac{\boldsymbol{w}_L}{\|\boldsymbol{w}_L\|_2} \boldsymbol{W}_{L-1}^T \boldsymbol{W}_{L-1} \frac{\boldsymbol{w}_L}{\|\boldsymbol{w}_L\|_2} \\
&\overset{(f)}{=} \frac{\boldsymbol{w}_L}{\|\boldsymbol{w}_L\|_2} \boldsymbol{w}_L \boldsymbol{w}_L^T \frac{\boldsymbol{w}_L}{\|\boldsymbol{w}_L\|_2} + \alpha^2 \frac{\boldsymbol{w}_L}{\|\boldsymbol{w}_L\|_2} (\bar{\boldsymbol{W}}_{L-1}^T \bar{\boldsymbol{W}}_{L-1} - \bar{\boldsymbol{w}}_L \bar{\boldsymbol{w}}_L^T) \frac{\boldsymbol{w}_L}{\|\boldsymbol{w}_L\|_2} \\
&\geq \|\boldsymbol{w}_L\|_2^2 - \alpha^2 \|\bar{\boldsymbol{W}}_{L-1}^T \bar{\boldsymbol{W}}_{L-1} - \bar{\boldsymbol{w}}_L \bar{\boldsymbol{w}}_L^T\|_2.
\end{aligned}
$$

where (e) used (78) and (f) used (79). Summing the inequalities gives

$$\|\boldsymbol{W}_l\|_2^2 \geq \|\boldsymbol{w}_L\|_2^2 - \alpha^2 \sum_{k=1}^{L-1} \|\bar{\boldsymbol{W}}_k^T \bar{\boldsymbol{W}}_k - \bar{\boldsymbol{W}}_{k+1} \bar{\boldsymbol{W}}_{k+1}^T\|_2. \tag{84}$$

From (83) and (84), we get a bound on the gap between the second powers of the Frobenius norm (or the $\ell_2$ norm of singular values) and operator norm (or the maximum singular value $s_l$) of $\boldsymbol{W}_l$:

$$\|\boldsymbol{W}_l(t)\|_{\mathrm{F}}^2 - \|\boldsymbol{W}_l(t)\|_2^2 \leq \alpha^2(\|\bar{\boldsymbol{W}}_l\|_{\mathrm{F}}^2 - \|\bar{\boldsymbol{w}}_L\|_2^2) + \alpha^2 \sum_{k=l}^{L-1} \|\bar{\boldsymbol{W}}_k^T \bar{\boldsymbol{W}}_k - \bar{\boldsymbol{W}}_{k+1} \bar{\boldsymbol{W}}_{k+1}^T\|_2, \tag{85}$$

which holds for any $t \geq 0$. The gap (85) implies that each $\boldsymbol{W}_l$, for $l \in [L-1]$, can be written as

$$\boldsymbol{W}_l(t) = s_l(t) \boldsymbol{u}_l(t) \boldsymbol{v}_l(t)^T + O(\alpha^2). \tag{86}$$

Next, we show that the "adjacent" singular vectors $\boldsymbol{v}_l$ and $\boldsymbol{u}_{l+1}$ align with each other as $\alpha \to 0$. To this end, we will get lower and upper bounds for a quantity $\boldsymbol{v}_l^T \boldsymbol{W}_{l+1} \boldsymbol{W}_{l+1}^T \boldsymbol{v}_l$.

$$
\begin{aligned}
\boldsymbol{v}_l^T \boldsymbol{W}_{l+1} \boldsymbol{W}_{l+1}^T \boldsymbol{v}_l &= \boldsymbol{v}_l^T \boldsymbol{W}_l^T \boldsymbol{W}_l \boldsymbol{v}_l - \alpha^2 \boldsymbol{v}_l^T \bar{\boldsymbol{W}}_l^T \bar{\boldsymbol{W}}_l \boldsymbol{v}_l + \alpha^2 \boldsymbol{v}_l^T \bar{\boldsymbol{W}}_{l+1} \bar{\boldsymbol{W}}_{l+1}^T \boldsymbol{v}_l \\
&\geq \|\boldsymbol{W}_l\|_2^2 - \alpha^2 \|\bar{\boldsymbol{W}}_l^T \bar{\boldsymbol{W}}_l - \bar{\boldsymbol{W}}_{l+1} \bar{\boldsymbol{W}}_{l+1}^T\|_2 \\
&= s_l^2 - \alpha^2 \|\bar{\boldsymbol{W}}_l^T \bar{\boldsymbol{W}}_l - \bar{\boldsymbol{W}}_{l+1} \bar{\boldsymbol{W}}_{l+1}^T\|_2, \\
\boldsymbol{v}_l^T \boldsymbol{W}_{l+1} \boldsymbol{W}_{l+1}^T \boldsymbol{v}_l &= \boldsymbol{v}_l^T (s_{l+1}^2 \boldsymbol{u}_{l+1} \boldsymbol{u}_{l+1}^T + \boldsymbol{W}_{l+1} \boldsymbol{W}_{l+1}^T - s_{l+1}^2 \boldsymbol{u}_{l+1} \boldsymbol{u}_{l+1}^T) \boldsymbol{v}_l \\
&= s_{l+1}^2 (\boldsymbol{v}_l^T \boldsymbol{u}_{l+1})^2 + \boldsymbol{v}_l^T (\boldsymbol{W}_{l+1} \boldsymbol{W}_{l+1}^T - s_{l+1}^2 \boldsymbol{u}_{l+1} \boldsymbol{u}_{l+1}^T) \boldsymbol{v}_l \\
&\leq s_{l+1}^2 (\boldsymbol{v}_l^T \boldsymbol{u}_{l+1})^2 + \|\boldsymbol{W}_{l+1}\|_{\mathrm{F}}^2 - \|\boldsymbol{W}_{l+1}\|_2^2.
\end{aligned}
\tag{87, 88}
$$

Combining (87), (88), and (85), we get

$$s_l^2 \leq s_{l+1}^2 (\boldsymbol{v}_l^T \boldsymbol{u}_{l+1})^2 + \alpha^2 \|\bar{\boldsymbol{W}}_l^T \bar{\boldsymbol{W}}_l - \bar{\boldsymbol{W}}_{l+1} \bar{\boldsymbol{W}}_{l+1}^T\|_2 + \|\boldsymbol{W}_{l+1}\|_{\mathrm{F}}^2 - \|\boldsymbol{W}_{l+1}\|_2^2$$

$$\leq s_{l+1}^2 (\boldsymbol{v}_l^T \boldsymbol{u}_{l+1})^2 + \alpha^2 (\|\bar{\boldsymbol{W}}_{l+1}\|_{\mathrm{F}}^2 - \|\bar{\boldsymbol{w}}_L\|_2^2) + \alpha^2 \sum_{k=l}^{L-1} \|\bar{\boldsymbol{W}}_k^T \bar{\boldsymbol{W}}_k - \bar{\boldsymbol{W}}_{k+1} \bar{\boldsymbol{W}}_{k+1}^T\|_2. \tag{89}$$

Next, by a similar reasoning as (87), we have

$$s_l^2 \geq \boldsymbol{u}_{l+1}^T \boldsymbol{W}_l^T \boldsymbol{W}_l \boldsymbol{u}_{l+1} \geq s_{l+1}^2 - \alpha^2 \|\bar{\boldsymbol{W}}_l^T \bar{\boldsymbol{W}}_l - \bar{\boldsymbol{W}}_{l+1} \bar{\boldsymbol{W}}_{l+1}^T\|_2. \tag{90}$$

Combining (89) and (90) and dividing both sides by $s_{l+1}^2$, we get

$$(\boldsymbol{v}_l(t)^T \boldsymbol{u}_{l+1}(t))^2 \geq 1 - \alpha^2 \frac{G_l}{s_{l+1}(t)^2} \tag{91}$$

for $t \geq 0$, where

$$G_l := \left\| \bar{\boldsymbol{W}}_l^T \bar{\boldsymbol{W}}_l - \bar{\boldsymbol{W}}_{l+1} \bar{\boldsymbol{W}}_{l+1}^T \right\|_2 + (\left\| \bar{\boldsymbol{W}}_{l+1} \right\|_F^2 - \left\| \bar{\boldsymbol{w}}_L \right\|_2^2) + \sum_{k=l}^{L-1} \left\| \bar{\boldsymbol{W}}_k^T \bar{\boldsymbol{W}}_k - \bar{\boldsymbol{W}}_{k+1} \bar{\boldsymbol{W}}_{k+1}^T \right\|_2.$$

By a similar argument, we can also get

$$\frac{(\boldsymbol{v}_{L-1}(t)^T \boldsymbol{w}_L(t))^2}{\left\| \boldsymbol{w}_L(t) \right\|_2^2} \geq 1 - \alpha^2 \frac{G_{L-1}}{\left\| \boldsymbol{w}_L(t) \right\|_2^2}, \tag{92}$$

where

$$G_{L-1} := 2 \left\| \bar{\boldsymbol{W}}_{L-1}^T \bar{\boldsymbol{W}}_{L-1} - \bar{\boldsymbol{w}}_L \bar{\boldsymbol{w}}_L^T \right\|_2.$$

From (91) and (92), we can note that as $\alpha \to 0$, the inner product between the adjacent singular vectors converges to $\pm 1$, unless $s_2, \ldots, s_{L-1}, \left\| \boldsymbol{w}_L \right\|_2$ also diminish to zero. So it is left to show that the singular values do not diminish to zero as $\alpha \to 0$. To this end, recall that we proved in the previous subsection that

$$\lim_{t \to \infty} \boldsymbol{X} \boldsymbol{W}_1(t) \cdots \boldsymbol{W}_{L-1}(t) \boldsymbol{w}_L(t) = \boldsymbol{y}.$$

A necessary condition for this to hold is that

$$\frac{\left\| \boldsymbol{y} \right\|_2}{\left\| \boldsymbol{X} \right\|_2} \leq \lim_{t \to \infty} \left\| \boldsymbol{W}_1(t) \cdots \boldsymbol{W}_{L-1}(t) \boldsymbol{w}_L(t) \right\|_2 \leq \lim_{t \to \infty} \prod_{l=1}^{L-1} s_l(t) \left\| \boldsymbol{w}_L(t) \right\|_2.$$

This means that after converging to the global minimum solution of the problem (i.e., $t \to \infty$), the product of the singular values must be at least greater than some constant independent of $\alpha$. Moreover, we can see from (87) and (90) that the gap between singular values squared of adjacent layers is bounded by $O(\alpha^2)$, for all $t \geq 0$; so the maximum singular values become closer and closer to each other as $\alpha$ diminishes. This implies that

$$\lim_{\alpha \to 0} \lim_{t \to \infty} s_l(t) \geq \frac{\left\| \boldsymbol{y} \right\|_2^{1/L}}{\left\| \boldsymbol{X} \right\|_2^{1/L}} \quad \text{for } l \in [L-1], \quad \lim_{\alpha \to 0} \lim_{t \to \infty} \left\| \boldsymbol{w}_L(t) \right\|_2 \geq \frac{\left\| \boldsymbol{y} \right\|_2^{1/L}}{\left\| \boldsymbol{X} \right\|_2^{1/L}}.$$

Therefore, we have the alignment of singular vectors at convergence as $\alpha \to 0$:

$$\lim_{\alpha \to 0} \lim_{t \to \infty} (\boldsymbol{v}_l(t)^T \boldsymbol{u}_{l+1}(t))^2 = 1, \quad \text{for } l \in [L-2], \quad \lim_{\alpha \to 0} \lim_{t \to \infty} \frac{(\boldsymbol{v}_{L-1}(t)^T \boldsymbol{w}_L(t))^2}{\left\| \boldsymbol{w}_L(t) \right\|_2^2} = 1. \tag{93}$$

So far, we saw from (86) that $\boldsymbol{W}_l(t)$'s become rank-1 matrices as $\alpha \to 0$, and from (93) that the top singular vectors align with each other as $t \to \infty$ and $\alpha \to 0$. These imply that, as $t \to \infty$ and $\alpha \to 0$, $\boldsymbol{\beta}_{\text{fc}}(\boldsymbol{\Theta}_{\text{fc}})$ is a scalar multiple of the $\boldsymbol{u}_1$, the top left singular vector of $\boldsymbol{W}_1$:

$$\lim_{\alpha \to 0} \lim_{t \to \infty} \boldsymbol{\beta}_{\text{fc}}(\boldsymbol{\Theta}_{\text{fc}}(t)) = c \cdot \lim_{\alpha \to 0} \lim_{t \to \infty} \boldsymbol{u}_1(t), \tag{94}$$

for some $c \in \mathbb{R}$.

In light of (94), it remains to take a close look at $\boldsymbol{u}_1(t)$. Note from the gradient flow dynamics of $\boldsymbol{W}_1$ that $\dot{\boldsymbol{W}}_1$ is always a rank-1 matrix whose columns are in the row space of $\boldsymbol{X}$, since $\boldsymbol{X}^T \boldsymbol{r} \in \text{row}(\boldsymbol{X})$. This implies that, if we decompose $\boldsymbol{W}_1$ into two orthogonal components $\boldsymbol{W}_1^\perp$ and $\boldsymbol{W}_1^\parallel$ so that the columns in $\boldsymbol{W}_1^\parallel$ are in $\text{row}(\boldsymbol{X})$ and the columns in $\boldsymbol{W}_1^\perp$ are in the orthogonal subspace $\text{row}(\boldsymbol{X})^\perp$, we have

$$\dot{\boldsymbol{W}}_1^\perp = \boldsymbol{0}, \quad \dot{\boldsymbol{W}}_1^\parallel = \dot{\boldsymbol{W}}_1.$$

That is, any component $\boldsymbol{W}_1^\perp(0)$ orthogonal to $\text{row}(\boldsymbol{X})$ remains unchanged for all $t \geq 0$, while the component $\boldsymbol{W}_1^\parallel$ changes by the gradient flow. Since we have

$$\left\| \boldsymbol{W}_1^\perp(t) \right\|_F = \left\| \boldsymbol{W}_1^\perp(0) \right\|_F \leq \alpha \left\| \bar{\boldsymbol{W}}_l \right\|_F,$$

the component in $\boldsymbol{W}_1$ that is orthogonal to $\text{row}(\boldsymbol{X})$ diminishes to zero as $\alpha \to 0$. This means that at the limit $\alpha \to 0$, the columns of $\boldsymbol{W}_1$ are entirely from $\text{row}(\boldsymbol{X})$, which also means that

$$\lim_{\alpha \to 0} \lim_{t \to \infty} \boldsymbol{\beta}_{\text{fc}}(\boldsymbol{\Theta}_{\text{fc}}(t)) \in \text{row}(\boldsymbol{X}).$$

However, recall that there is only one unique global minimum of $\boldsymbol{X}\boldsymbol{z} = \boldsymbol{y}$ in $\text{row}(\boldsymbol{X})$: namely, $\boldsymbol{z} = \boldsymbol{X}^T (\boldsymbol{X}\boldsymbol{X}^T)^{-1} \boldsymbol{y}$, the minimum $\ell_2$ norm solution. This finishes the proof.

