# OpenReview forum: "A unifying view on implicit bias in training linear neural networks"
_ICLR.cc/2021/Conference — ICLR 2021 Poster_

### Official Review · AnonReviewer1 · 2020-10-26
**A unified characterization of the implicit bias via tensors for linear networks**

**Rating:** 6
**Confidence:** 4

**Review:**

This paper first provides a general framework to represent fully-connected networks, convolutional networks, diagonal networks, etc., using tensors. For classification, with the exponential loss, linearly separable data and gradient flow, it is proved that the parameters of a linear tensor network converge to the singular vectors of the limiting data tensor. When the data tensor is orthogonally decomposable, which is true for linear diagonal networks and linear full-length convolutional networks, the function computed by the network is shown to converge in direction to a stationary point of the l_{2/L}-margin maximization problem, where L denotes the depth of the network. For underdetermined regression and the squared loss, this paper proves that gradient flow finds a global minimum which also minimizes a norm-like function. This paper further considers the special case where there is only one data point and two layers, and provides empirical support.

I think the general framework which represents networks using tensors is interesting, and may be useful elsewhere. It is also nice to have a unified analysis of the implicit bias which also removes a few assumptions from prior results.

On the other hand, I think it would be better to include a thorough comparison between proof techniques used in this paper and in prior work. For example, Ji & Telgarsky (2020) prove that the weights and the gradients converge to the same direction, and Theorem 1 of this paper assumes that the weighted average of the data points by the residual vector r also converges in direction; the claim of Theorem 1 then almost already follows. What remains to be proved is that d|v_l|^2/dt is the same for all layer l, but similar results have appeared in (Arora et al., 2018), (Simon S Du, Wei Hu, and Jason D Lee. Algorithmic regularization in learning deep homogeneous models: Layers are automatically balanced), etc.

Here are some minor comments:
1. For linear tensor networks, I think eq. (6) is actually a multi-linear form. I think it is worth pointing out and can help the readers.
2. In the equation U_l^H U_l = I_{k_l} in Assumption 1, it seems that the dimensions don't match.
3. It seems that the last sentence of Section 5.2 shouldn't be there.

---

> ### Author Response · Authors · 2020-11-20
> **Author Response to Reviewer 1**
>
> We appreciate the reviewer for the review and the thoughtful comments. Below, we reply to the comments raised by the reviewer. Also, please take a look at our revised manuscript.
>
> Re. proof techniques: It is true that our classification results rely on the results from [Ji & Telgarsky, 2020] and a rather “standard” technique for showing $d|v_l|^2/dt$ is the same over all layers. For Theorem 1, our key contribution lies in a novel view of the limit directions of network parameters as singular vectors of tensors, which leads to Corollary 1 on the implicit bias of fully-connected networks. Corollary 1 was also previously proved in [Ji & Telgarsky, 2020], but we provide an alternative proof.
>
> For the remaining theorems on classification, we identify a class of linear tensor networks (Assumption 1) that we can provide a more complete characterization of implicit bias. Specifically, one of the implications of Theorem 2 is that, for $L=2$, the network parameters converge to the top singular vectors of the data tensor, which is a more specific characterization than what [Ji & Telgarsky, 2020] provides us. Please note also that the regression results do not rely on the directional convergence results [Ji & Telgarsky, 2020]; we instead characterize the limit points using the solutions of the gradient flow dynamics.
>
> Below, we address the numbered comments:
> 1. Thanks for pointing this out. We have added a comment on the multi-linear form of (6).
> 2. $U_l^H U_l$ should be an $m$ by $m$ identity matrix. Thanks for the catch, and we corrected this in the new version.
> 3. The last sentence meant to be a concluding remark for Section 5 and a pointer to the corollaries (Appendix A). We deleted the sentence as per your suggestion.
>
> Again, thank you for your valuable comments. Please do let us know if you have any remaining questions/comments.
>
> Best,
> Authors

---

### Official Review · AnonReviewer3 · 2020-11-02
**Reviewer 3**

**Rating:** 7
**Confidence:** 2

**Review:**

The paper gives a general framework of what they call linear tensor networks, which is essentially linear neural networks that is expressed in the tensor formulation. Under certain assumptions, they show that the network parameters converge to the direction of singular vectors of the tensor. For linear fully-connected neural networks, they recovered prior result. For classification and underdetermined linear regression, they show under decomposable assumption the limit point of gradient flow and its characterization.  They further corroborate their results partly by a few very simple regression and classification task.

* Strengths:

1. The paper is organized in a way easy to read and follow. Given the amount of different pieces of results it has, it has a clear introduction section to explain the motivation and their contributions. The framework is written with examples of simple models. Within each piece of result when studying the limiting behavior of gradient flow, the assumptions required are stated clearly.

2. The contribution of the paper, especially with respect to a number of related works, has been listed in a rather detailed and clear way in the paper. For instance for separable classification, the relationship with counterexamples for certain initializations in prior work; the comparison with prior results that only consider full-length filters etc. These comparisons and relationships can be found everywhere in the paper, and I think it meets the satisfactory criterion for a framework paper that aims at unifying and generalizing the prior understandings.

3. The incremental contribution from prior work is valid. I think removing the convergence assumption for a number of prior works in characterizing the gradient flow and that generalizing the Woodworth et al.'s result for beyond the difference structure are both cool. While the setting that the paper has considered is still arguably over-simplified, I still view it as a nice step toward a better understanding for the implicit bias of the gradient flow for linear neural networks that offer nice insights.

* weaknesses:

1. To me some motivation and explanation behind the assumptions are lacking. Especially in Remark 1, if the initial direction is for guaranteeing convergence, then it might be more restrictive in use as compared with the convergence assumption, right? If not, some examples will be nice. Also, I believe it will be helpful to say a few words first on the assumptions of matrices, for theorems like Theorem 7.

2. The classification experiments in Section 6 couldn't serve as corroboration for the theoretical results in the paper. Even though it is in align with some recent papers, it actually only shows the difference of limiting / finite-horizon behavior in dependence of the initial distribution. In that sense, I don't know how much it corroborates, or matters considering the points that this paper is making here. Also, the synthetic  dataset for the experiments seem a little bit too simple.

3. Overall, I find this discussion on the problem's implicit bias, in terms of initial distribution for the 2-layer one data point case a bit confusing. It will be nice to state the motivation for considering this case, and also its implication for some for more general overparametrized setups.

---

> ### Author Response · Authors · 2020-11-20
> **Author Response to Reviewer 3**
>
> We thank the reviewer for their efforts as well as their valuable comments. Below, we address the weaknesses of the paper the reviewer pointed out. Please also check our revised paper.
>
> Re. 1: In order to explain/motivate Assumption 1 further, we have added comments after Assumption 1 and Theorem 2 explaining what this assumption means for $L=2$ and how it helps proof of Theorem 2.
>
> As for the conditions on initial directions, it is true that the conditions only provide sufficient conditions for convergence of loss, hence stronger/more restrictive than the convergence assumption. However, we believe proving (instead of assuming) convergence of loss is meaningful because there are known examples where linear neural networks never converge to the true solution under some initializations [Bartlett et al., 2018; Arora et al., 2019a] (please see Remark 1). Given such examples, it is not clear a priori whether a given initialization direction will satisfy the convergence assumption. In contrast, our theorems provide sufficient conditions on initialization for convergence, which is easier to test. Also, as we discussed in Remark 1, our sufficient condition can lead to an initialization scheme (setting $v_L = 0$ and randomly sampling others) that ensures global convergence of loss with probability 1.
>
> The conditions on the matrices in Theorem 7 are needed for proving convergence of loss to zero. This is a generalization of the zero-asymmetric initialization (setting $\bar W_1 = \dots = \bar W_{L-1} = I_d$ and $\bar w_L = 0$) proposed by [Wu et al., 2019]. We have added a remark on this after Theorem 7.
>
> Re. 2: What you pointed out is correct; the classification experiments rather suggest that finite-horizon behavior of gradient flow is different from the infinite horizon limit. We included these classification experiments because they suggest an important future direction of research, rather than as corroboration of our theoretical results on classification.
>
> Re. 3: The motivation for studying the single data point lies in the fact that we can analyze any 2-layer networks that can be represented in the linear tensor network formulation. Although single point is restrictive, we believe our analysis provides insights for understanding networks that do not satisfy the orthogonal decomposition assumption (Assumption 1), e.g., convolutional networks with small filters. Please recall that such architectures have not been analyzed in the existing results.
>
> Again, we appreciate the reviewer for the time and efforts. Please let us know if you have any other comments/questions.
>
> Best,
> Authors

---

### Official Review · AnonReviewer4 · 2020-11-02
**Tensor formulation to unify implicit bias results on linear networks**

**Rating:** 7
**Confidence:** 4

**Review:**

The paper proposes a tensor formulation as a unified framework for studying different architectures of neural networks. This formulation is mainly used under the special case of linear networks with “orthogonal” tensor representation to provide a unified proof of many existing results on the implicit bias of gradient descent under different architectures. Overall, the tensor formulation is an interesting and new (to my knowledge) abstraction.

On the other hand, in terms of writing, I found the key ideas behind the unified results regarding implicit bias to be a bit obscured within the tensor formulation. Since this paper is positioned as a new formulation with a goal of providing useful generalized abstraction, I believe it would be beneficial to be more explicit about the core ideas and their connections to existing concepts. In particular, have the following concrete suggestions:

1. Although the tensor formulation is more general, introducing the results including the orthogonality condition for the case of L=2 layers first would be very useful. For 2 layers the analysis only requires matrix algebra and it is easy to motivate the generalization and identify the key unifying concepts.

2. The condition on orthogonality of tensor formulation, which unifies existing results on diagonal and convolutional networks have an interesting connection to linear diagonal networks: I believe the condition is equivalent (in the sense of if and only if) to stating that there is exists *some* orthonormal transformation of the layer parameters v_l (v_l -> U_lv_l) and some linear transformation of input x (x->Sx), such that after the transformations, the networks behaves like a linear diagonal network. Thus, at the core, networks with orthogonal tensor representation are equivalent diagonal networks in a different parametrization. Moreover, from the proof, one can also show that this specific of change in parametrization does not affect the gradient flow path, i.e, that gradient flow on v_l is equivalent to gradient flow on the orthonormal linear transformation of the parameters tilde{v}_l=U_lv_l. It would be useful to discuss the connection between the orthogonality condition and diagonal networks explicitly in the text as a motivation/clarification for the orthogonality condition or as proof sketch.

2b. Another connection is that, for L=2 where M(x) is a matrix, the orthogonality condition is also equivalent to commutativity of M(x_i) for the dataset inputs x_i -- this also highlights why for a single datapoint the analysis goes through as M(x) is a single matrix, which is commutative with itself.

3. One of the main differences from the existing work is claimed to be the fact that this analysis does not make assumptions on convergence of loss. However, the main theorems introduce additional assumptions on initialization that effectively guarantee convergence of loss. It is not clear if this assumption is more useful than assuming the loss converges --  the assumption on initialization is more easy to control, however such constraints on initialization in turn affect the implicit bias explicitly. For e.g., in regression, the effect is explicit in the definition of Q, and for classification although the asymptotic direction is independent of initialization, recent work on diagonal networks show that the initialization has strong effects on the convergence to the asymptotic max margin solution (see https://arxiv.org/abs/2007.06738).

---

> ### Author Response · Authors · 2020-11-20
> **Author Response to Reviewer 4**
>
> We appreciate the reviewer for the feedback and insightful comments. We will give responses to the suggestions by the reviewer. Please also take a look at our revised submission.
>
> Re. 1: Thanks for the good suggestion. Due to space constraints, we could not add separate statements for $L=2$; however, as you can see in our revision, we have added comments explaining the $L=2$ cases after (6), in the beginning of Sec 4.2, and after Assumption 1.
>
> Re. 2: After the statement of Theorem 2, we have also added the connection of orthogonal decomposition (Assumption 1) and linear diagonal networks. Thank you for the suggestion.
>
> Re. 2b: We believe there is a slight misunderstanding here. Assumption 1 does not necessarily mean that, in $L = 2$, two data tensors $M(x_1)$ and $M(x_2)$ commute. For example, in case of full-length linear convolutional networks ($d = 3$), we have
>
> $M(x) = \\begin{bmatrix} [x]_1 & [x]_2 & [x]_3 \\\\ [x]_2 & [x]_3 & [x]_1 \\\\ [x]_3 & [x]_1 & [x]_2 \\end{bmatrix}$,
>
> and one can quickly check that $M([1,0,0])$ and $M([0,1,0])$ do not commute. Instead, the key to Assumption 1 is that the singular vectors (in $U_l$’s) of the data tensor $M(x)$ are fixed and independent of the data point $x$. This makes our proofs (of Theorems 2 and 5) work, and it is also the reason why the single point analyses (Theorems 3 and 6) work for networks that do not satisfy Assumption 1.
>
> Re. 3: We believe proving (instead of assuming) convergence of loss is meaningful because there are known examples where linear neural networks never converge to the true solution under some initializations [Bartlett et al., 2018; Arora et al., 2019a]. Given such examples, it is not clear a priori whether a given initialization direction will satisfy the convergence assumption. In contrast, our theorems provide sufficient conditions on initialization for convergence, which is easier to check. Please also revisit Remark 1. Of course, initialization has a direct impact on the implicit bias of gradient flow, as you correctly point out; however, this impact is also present in other existing results assuming convergence of loss to zero, e.g., Theorem 1 (General case) of [Woodworth et al., 2020] also shows the effect of initialization $w_0$ in their $Q_{\alpha, w_0}$, similar to our results.
>
> Thanks again for your time and efforts in reviewing our paper. Please do let us know if you have any further comments/questions.
>
> Best,
> Authors

---

### Author Response · Authors · 2021-09-14
**Post-conference paper update**

Dear all,

We would like to announce that we made a revision of the paper after the ICLR 2021 camera-ready version on this page.
For the newest version, please see: https://arxiv.org/abs/2010.02501

Thank you,
Authors

---

### Decision · Program_Chairs · 2021-01-07
**Final Decision**

**Decision:**

Accept (Poster)

**Comment:**

This paper suggests an extension of previous implicit bias results on linear networks to a tensor formulation and arguably weakens some of the assumptions of previous works (e.g. loss going to zero is replaced with initialization assumptions). The reviewers were all positive about this work, saying it is clearly written and an original significant contribution. There were a few issues raised (e.g. the novelty of the proof techniques) and the authors responded. The reviewers did not clarify if this response satisfied these concerns, but did not change their positive scores. I will take this to indicate they still recommend acceptance.